# Intrinsic macroscale oscillatory modes driving long range functional connectivity in female rat brains detected by ultrafast fMRI

Joana Cabral [1,2,3] ✉, Francisca F. Fernandes [1] & Noam Shemesh [1] ✉

Spontaneous fluctuations in functional magnetic resonance imaging (fMRI) signals correlate across distant brain areas, shaping functionally relevant intrinsic networks. However, the generative mechanism of fMRI signal correlations, and in particular the link with locally-detected ultra-slow oscillations, are not fully understood. To investigate this link, we record ultrafast ultrahigh field fMRI signals (9.4 Tesla, temporal resolution = 38 milliseconds) from female rats across three anesthesia conditions. Power at frequencies extending up to 0.3 Hz is detected consistently across rat brains and is modulated by anesthesia level. Principal component analysis reveals a repertoire of modes, in which transient oscillations organize with fixed phase relationships across distinct cortical and subcortical structures. Oscillatory modes are found to vary between conditions, resonating at faster frequencies under medetomidine sedation and reducing both in number, frequency, and duration with the addition of isoflurane. Peaking in power within clear anatomical boundaries, these oscillatory modes point to an emergent systemic property. This work provides additional insight into the origin of oscillations detected in fMRI and the organizing principles underpinning spontaneous long-range functional connectivity.

Spontaneous fluctuations in signals detected with functional Magnetic Resonance Imaging (fMRI) correlate across spatially distributed brain areas forming functional networks that appear disrupted in numerous psychiatric and neurological disorders, pointing to a key role in brain function[1–5]. However, the organizing principle driving long-range correlations between brain areas remains unclear.

A wide range of low-rank decomposition techniques have been put forward to characterize the spatial organization of spontaneous fMRI signal fluctuations, including among others independent component analysis[6,7], co-activation patterns[8–11], low-dimensional gradients[12,13], leading eigenvector dynamics analysis[14,15], dynamic mode decomposition[16] and quasi-periodic patterns (QPPs)[17,18]. Despite the differences inherent to each technique, most methods converge in a discrete repertoire of intrinsic modes exhibiting features of

stationary wave patterns, where correlated activity is detected among spatially distributed regions (or poles), with gradually varying phase relationships across space[19]. These intrinsic modes have been shown to emerge transiently and recurrently during rest[20,21], to be selectively recruited during specific tasks[7], and to replicate across mammals[10,22–25].

In the frequency domain, correlated fluctuations in fMRI signals exhibit power at ultra-slow frequencies, peaking typically below 0.1 Hz in human brains at rest, although intrinsic functional networks have been detected at frequencies extending even beyond 0.5 Hz[26–30]. Crucially, it remains unclear whether the spectral power at low frequencies is associated solely with aperiodic activations of the characteristically slow and region-specific hemodynamic response function or additionally reflects the existence of damped oscillatory components[16,30–32]. Mainly detected with electro- and magnetoencephalography (EEG/

[1]Preclinical MRI Lab, Champalimaud Research, Champalimaud Foundation, Lisbon, Portugal. [2]Life and Health Sciences Research Institute, School of Medicine, University of Minho, Braga, Portugal. [3]ICVS/3B's - Portuguese Government Associate Laboratory, Braga/Guimarães, Portugal. ✉e-mail: joanacabral@med.uminho.pt; noam.shemesh@neuro.fchampalimaud.org

MEG), macroscale oscillatory components in brain activity have been targeted by neural field theories, demonstrating how the frequency spectrum and correlation structure can be predicted from brain geometry[33–35]. Although intrinsic modes detected with fMRI have been shown to spatially align with eigenmodes of brain structure (either from surface geometry or diffusion networks), theoretical predictions of mode-specific temporal responses remain to be adequately addressed in fMRI[36–39]. Given recent insights demonstrating that the fMRI signals underpinning intrinsic networks relate to macroscopic waves of propagating activity[16,40–42], it is crucial to obtain a detailed characterization of the modes' spatial and temporal signatures to empirically investigate their oscillatory nature and their link with neural field theories.

Studies in rodents and humans have shown that some ultra-slow frequency components in fMRI signals have a periodic nature and are coupled with electrophysiological and electroencephalographic (EEG) signals[43–46]. These periodic fluctuations have been proposed to be linked to arteriole vibrations entrained by fast oscillations in local field potentials, pointing to a potentially more direct relationship with the underlying neural activity[47,48]. Still, how these oscillations organize at the macroscopic level and their relationship to functional connectivity between brain areas remains unclear.

Intrinsic networks analogous to the ones identified in humans have been identified in rats and are modulated by the sedation/anesthesia state[10,49–52]. In particular, sedation with low doses of medetomidine has been shown to reveal consistent intrinsic networks but also to drive abnormal high amplitude oscillations in fMRI signals at frequencies extending beyond 0.1 Hz[44,50]. The addition of isoflurane at low concentrations suppresses these high-frequency oscillations while maintaining the typical human resting-state frequencies <0.1 Hz, such that the combination medetomidine/isoflurane (MED/ISO) is currently the state-of-the-art protocol to approximate resting-state brain activity in rats[53,54].

The solid evidence from rat experiments across sedation/anesthesia protocols offers an ideal setting to analyze the spatiotemporal organization of fMRI signal oscillations and their relationship with intrinsic network patterns[54–56]. However, the precise characterization of oscillations across space and over time is complex and benefits from an adequate spatiotemporal resolution and high signal-to-noise (SNR) ratio to adequately capture transient phase relationships between voxels. Ultra-high field fMRI studies in rats achieve increased SNR by attenuating thermal noise using cryogenic coils[57,58]. Moreover, for increased precision in the characterization of oscillatory signals, long scanning times are needed to ensure high-frequency specificity at slow frequencies, and fast sampling helps prevent frequency aliasing from undersampled periodic components of physiological and/or scanner artifacts. At the spatial level, a large field of view is necessary to capture macroscale organization, while ensuring a sufficient spatial resolution to resolve distinct brain regions.

Therefore, we harness an ultrafast ultrahigh field fMRI approach, with long scan durations of 10 min sampled at 38 ms resolution (16,000 frames per scan) to characterize the spatial organization of oscillations detected in fMRI signals in a single slice of the rat brain, achieving high SNR ratio via a 9.4 T magnetic field and a cryogenic coil. This approach exposes unreported features of rat brain activity, providing insights into the fundamental organizing principles driving long-range functional connectivity in the brain.

## Results

### Long-range functional connectivity

A typical seed-based functional connectivity analysis was performed to confirm the detection of long-range correlations in fMRI signals in the range of frequencies typically considered in resting-state studies, i.e., ranging between 0.01 and 0.1 Hz Fig. 1a, b. In panel c, we plot the band-pass filtered fMRI signals in the same 3 seeds together with their contralateral voxels (cf. Supplementary Fig. S1 for the corresponding

brain atlas). When applying sliding-window correlation (SWC) analysis, fluctuations were detected between sustained periods of positive correlation (yellow shades) and periods of weak or even negative correlation (orange to magenta shades). The same analysis was applied to a scan from a post-mortem rat to ensure that the periods of sustained correlations are detected at levels beyond any conceivable artifacts (Supplementary Fig. S2).

### Space-frequency analysis of fMRI signals across conditions

To investigate whether the transient correlations are associated with oscillatory phenomena, we turn to analyze the spectrum of frequencies detected in brain voxels across three different sedation/anesthesia protocols: under medetomidine only (which we term sedation), after the addition of isoflurane at 1% concentration (light anesthesia), and after increasing isoflurane concentration to 3% (deep anesthesia) (see the "Methods" section for details). Applying a space-frequency analysis on 36 ultrafast fMRI scans (12 per condition, each 10 min long at 26.3 Hz sampling rate), the 3 anesthesia conditions are compared in terms of average power at different frequency bands with respect to a baseline defined from postmortem scans (Fig. 2). Power at frequencies up to 0.30 Hz—extending well beyond the range typically considered in resting-state studies—is detected in the brains of sedated and lightly anesthetized rats significantly above deep anesthesia levels (Fig. 2b, c and Supplementary Fig. S3).

Power in fMRI signals is found to peak within well-defined cortical boundaries consistently between 0.20 and 0.25 Hz in rats sedated with medetomidine (Fig. 2). Voxels in the striatum (subcortical) exhibited power at frequencies peaking between 0.1 and 0.15 Hz. The addition of isoflurane at 1% specifically affects the power of fMRI signal fluctuations between 0.15 and 0.25 Hz, whereas isoflurane at 3% significantly decreases the power in the broad frequency range between 0.05 and 0.3 Hz (Fig. 2b, c and Supplementary Figs. S3–S6). Above 0.40 Hz, only signatures associated with cardiac and respiratory frequencies are detected (Supplementary Fig. S7 and Table S1). Ultra-slow fluctuations below 0.05 Hz, i.e., with a period longer than 20 s, persist even in deeply anesthetized animals. This change in frequency across conditions is clearly visible in the carpet plots in Supplementary Figs. S6–S9.

### Spatial, temporal, and spectral properties of principal components

While the space-frequency analysis provides information about which voxels have more power in each frequency band, it does not reveal how the signals evolve in time or organize in space. As can be seen in Supplementary Movies 1 and 2, fluctuations are not globally correlated but instead exhibit complex phase relationships across space that appear recurrent over time and consistent across different rats in the same condition. Signals co-varying in phase across distant voxels symmetrically aligned with respect to the vertical midline point to a link with long-range functional connectivity. In deep anesthesia, despite applying the same filtering, no particular spatial organization or fine structure is detected except for ultra-slow globally correlated fluctuations.

To detect whether the fluctuations have a characteristic spatial organization, we extract the principal components of the fMRI signals filtered below 0.3 Hz in each condition (e.g., the eigenvectors of the covariance matrix, see Fig. 3a–c and see the "Methods" section for details). Principal component analysis has the advantage of returning orthogonal modes of covariance without making any assumption regarding the oscillatory properties of the components, unlike other decomposition techniques that a priori assume an oscillatory nature of the components, such as dynamic decomposition analysis[16,59]. The spatial patterns associated with the 10 principal components detected above the postmortem baseline reveal spatially fixed phase relationships between the fMRI signals in distinct brain subsystems (Fig. 3d). These phase relationships varying gradually and symmetrically across space exhibit characteristics of standing waves, where regions of

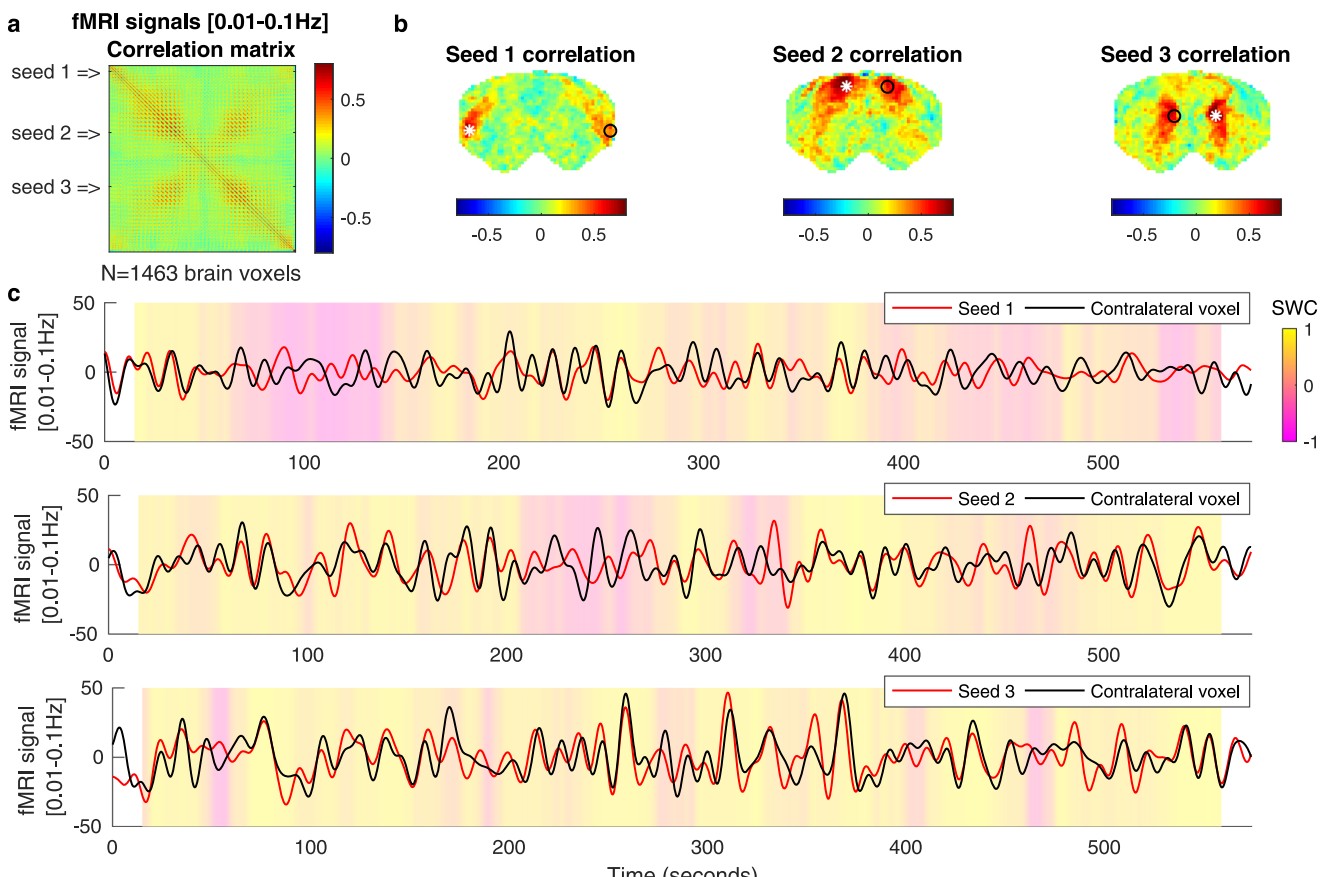

**Fig. 1 | Static and dynamic resting-state functional connectivity analysis in band-pass filtered fMRI signals. a** Correlation matrix of the fMRI signals in all voxels within the brain mask, bandpass filtered in a range typically considered in resting-state studies, i.e., 0.01–0.1 Hz (no nuisance regressor nor spatial smoothing applied). Each line/column in the matrix corresponds to the correlation map of each voxel. **b** Seed-based correlation maps are represented for three different seeds (white asterisks), where each voxel is colored according to its degree of correlation with the seed. A voxel contralateral to each seed is represented by a black circle. All color bars are truncated between −0.8 and 0.8. **c** Filtered fMRI signals were recorded in each seed (red) and corresponding contralateral voxel (black). Colored shades represent the sliding window correlation (SWC) using a 30-s window, showing that the correlation is not constant but fluctuates between transients of long-range phase locking. The same figure obtained from a post-mortem scan is reported in Supplementary Fig. S2.

strong amplitude (red/blue) represent the anti-nodes of the wave, whereas regions of low amplitude between anti-nodes (green) represent the wave's nodes (points of no motion).

To investigate the dynamic behavior of the principal components, we subsequently analyze the temporal signatures associated with each spatial pattern. In particular, we aim to verify the existence of periodicity between positive and negative representations of the spatial patterns (Fig. 3d), which is not necessarily a property of principal components, given that signals can co-vary aperiodically. The temporal signature $\tau_\alpha^S(t)$ of each principal component $\alpha$ for each scan $S$ is obtained by performing a matrix multiplication that contracts the 'n' dimension as

$$\tau_\alpha^S(t) = \psi_\alpha(n)\, \Psi^S(n,t), \tag{1}$$

where $\psi_\alpha(n)$ represents the spatial pattern of each principal component $\alpha$ and $\Psi^S(n,t)$ represents the activity recorded with fMRI across all voxels $n$ and timepoints $t$ for scan $S$ (Fig. 3e).

The reconstruction

$$\Psi^R(n,t) = \sum_\alpha \psi_\alpha(n)\tau_\alpha^S(t) \tag{2}$$

describes the linear superposition of a basis of wave patterns locked in space $\psi_\alpha(n)$ and evolving in time $\tau_\alpha^S(t)$. Video 2 illustrates how the

essential macroscopic dynamics of the recorded fMRI signals in a rat brain are captured using a low-rank approximation considering only the reduced common basis of 10 principal components $\psi_\alpha(n)$ detected across sedated rats, and the 10 scan-specific temporal signatures $\tau_\alpha^S(t)$ associated with these components. As observed in Fig. 4 and Supplementary Movie 3 (still frame reported in Fig. 4), the principal components are found to oscillate around the mean with slowly fluctuating amplitudes, generating patterns akin to those of transiently resonating stationary waves.

The detection of oscillations associated with the spatial patterns benefited from the fast sampling combined with long scan durations (totaling 16,000 images per 10-min scan), by preventing frequency aliasing from physiological rhythms (i.e., with Nyquist frequency above breathing and cardiac frequencies) and by ensuring sufficient resolution in the power spectrum at low frequencies, i.e., with precision below 0.01 Hz (see Supplementary Figs. S10, S11). As shown in Fig. 5 (top row), when projecting the $1 \times N$ spatial component (here $\psi_{\alpha=7}$) on the $N \times T$ ultrafast unfiltered fMRI signals, the temporal signature $\tau_{\alpha=7}^S$ exhibits clearly visible oscillations between positive and negative representations of the spatial pattern. As the sampling factor is increased (Fig. 5 bottom rows), even if a resonant peak frequency can still be detected, the signal to noise ratio is decreased. Indeed, we find that the principal component $\psi_{\alpha=7}$ fails to be detected with a sampling as fast as 380 ms, which coincides with the sampling rate at which the breathing frequency

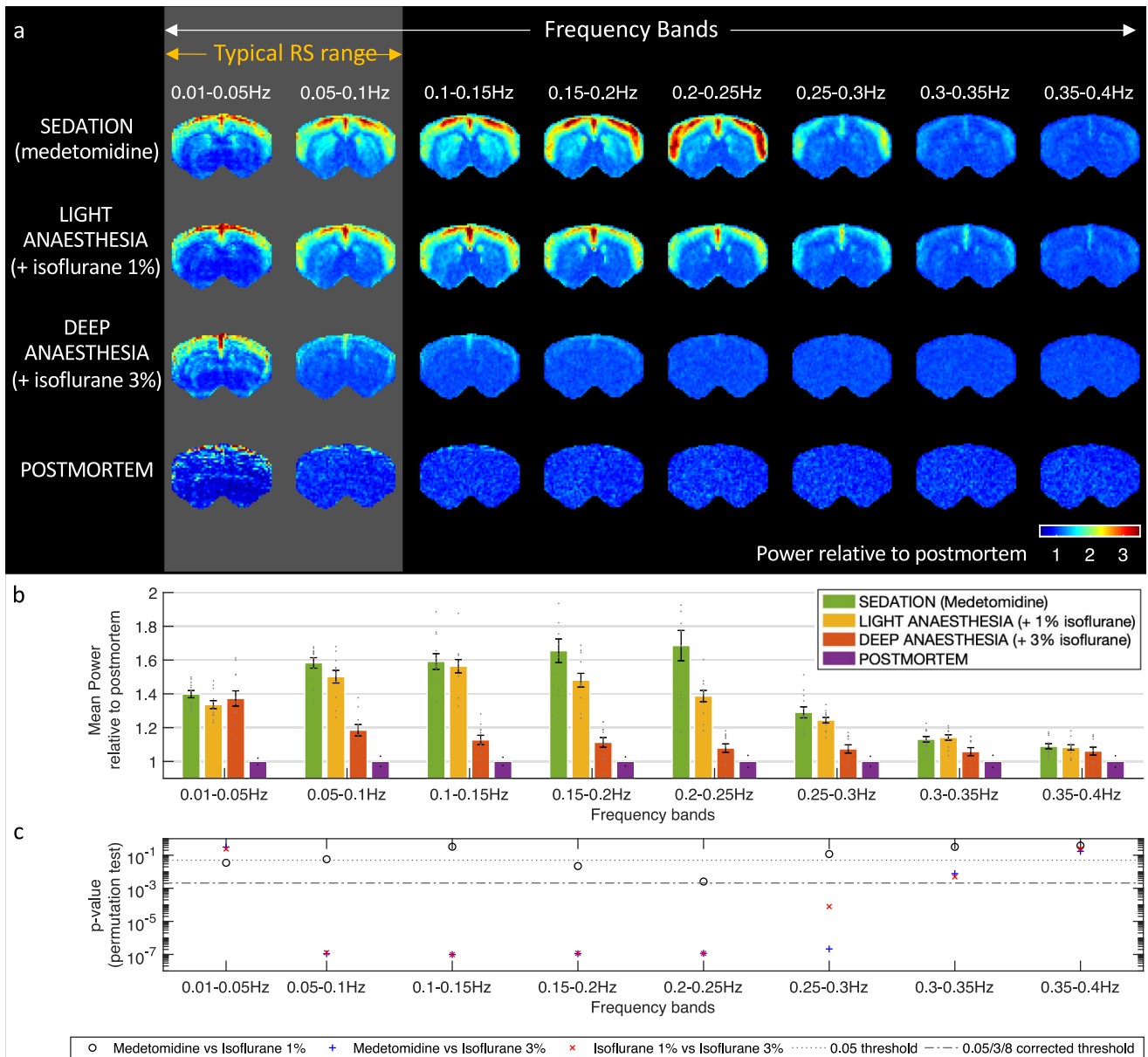

**Fig. 2 | Spectral power of fMRI signals differs significantly between anesthesia conditions. a** Spatial maps of spectral power in 8 non-overlapping frequency bands, averaged across the $s = 12$ fMRI scans recorded under each anesthesia conditions and normalized by the mean power in $s_{PM} = 2$ postmortem scans. fMRI recordings were recorded from $n = 6$ genetically similar female rats, each scanned twice in 3 anesthesia conditions (with isoflurane at 0%, 1%, and 3%). **b** Power in 8 frequency bands averaged across all brain voxels relative to the mean power in 2 postmortem scans. Dots correspond to the mean power in each scan and error bars represent the mean ± standard error across the $s = 12$ scans. **c** Two-sided $p$-values obtained from 10,000 $t$-tests on randomly permuted data comparing the power in $s = 12$ scans over the 3 anesthesia conditions and in each of 8 non-overlapping frequency bands, reported with respect to the standard threshold of 0.05 and the Bonferroni-corrected threshold of 0.0021. RS resting-state. $p$-values are reported in Supplementary Fig. 3. Source data are provided as a Source Data file and Source Codes are provided in Supplementary Material.

(~2 Hz) cannot be adequately resolved given the Nyquist theorem (analysis shown in Supplementary Figs. S12–S16). In Supplementary Fig. S15 we reorder the spatial patterns either using a spatial randomization or sorting according to a left-right gradient, and demonstrate that the temporal signatures exhibit lower amplitudes and less spectral power when the spatial organization of the phases in wave patterns is disrupted.

With the addition of isoflurane at 1% concentration (Fig. 6 top), the repertoire of principal components is modified, not only in number—with only 6 components detected above the postmortem baseline—but also in terms of spatial configuration, with different brain sub-systems oscillating in phase or anti-phase with each other. Regarding

the temporal signatures associated with the different components, although some transient oscillations can still be detected, these last visibly shorter and with different periodicity over time (Fig. 6e), which is reflected in a broader distribution of power across the spectrum, peaking at lower frequencies with respect to the sedation condition (Fig. 6f).

When the concentration of isoflurane is further increased to 3% (Fig. 6 bottom), the variance above postmortem baseline is explained by a single principal component where the cortex and striatum oscillate together in phase (Fig. 6d') and at very low frequencies (Fig. 6f'). These ultraslow global fluctuations are particularly visible in the carpet plot in Fig. 6b'.

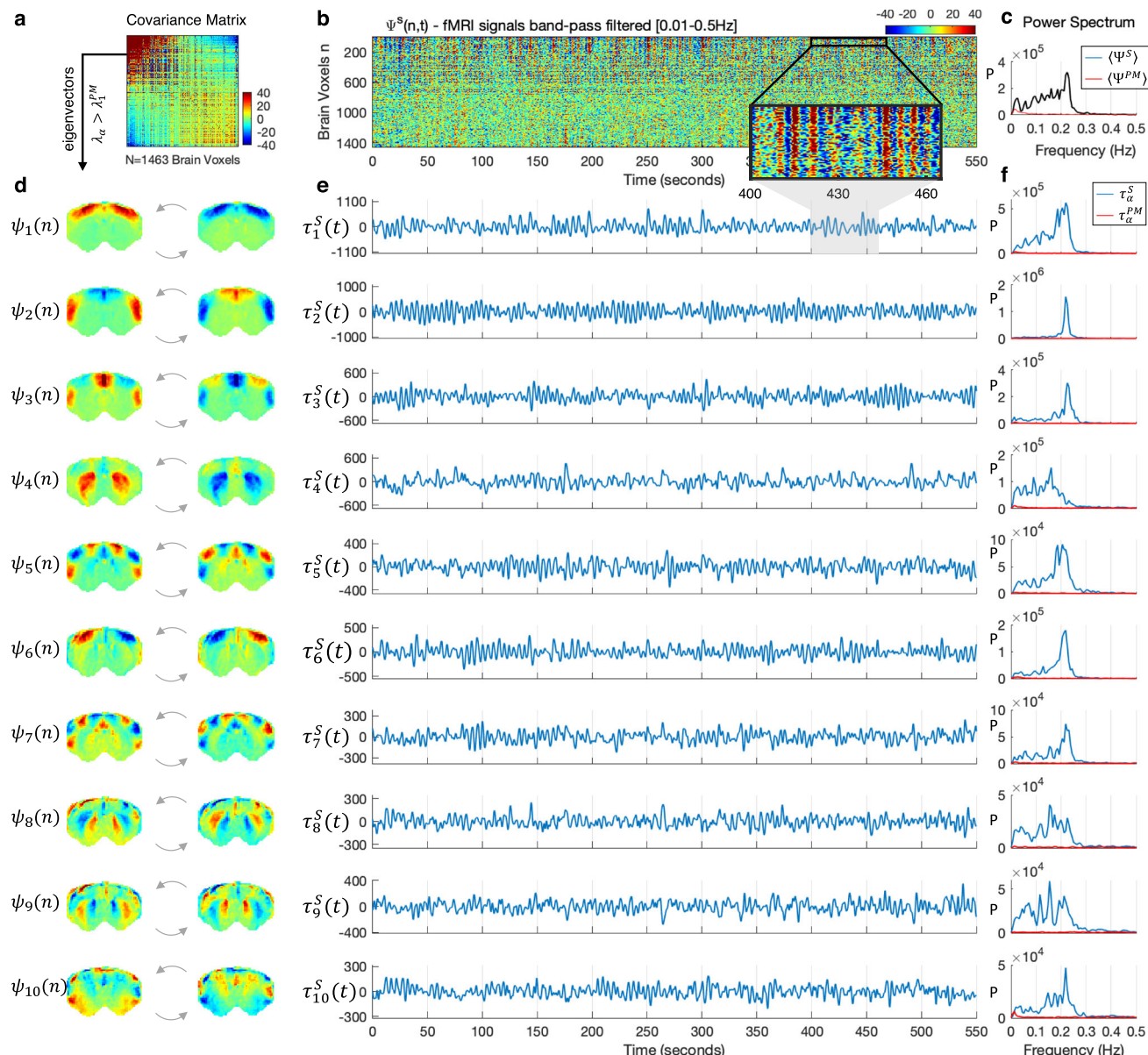

**Fig. 3 | Spatial, temporal, and spectral signatures of the principal components in medetomidine-sedated rats. a** The NxN covariance matrix of fMRI signals (filtered within the range where significant spectral power was detected in the cortex, i.e., between 0.01 and 0.3 Hz) averaged across the 12 sedated rat scans. **b** Carpet plot of the fMRI signals in all brain voxels, $n$, over time, $t$, represented by the wave function $\Psi^S(n,t)$, here shown for a representative scan $S$ of a sedated rat in the frequency range [0.01–0.5 Hz]. Voxels are sorted according to the elements in the largest magnitude eigenvector $\psi_1$. Values correspond to fMRI signal change with respect to the mean in each voxel. A zoom into the first 100 voxels over 60 s is inserted to illustrate oscillations in the signals. **c** Power spectrum of the mean fMRI signal across voxels for: (black) the scan shown in **b** and (red) a scan performed postmortem (PM). **d** The 10 principal components $\psi_\alpha$ obtained as the eigenvectors from (**a**) with eigenvalue $\lambda_\alpha$ above PM baseline are scaled by 1 (left) and −1 (right) to illustrate the activity pattern when the temporal signature oscillates between positive and negative values. **e** Temporal signature associated with each of the 10 principal components given by $\tau_\alpha^S(t) = \psi_\alpha(n)\Psi^S(n,t)$ for the same scan shown in (**b**). Clear oscillations with fluctuating amplitude can be observed. **f** Power spectra of the temporal signatures from **e** (blue) and in a postmortem scan (red). See Supplementary Movie 2 to observe the behavior of each principal component over time.

## The oscillatory nature of principal components

While in Figs. 3 and 6 we show the results from one representative animal, in Fig. 7 we report the peak frequency and stability of the oscillations associated with each principal component in each of the 36 scans, i.e., of the 6 rats scanned twice in each condition. The stability of the oscillations is assessed from the resonance $Q$-factor, which is proportional to the number of cycles before the amplitude decays to ~37% ($e^{-1}$) of its initial value, consisting of the ratio between the peak frequency and the power spectrum's full-width-at-half-maximum (FWHM). Both the peak frequencies and the $Q$-factors were found to be significantly higher (and with larger variability across scans) in

sedation and light anesthesia with respect to deep anesthesia (Bonferroni-corrected $p$-values reported in Supplementary Table S2).

The autocorrelation functions of the wave temporal signatures (here shown for $\tau_1^S(t)$ in each condition) illustrate that the number of sustained cycles before the amplitude decays to $1/e$ decreases with increasing levels of isoflurane, as estimated by the $Q$-factor (Fig. 7c). We use the Hilbert transform to obtain a representation of the autocorrelation functions in the complex domain (with real and imaginary components) and plot the corresponding phase portraits (Fig. 7c bottom). The representation of the phase portraits serves to classify the temporal signatures of the components within the framework of

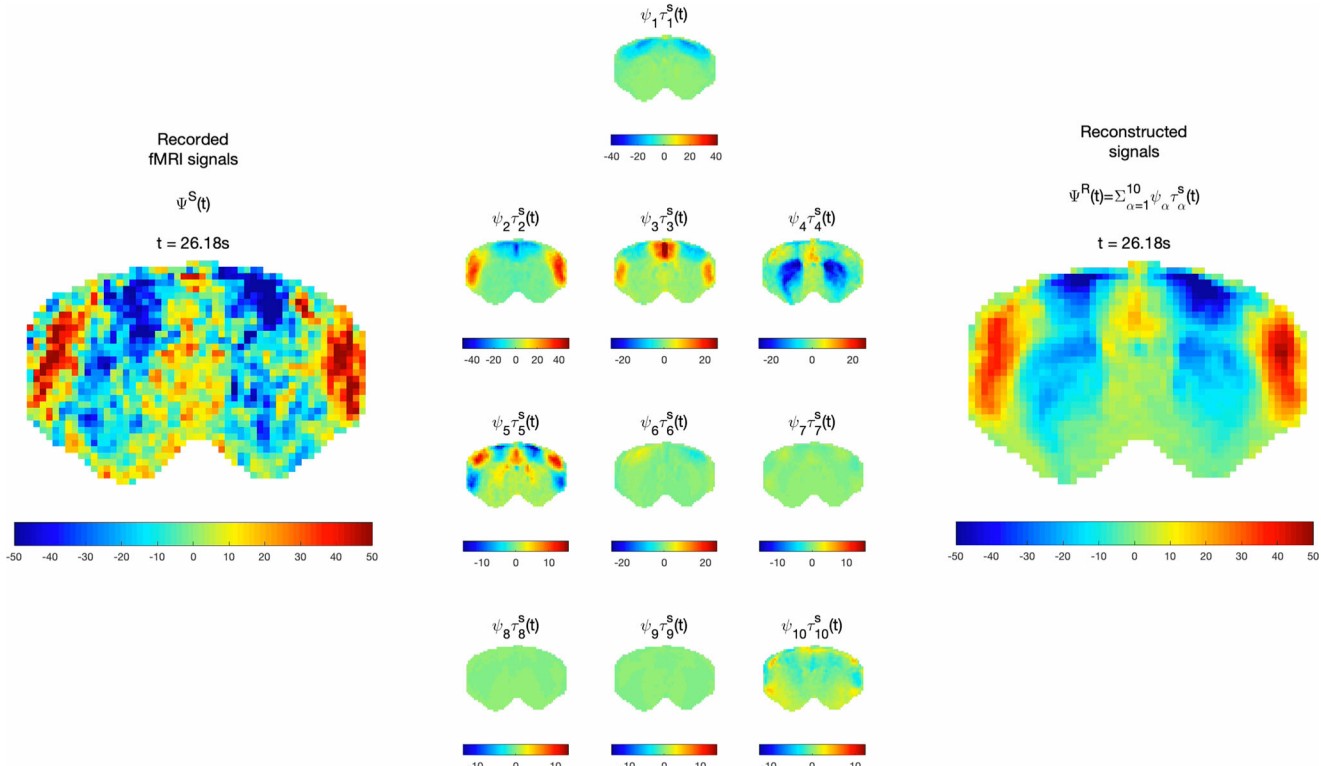

**Fig. 4 | Recorded signals are reconstructed as the linear superposition of 10 condition-specific principal components with scan-specific temporal signatures.** This image is a still frame from Supplementary Movie 3. (Left) fMRI signals in $N = 1463$ brain voxels band-pass filtered between 0.01 and 0.3 Hz recorded from a representative rat under medetomidine only. Middle) Each of the 10 spatially defined principal modes of covariance is scaled over time by its corresponding temporal signature in scan $S$ to illustrate the standing wave dynamics. (Right) The signals recorded in scan $S$ are reconstructed as the linear sum of the 10 principal components multiplied by their corresponding temporal signature in scan $S$. To account for differences in power across components, colorbar limits are set to $\pm 4$ standard deviations of the corresponding temporal signatures.

dynamical systems stability theory, demonstrating that the components have a 'spiral sink' trajectory back to equilibrium according to the Poincaré diagram[60].

## Stochastic resonance of standing waves

The presence of a spiral sink in the autocorrelation function of a dynamical system is indicative of underdamped oscillatory motion, where the system returns to a fixed-point equilibrium upon perturbation with an oscillation with a decaying amplitude. An underdamped system will resonate at its natural frequency either when perturbed at a natural frequency or in the presence of background noise due to stochastic resonance (see Supplementary Fig. S16 for an illustration).

Given that the principal components detected in rat brain activity have spatial features of standing waves (in line with previous studies) and, as we demonstrate here, exhibit transient oscillations over time, it can be hypothesized that their phenomenology is associated with the stochastic resonance of standing waves. In such a mechanistic scenario, the differences detected across conditions can be further hypothesized to be related to alterations in the properties of the medium through which the waves propagate, while the anatomical structure remains unchanged. Indeed, while medetomidine is found to increase the number, peak frequency and $Q$-factor of resonant modes, isoflurane is found to gradually dampen the resonant modes of the system, with only global aperiodic fluctuations being detected under deep anesthesia (Fig. 7).

To demonstrate that the stochastic resonance of stationary wave patterns can generate the patterns of intrinsic functional connectivity observed experimentally, we model the signals in the brain slice as the superposition (i.e., linear sum) of modes whose spatial configuration $\psi_\alpha(n)$ is fixed and given by the principal components detected

empirically, and the temporal signature $Z_\alpha(t)$ is obtained using the Stuart–Landau equation to simulate the behavior of an oscillator in the underdamped regime in the presence of background noise as

$$\psi^{\text{Model}}(n, t) = \sum_\alpha \psi_\alpha(n) Z_\alpha(t), \tag{3}$$

with

$$dZ_\alpha/dt = Z_\alpha(i\omega_\alpha - |Z_\alpha|^2 + a) + \beta\eta, \tag{4}$$

where $\omega_\alpha$ is the resonant frequency of each mode, $a$ (negative) scales the decay rate[61] and $\eta$ is the added Gaussian white noise $\eta$ with standard deviation $\beta$.

As shown in Fig. 8, the stochastic resonance of a repertoire of standing waves (here considering the repertoire detected empirically in sedated animals) results in a spatiotemporal pattern sharing features with what is detected from fMRI recordings. This model includes the possibility to tune the oscillators in the overdamped regime, in which case it can approximate the results obtained in deeply anesthetized animals, where no resonant oscillations are detected but only aperiodic fluctuations. In other words, the model of stochastic resonance does not exclude the hypothesis of scale-free fluctuations driving the fMRI signals, but it considers it to be a particular case where the oscillatory modes are overdamped.

In Fig. 8d, we show snapshots of activity generated from the superposition of standing waves resonating in the presence of background noise to illustrate the multiplicity of patterns that can be generated at the instantaneous level, as observed in empirical recordings. Finally, to link with long-range functional connectivity, we compute the correlation matrix of the simulated signals,

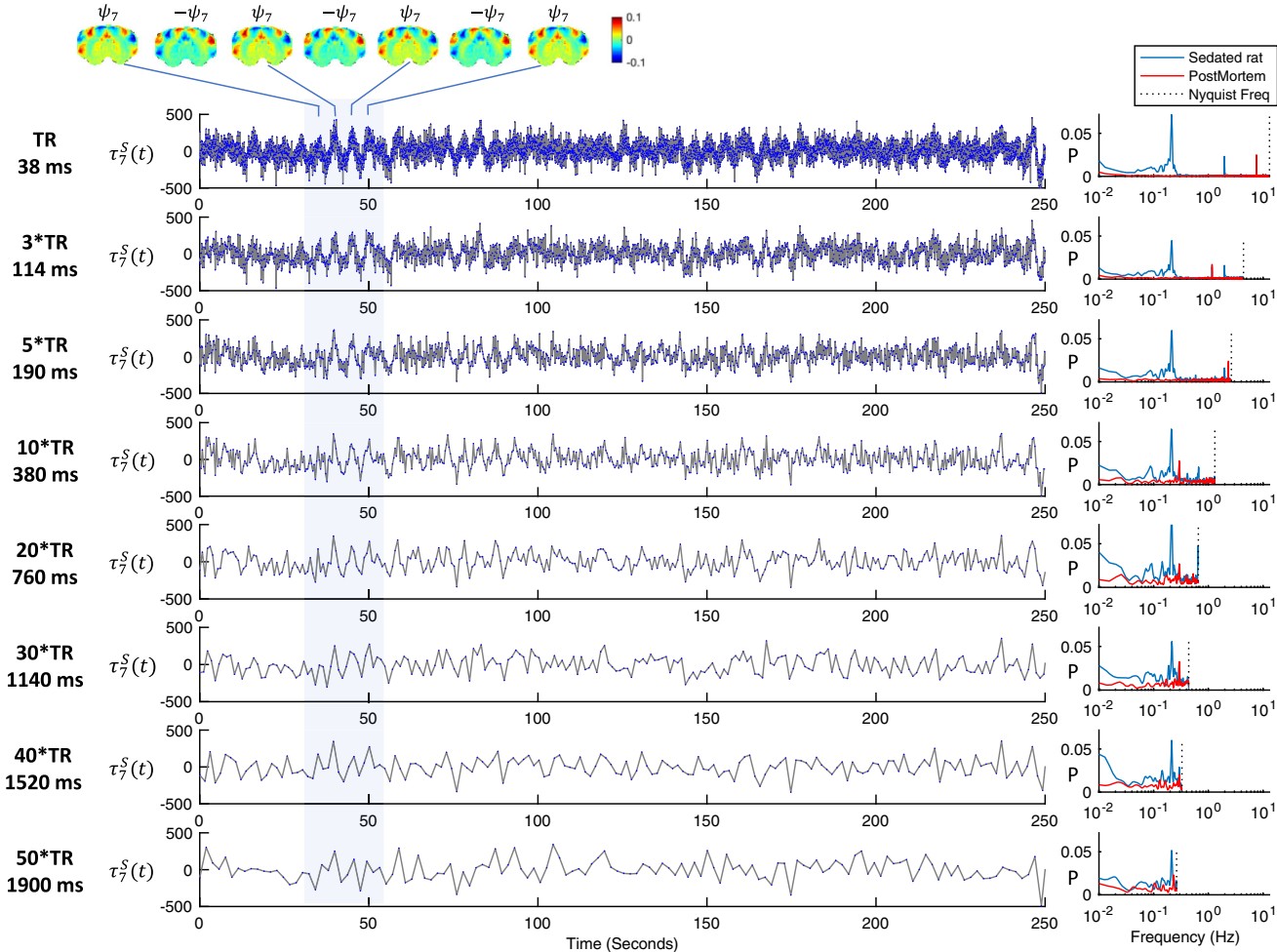

**Fig. 5 | Effect of the sampling rate in the power spectrum given a fixed scan duration.** Left: The unfiltered temporal signal $\tau_7^S(t)$ associated with the 7th principal component detected in ultrafast fMRI signals from medetomidine-sedated rats (Time of Repetition, TR = 38 milliseconds, ms) is downsampled by considering only one in every 3, 5, 10, 20, 30, 40 and 50 frames (corresponding to intervals of 114, 190, 380, 760, 1140, 1520 and 1900 ms between frames). Plots are shown for 250 s from a representative scan $S$ (same as Fig. 3). Right: The power spectral density (PSD) of the sampled signals computed for scan $S$ (blue) and for a scan performed postmortem (red). For each downsampling factor, both PSD (red and blue) are normalized by the total power in the postmortem scan. PSDs are computed over the entire scan duration of 590 s.

demonstrating that the stochastic resonance of standing waves is a possible mechanism to generate correlations between contralateral brain regions located at the wave antinodes.

## Expansion to the whole-brain level

To expand our results obtained in a single slice to the whole-brain level, the principal components were obtained from six 15-min-long fMRI scans covering 12 brain slices of 3 rats sedated with medetomidine. Despite the necessarily lower temporal resolution of multi-slice acquisitions (here TR = 350 ms), oscillations can still be observed (see Supplementary Movie 4), organizing with phase relationships that overlap (particularly in slice 6) with those detected in the frontal slice of ultrafast fMRI recordings, supporting the hypothesis that the conclusions drawn from the single slice ultrafast acquisitions can be expanded to the whole-brain level (Supplementary Figs. S17–S19). However, although consistent principal components were detected at the spatial level (the first 5 are rendered in a transparent brain in Fig. 9a), the limited temporal resolution and the added artifacts resulting from multi-slice acquisitions were found to reduce the sensitivity to transient oscillations such that even in the fMRI scan that exhibited most

power > 0.15 Hz, the sensitivity to frequency-specific oscillations is much lower than the one observed in single-slice acquisitions (Supplementary Fig. S19).

In summary, our experiments revealed that: (i) power at frequencies extending up to 0.3 Hz is consistently detected in the fMRI signals from rat brains, peaking in power in the cortex of rats sedated under medetomidine; (ii) fMRI signal fluctuations organize into a discrete repertoire of modes with fixed phase relationships across space; (iii) high sampling rates allow detecting transient fine-tuned oscillations in the modes' temporal signatures; (iv) the oscillatory modes are sensitive to anesthesia varying both in number, frequency, stability and spatial configuration; (v) the oscillatory modes detected exhibit features of a dynamical system operating in the subcritical range of a Hopf bifurcation; (vi) the stochastic resonance of stationary patterns generates patterns of long-range functional connectivity similar to the ones detected empirically; (vii) these findings support the emerging hypothesis that resting-state activity detected with fMRI results from the superposition of standing waves resonating transiently within the brain's anatomical structure, which in turn drive fluctuations in sliding-window correlations between the brain subsystems located at the wave antinodes (Fig. 9).

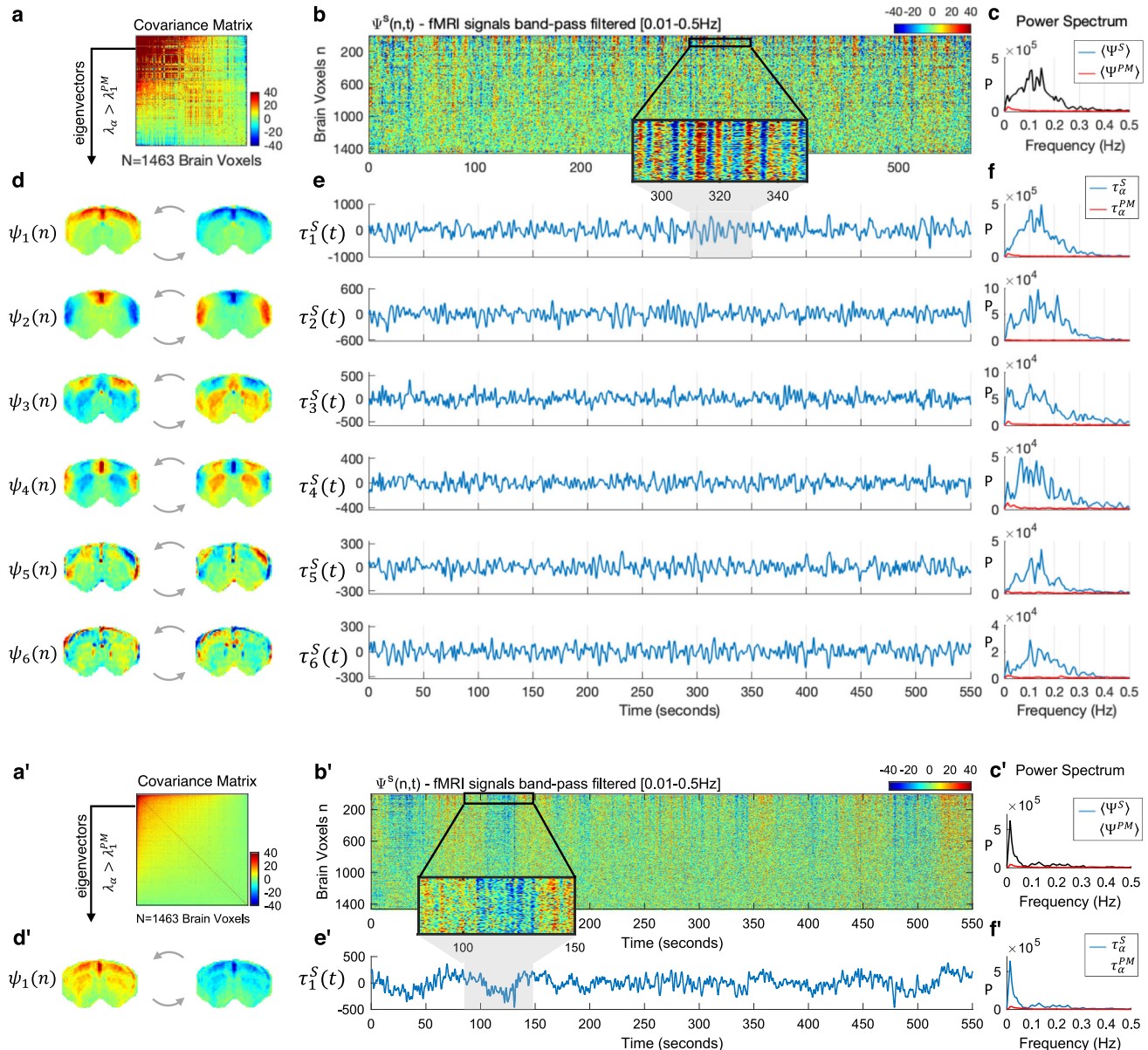

**Fig. 6 | The addition of isoflurane at 1% and 3% concentrations alters the spatial, temporal, and spectral signatures of principal components. a, a'** The $N \times N$ covariance matrix of fMRI signals band-pass filtered between 0.01 and 0.3 Hz, averaged across 12 scans after the addition of isoflurane at 1% (top) and 3% (bottom) concentrations. **b, b'** Carpet plot of the fMRI signals recorded in all brain voxels, $n$, over time, $t$, represented by the wave function $\Psi^S(n,t)$, here shown for two scans $S$ of the same rat from Fig. 3 in the frequency range [0.01–0.5 Hz]. Voxels are sorted according to the elements in the largest magnitude eigenvector $\psi_1$. Values correspond to fMRI signal change with respect to the mean in each voxel. A zoom into the first 100 voxels over 60 s is inserted to illustrate oscillations in the signals. **c, c'** Power spectrum of the mean fMRI signal across voxels. **d, d'** The principal components detected with eigenvalue above baseline, are scaled by 1 (left) and −1 (right) to illustrate the activity pattern when the temporal signature oscillates between positive and negative values. **e, e'** Temporal signature associated with each of the supra-threshold principal components given by $\tau_\alpha^S(t) = \psi_\alpha(n)\Psi^S(n,t)$ for the same scan shown in (**b**). **f, f'** Power spectra of the temporal signatures from (**e**).

## Discussion

Rhythms at frequencies ranging from 0.5 up to >100 Hz have been shown to emerge from intrinsic neural processes[62–64]. However, the role and generative mechanisms of rhythms below 0.5 Hz detected both with fMRI, EEG, and electrophysiology remain under vigorous debate[31,47,53,65,66]. Using fMRI experiments with hitherto unprecedented spatiotemporal resolution we provide insights into this problem by demonstrating the existence of intrinsic macroscale oscillatory modes in fMRI signals, which organize with mode-specific phase relationships across extended areas across the cortex and subcortex, driving correlated activity between distant regions.

The oscillatory modes detected were found to be consistent across rats within the same anesthetic condition, but to vary in terms of spatial configuration, peak frequency, and damping coefficient across conditions. Despite these differences, the modes detected across conditions are qualitatively similar in terms of the organization through phase gradients within anatomically defined cortical and subcortical boundaries, indicating they likely share a common generative principle. Thus, the metrics that become accessible when characterizing these intrinsic modes may be beneficial compared with the more conventional dynamic functional connectivity metrics, as they provide quantitative parameters on the nature of the correlation. It is further worth noting that the snapshots of our intrinsic mode oscillations resemble the spatial patterns exposed by quasi-periodic patterns (QPPs)[17,18] and co-activation patterns[67], which were found to improve the characterization of early Alzheimer's disease stages[68,69]

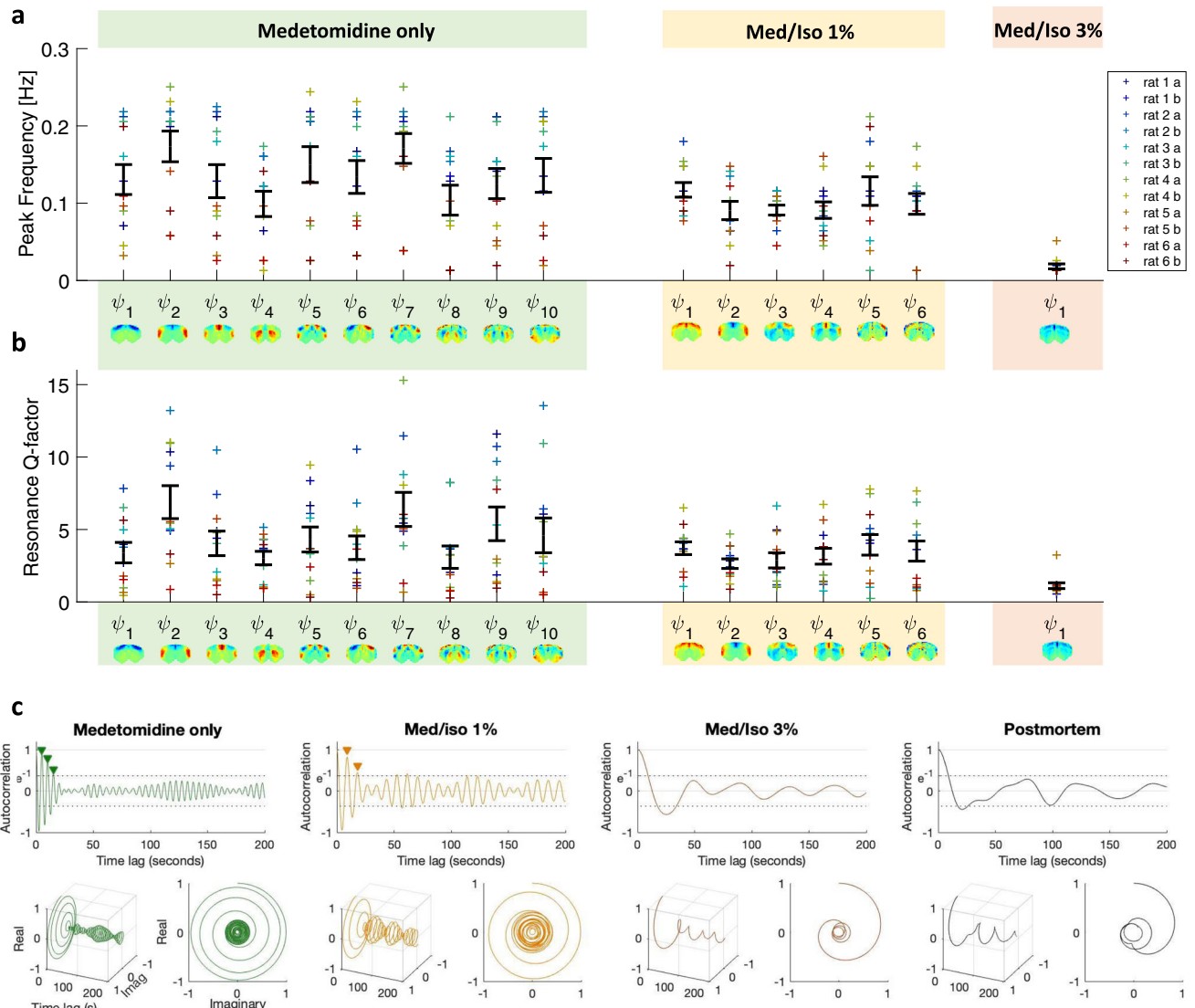

**Fig. 7 | Principal components oscillate at higher frequencies and with less damping under medetomidine.** The temporal signatures associated with the principal components detected in each condition are characterized in terms of peak frequency (**a**) and $Q$-factor (**b**) for each of the $s = 12$ scans in each condition (2 scans per rat). Error bars represent the mean ± standard error across scans. **c** To illustrate the stability of the oscillations, the autocorrelation functions of the temporal signatures $\tau_1^S$ associated with the first principal component in each condition are reported. Examples are shown for 3 scans from the same rat and from a postmortem scan. As can be seen, the autocorrelation function under medetomidine exhibits 3 oscillations before the amplitude decays to 1/e (-37%), 2 cycles after adding isoflurane at 1% and no complete cycle under deep anesthesia, similar to what is observed in the postmortem scan.

compared with more conventional resting-state fMRI functional connectivity metrics. Extending such characterizations using the Intrinsic macroscopic oscillatory mode framework may shed further light into disease, as well as how activating or silencing specific areas contributes to whole-brain modulations[70,71].

Our results also align with neural field theories for macroscale brain dynamics describing large-scale wave propagation of neuronal activity including a spatial Laplacian to incorporate the brain geometry of the brain[72–75]. While these neural field theories have historically been used to describe macroscale brain activity detected with EEG, recent studies suggest that the structural eigenmodes (defined either from brain surface geometry or from diffusion networks) may also be at the origin of macroscopic activity patterns detected with fMRI, namely the so-called resting-state networks or intrinsic connectivity networks[35–39]. This has been recently reinforced by a study demonstrating high spatial similarity between the covariance eigenvectors of fMRI signals and the theoretical prediction of Helmholtz eigenmodes of the Laplace–Beltrami operator starting from a brain surface mesh[66].

Overall, these studies support our interpretation that the principal components detected empirically from the fMRI signals are eigenmodes intrinsic to the brain structure, including not only the cortex but also subcortical structures, such as the striatum. This hypothesis could not, however, be fully validated in the current work, given that the eigenmodes of covariance were obtained from the fMRI signals alone and not compared with modes predicted from the structure. Although these spatially defined modes were found to be consistent across animals and to amplify the signal with respect to spatially reordered vectors (i.e., either by randomization or by sorting the phases in a gradient from left to right), our validation tests may not be sufficient to demonstrate that this is the adequate basis of spatial patterns to describe functional neuroimaging data and further validations are needed[76]. Even so, a closer inspection of the spatial configuration of the modes shown in Fig. 3 reveals that distinct eigenmodes emerge not only in the cortex but also in the striatum, with some possibly representing fundamental (i.e., global) modes of specific brain structures (i.e., $\psi_1$ for the cortex and $\psi_4$ for the striatum),

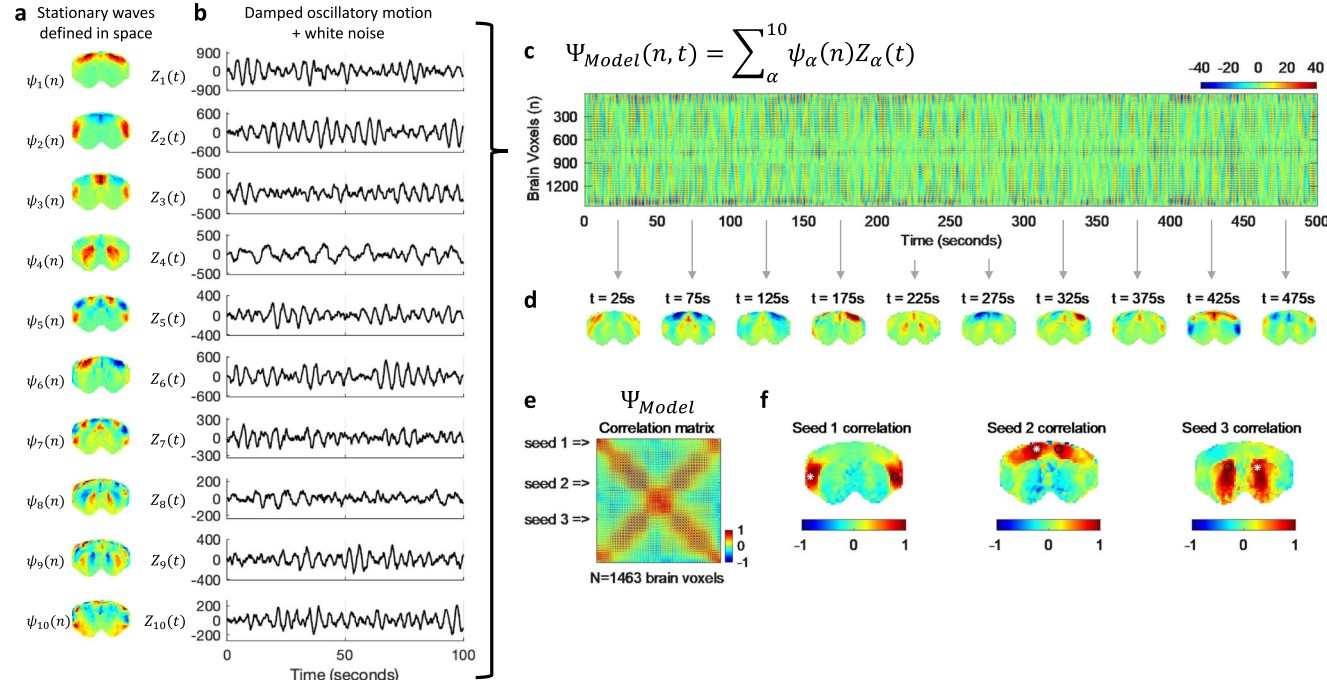

**Fig. 8 | Stochastic resonance of standing waves drives transient long-range correlations in simulated signals.** The spatial configurations and temporal signatures of the principal components align with the hypothesis that they represent standing waves, whose phenomenology is inherently associated with resonance phenomena. To model the dynamics emerging from the transient resonance of standing waves in the presence of background noise, for each of the spatial patterns detected in medetomidine-sedated rats (**a**), we simulate a temporal signature as the behavior of an underdamped oscillator perturbed with gaussian white noise (**b**). **c** Multiplying each $N \times 1$ spatial pattern by the corresponding $1 \times T$ temporal signature results in an $N \times T$ spatiotemporal pattern for each mode. The linear sum of

these spatiotemporal patterns represents the superposition of a repertoire of standing waves resonating transiently in the presence of noise. **d** Still frames of the simulated dynamics obtained at distinct time points reveal the multiplicity of patterns that can be generated over time. **e** To demonstrate that this scenario generates patterns of long-range functional connectivity, we compute the correlation matrix of the simulated signals, as performed initially on empirical fMRI data (Fig. 1). **f** The seed correlation maps obtained from the simulated signals reveal correlation patterns visually similar to the ones detected from real fMRI recordings. Source Code is provided with this work.

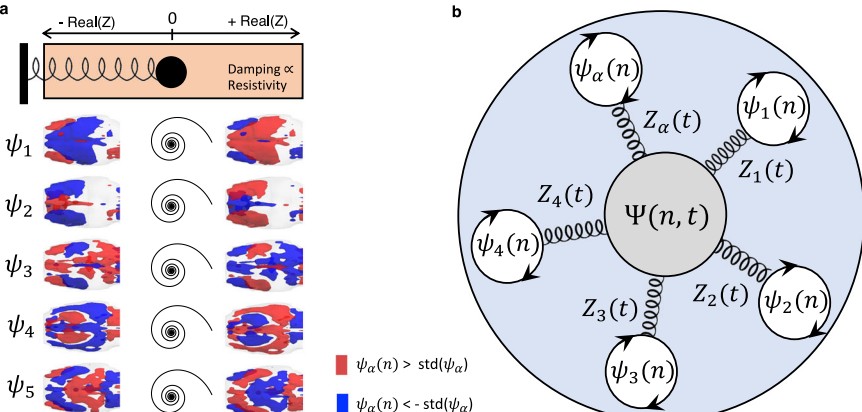

**Fig. 9 | Mechanistic model for the spontaneous resonance of standing waves driving the activation of functional brain networks. a** Like the response of spring, the temporal signature of brain modes can be approximated by a damped oscillator. Despite the lower temporal resolution inherent to multi-slice acquisitions hindering the detection of resonant behavior, the consistency of spatial patterns reinforces the hypothesis that the damped oscillatory response of

functional networks extends to the whole-brain level, here represented by the first 5 eigenvectors of the average covariance matrices across 6 whole-brain scans (from 3 different rats). **b** Diagram illustrating a mechanistic scenario for brain activity, where each functional network is represented by a spatial pattern $\psi_\alpha$ responding to perturbation with a damped harmonic motion.

while others may represent subsequent harmonic modes with increasing spatial frequencies (i.e., $\psi_{6,2,3,5,7}$ for the cortex and $\psi_8$ for the striatum).

While under medetomidine alone, strong oscillations were detected up to 0.3 Hz in agreement with previous literature[44], the addition of, isoflurane at 1% concentration was found to particularly

suppress the power between 0.10 and 0.25 Hz, leaving the power in the typical range considered in resting-state studies (i.e. <0.1 Hz) mostly unchanged. Further increasing isoflurane concentration to 3%, most oscillatory power is lost and only very slow (<0.05 Hz) global and aperiodic fluctuations are detected. It remains unclear whether these non-linear effects are related to the differential effects of

medetomidine and isoflurane on blood vessels or can be explained by more direct changes in the resistivity of the medium through which the waves propagate, altering its resonant quality.

The differences detected across anesthetic conditions question the theoretical predictions of modes depending on the brain's geometric structure alone because it is implicit that the anatomy of the brain is invariant across conditions. Indeed, it is generally known that the resonant modes of a system depend not only on the structural geometry of the system but also on the resistivity of the propagating medium, which directly affects not only the spatial patterns but also the resonant frequency and the stability of the oscillations. Given that anesthetics directly affect diverse properties of the brain tissue and vasculature, our results raise the importance of considering not only the brain geometry but also the resistivity of the medium through which the waves propagate to possibly explain the differences in resonant quality observed across anesthetic conditions.

Though many models can be used to describe damped oscillatory motion, we find the Stuart–Landau equation in the subcritical range of a Hopf bifurcation to be appropriate since it is the canonical form to describe a system in the vicinity of a limit cycle, responding to perturbation with oscillations with decaying amplitude from basic mathematical principles, with the imaginary component ensuring the conservation of angular momentum[77]. Hopf bifurcation models have been used in models of spontaneous brain activity to represent local field oscillations interacting through the connectome structure[78–80]. Here instead, we demonstrate that the oscillations detected with fMRI are not purely local since they lock-in phase across distinct subsystems and therefore each oscillator model is associated with a distributed spatial map of phase relationships, analogous to the modes of vibration in a violin string or a drum membrane[66,81,82]. We show that the temporal signature of the wave patterns can be approximated by an oscillator model with varying natural frequency and damping coefficients. In such a framework, the less damped the system is, the more it resonates at its natural frequency in the presence of naturally occurring noise. We hypothesize that this is what is occurring under medetomidine. If the damping is increased (as observed with the addition of isoflurane), then the oscillations are sustained over fewer cycles and at slower frequencies.

Being inherently associated with resonance, standing waves are a fundamental property of matter, resulting from the constructive interference of waves, driving correlated (and anti-correlated) oscillations in the wave's anti-nodes. The general principle of wave superposition implies that the brain can engage simultaneously in multiple functional networks, instead of switching from one functional network to another, as often considered in the analysis of dynamic functional connectivity[14,83–85]. In other words, our results substantiate that the activity recorded herein with fMRI aligns with neural field theories[66,73,86,87], where at any given moment, the wave function $\Psi$— representing the collective systemic activity—results from the superposition of a discrete set of wave functions with a damped oscillatory response. This resonance framework offers simultaneously an explanation for (i) the spontaneous emergence of ultra-slow oscillations in brain activity, (ii) the profile of phase relationships across space (as observed in gradient-like functional connectivity patterns), and (iii) the difference in damping coefficients across conditions. We note, however, that the temporal coordination between the different modes was not addressed in the current work and requires further investigation.

The generalization of these findings to other animal species including humans can only be discussed in the light of existing literature and needs further experimental validation. On one hand, the similarity of the principal components detected herein with intrinsic network patterns detected using different methodologies suggests these are expressions of the same emergent phenomena, typically referred to as resting-state networks (RSNs) or intrinsic connectivity networks (ICNs). Since rats, mice, monkeys, and humans exhibit qualitatively similar RSNs/ICNs, it can be expected that they are expressions of the same phenomenon. Indeed, a wide range of studies have demonstrated that intrinsic macroscale modes of brain activity (detected across modalities) exhibit spatial features of standing waves, so it can be expected that their temporal signature exhibits a damped harmonic motion. Even if no clear periodicity is detected in resting state fMRI in humans and the fluctuations closely approximate the canonical hemodynamic response function, one cannot exclude the possibility that the fluctuations reflect an overdamped oscillatory response associated with the transient and short-lived resonance of a stationary wave, providing an additional generative hypothesis for the dynamic patterns observed empirically.

A question that typically arises in this context is how closely the fMRI signals track the underlying neural activity, mainly due to the involvement of neurovascular coupling mechanisms. Although this study did not attempt to deconfound neural activity from vascular coupling, it is interesting to note that the mode temporal signatures did not follow the canonical hemodynamic response function. Instead, a ubiquitous transiently sustained periodicity occurs within a range of frequencies extending significantly above the range typically associated with the BOLD signal, but below cardiac and respiratory physiological rhythms. Recent studies combining simultaneous electrophysiological recordings of local field potentials (LFPs) and fMRI in rats reported significant coherence between the two signals precisely in the range of frequencies detected herein[43,44]. Therefore, one cannot exclude the possibility that the oscillations observed with fMRI are linked with other factors beyond blood flow/volume effects alone, and may provide a more direct measurement of neuronal activity[30,88]. Still, given that hemodynamic blurring is expected[32] further local spectral properties may have been obscured by this blurring.

The advantages of exploring fMRI signals at faster resolutions extend well beyond this work and certainly deserve further exploration. While previous ultrafast fMRI experiments in rodents have reached up to 20 frames per second, they have focused mostly on stimulus-driven responses in specific regions of interest[89–93], and much remains to be explored at the level of spontaneous long-range interactions. Allowing for a large span of scales both in space (from micrometer to meter) and in time (from millisecond to hour), exploring the full possibilities of MRI may provide relevant insights into the brain organizational principles in the spatial, temporal, and spectral domains[88]. Indeed, the traditionally low temporal resolution of fMRI studies has limited the analysis of spatial correlations between ultra-slow fluctuations in distant areas. More recent dynamic analysis of functional connectivity has revealed the non-stationary nature of network interactions[20,94,95]. Still, under the Connectomics framework, even dynamic studies measure spatial connectivity patterns over time, rather than investigating deeper origins of the signals. These additional insights may help resolve the disparate—yet not mutually exclusive—hypotheses that have been put forth on the nature of functional connectivity, ranging from stochastic resonance[96], metastable synchronization[80,97,98], superposition of harmonic modes[36,37,66] that can constrain brain function[99] or transitions between phase-locking patterns[29], among others.

In conclusion, this work reveals evidence for macroscopic oscillatory modes in spontaneous fMRI signals that organize across the brain in standing wave patterns, providing fresh insight into the organizing principles giving rise to intrinsic connectivity networks. Future work disentangling the different underlying sources of the fMRI signal, as well as studying the impact of specific therapeutic effects, such as direct electromagnetic stimulation or pharmacological manipulations, should deepen our understanding of intrinsic oscillatory modes in the brain. Ultimately, by promoting a better understanding of brain dynamics, this work provides perspective avenues for the advance in the diagnosis and treatment of brain disorders.

## Methods

### Ethical statement

All animal experiments complied with the European Directive 2010/63 (established by Portuguese law Decreto-Lei 113/2013) and followed the Federation of European Laboratory Animal Science Associations (FELASA) guidelines and recommendations concerning laboratory animal welfare. Experiments were preapproved by the Champalimaud Foundation's Internal Review Board (ORBEA) and by the Portuguese competent authority for animal welfare (DGAV, Direcção Geral de Alimentação e Veterinária) with license number 0421/000/000/2016.

### Experimental design

Ultrafast resting-state fMRI recordings were obtained from a single brain slice of 6 rats scanned under 3 different conditions, namely medetomidine[51] combined with 3 concentrations of isoflurane: 0% (sedation), 1% (light anesthesia), and 3% (deep anesthesia). To minimize variability across animals, we chose all animals of the same sex. Two additional scans were recorded from a seventh rat postmortem to serve as a baseline. Moreover, resting-state fMRI recordings covering 12 slices of the rat brain were acquired from 3 rats under medetomidine. Below, we elaborate on each phase.

### Animal preparation

Long-Evans female rats (($N = 6$ alive + $N = 1$ postmortem)) weighing $216 \pm 30$ g and aged $8.7 \pm 1.6$ weeks were used in this study. To minimize variability across animals, we chose all animals of the same sex. Animals were reared in a temperature-controlled room and held under a 12 h/12 h light/dark cycle with ad libitum access to food and water.

Anesthesia was induced with 5% isoflurane (Vetflurane™, Virbac, France) mixed with oxygen-enriched (27–30%) air in a custom-built plastic box. Rats were then weighed, moved to the animal bed (Bruker, Germany) and isoflurane was reduced to 2.5%. Eye ointment (Bepanthen Eye Drops, Bepanthen, Germany) was applied to prevent eye dryness.

In the six living rats, a 0.05 mg/kg bolus of medetomidine solution (Dormilan, Vetpharma Animal Health, Spain: 1 mg/ml, diluted 1:10 in saline) was injected subcutaneously 5–8 min after induction, immediately followed by a constant infusion of 0.1 mg/kg/h s.c. of the same solution (25), delivered via a syringe pump (GenieTouch™, Kent Scientific, USA) until the end of the experiment, and by a 10 min-long period where isoflurane was gradually decreased to 0%. fMRI acquisitions began once the animals stabilized in this condition (the time after isoflurane is reported in Table S1 for each scan). For each rat, 2 fMRI scans were first acquired under medetomidine only (sedation condition). Subsequently, fMRI scans were acquired after increasing isoflurane concentration to 1% (light anesthesia condition) and finally to 3% (deep anesthesia condition), waiting 10 min after each isoflurane increase for anesthesia stabilization.

The breathing frequency and rectal temperature were monitored throughout the MRI sessions using a pillow sensor and a 9 ft optic fiber probe with a 1 mm tip (Model 1030 Monitoring & Gating System and Fiber Optic Temperature Module & Sensor, SA Instruments Inc., Stony Brook, USA), respectively, with PC-SAM 8.02 software. At the end of the experiments, medetomidine sedation was reverted by injecting 0.25 mg/kg s.c. of atipamezole (Antisedan, Vetpharma Animal Health, Spain: 5 mg/ml, diluted 1:10 in saline).

The seventh rat, reared in the same conditions, was injected with 1 mL (60 mg) pentobarbital i.p. and scanned postmortem with the same MRI protocol to serve as a control.

### MRI protocol

MRI data were acquired using the software ParaVision 6.0.1. Animals were imaged using a 9.4 T BioSpec® MRI scanner (Bruker, Germany) equipped with an AVANCE™ III HD console, producing isotropic pulsed field gradients of up to 660 mT/m with a 120 μs rise time. RF transmission was achieved using an 86 mm-ID quadrature resonator, while a 4-element array cryoprobe (Bruker, Fallanden, Switzerland) was used for signal reception. Following localizer experiments and routine adjustments for center frequency, RF calibration, acquisition of $B_0$ maps, and automatic shimming, anatomical images were acquired using a $T_2$-weighted RARE sequence in the coronal plane: TR/TE = 2000/36 ms, FOV = $18 \times 16.1$ mm$^2$, in-plane resolution = $150 \times 150$ μm$^2$, RARE factor = 8, slice thickness = 0.6 mm, 22 slices, $t_{acq} = 3$ min 28 s, and sagittal plane: TR/TE = 2000/36 ms, FOV = $24 \times 16.1$ mm$^2$, in-plane resolution = $150 \times 150$ μm$^2$, RARE factor = 8, slice thickness = 0.5 mm, 20 slices, $t_{acq} = 3$ min 28 s. These images were used to place the slices of interest.

### Single-slice ultrafast fMRI acquisitions

To minimize the repetition time, we focused our analysis in a single 1.2 mm-thick slice of the rat brain, choosing a frontal slice that covered a large cortical area and with a FOV of $21 \times 21$ mm$^2$, as shown in Fig. S1 A, B. The slice was placed between −0.2 and 1.0 mm from Bregma according to the Paxinos & Watson rat brain atlas[100] (Supplementary Fig. S1c).

Six resting-state scans (2 per condition) were acquired from each of $N = 6$ living rats (totaling 36 scans) using a gradient-echo echo planar imaging (GE-EPI) sequence (TR/TE = 38/11 ms, flip angle = 15°, matrix size = $84 \times 84$, in-plane resolution = $250 \times 250$ μm$^2$, number of time frames = 16,000, $t_{acq} = 10$ min 8 s). Two postmortem scans were acquired with the same parameters. Additionally, a Multi-Gradient Echo sequence (MGE, TE = 2.5:5:97.5 ms, TR = 300 ms, flip angle = 40°, matrix size = $210 \times 210$, in-plane resolution = $100 \times 100$ μm$^2$, $t_{acq} = 4$ min 12 s) and a Time-Of-Flight (TOF) FLASH sequence (TR/TE = 8.2/3.3 ms, flip angle = 80°, matrix size = $210 \times 210$, in-plane resolution = $100 \times 100$ μm$^2$, $t_{acq} = 17$ s 219 ms) sequence were acquired from all rats to obtain additional anatomical and vascular information about the slice, respectively. Details of time after isoflurane induction, breathing frequency and rectal temperature are reported for each scan in Table S1.

### Whole-brain fMRI acquisitions

Resting-state data was acquired twice under medetomidine sedation from a subsample of $N = 3$ of the living rats using a multi-slice GE-EPI sequence covering the entire rat brain, from the frontal part of the cerebellum to the posterior part of the olfactory bulb, and with the following parameters: TR/TE = 350/11 ms, flip angle = 40°, FOV = $24 \times 24$ mm$^2$, matrix size = $70 \times 70$, in-plane resolution = $342.9 \times 342.9$ μm$^2$, slice thickness = 1.2 mm, slice gap = 0.15 mm, 12 slices, number of time frames = 2572, $t_{acq} = 15$ min 0 s 200 ms.

### Brain masks

Individual brain masks were defined manually and aligned across rats to a common central coordinate. All individual rat masks were superposed to define a common brain mask containing $N = 1463$ voxels in the single slice and $N = 7426$ voxels in the whole brain.

In the single slice, no spatial or temporal interpolation was applied to the signals, such that the signal in each brain voxel corresponds to the raw fMRI signal recorded.

In multi-slice acquisitions, the slice-timing correction was applied using linear interpolation over time.

### Space-frequency analysis of fMRI data

Power spectra were computed for the fMRI signals on each of the $84 \times 84 = 7056$ voxels using the fast Fourier transform, after removing the first 1000 frames (38 s) and detrending. Voxel power spectra were obtained up to the Nyquist frequency of $(2TR)^{-1} = 13.1579$ Hz. Images of the power across a selected range of frequencies were obtained by averaging the band-limited power in each voxel across scans in the same condition. All spectral analyses were performed at the single scan

level and metrics statistically compared between conditions. Analysis up to the Nyquist frequency is reported in Supplementary Figs. S5 and S11.

## Principal component analysis

For each scan, the fMRI signals in $N = 1463$ brain voxels were band-pass filtered between 0.01 and 0.3 Hz and the $N \times N$ covariance matrix was computed. The largest magnitude eigenvalue, $\lambda_1^{PM}$ was calculated for the two postmortem scans, and the largest one selected as the baseline threshold. The covariance matrices were averaged across the 12 scans in each condition, and, for each condition the $\alpha$ $N \times 1$ eigenvectors with eigenvalue $\lambda_\alpha^{condition} > \lambda_1^{PM}$ were extracted, representing the principal components detected in each condition with magnitude above the postmortem baseline.

The same analysis performed using decreasing sampling rates is reported in Supplementary Fig. S10. The modes detected using correlation—instead of covariance—above the postmortem baseline are reported in Supplementary Fig. S11.

## Statistical analysis

We compared the power between conditions at different frequency bands using a non-parametric permutation-based $t$-test (10,000 permutations to ensure the robustness of results) to detect the frequency range most sensitive to the three different conditions. $p$-values were conservatively corrected by the number of comparisons performed (Bonferroni correction), considering both the number of between-group comparisons (considering only independent hypotheses) as well as the number of frequency bands considered (considering dependent hypotheses as well, which is even more conservative).

The resonance $Q$-factors and peak frequencies were statistically compared using the same permutation test.

The standard error is calculated as the standard deviation divided by the square root of the number of values compared.

## Resonance analysis

Resonance was evaluated by computing the $Q$-factor, a measure typically used in acoustics and engineering to quantify resonance phenomena. Importantly, it is not implied by definition that a covariance mode will oscillate, since signals can co-vary aperiodically, without necessarily oscillating. The $Q$-factors were estimated for the temporal signatures associated with the principal component detected in each condition, for all the 12 scans in each condition, and statistically compared between conditions ($p$-values reported in Table S2).

## Damped oscillator model

To illustrate the response of an oscillatory system with different damping coefficients, we used the Stuart–Landau equation:

$$\frac{dZ}{dt} = Z\left(i\omega - |Z^2| + a\right)$$

where $Z$ is complex (with real and imaginary components), $\omega$ is the natural frequency, and $a$ defines the position of the system with respect to the Hopf bifurcation at $a = 0$, such that for $a > 0$ the system displays self-sustained oscillations with constant amplitude scaled by $a$, whereas for $a < 0$ the oscillations are damped and the system decays back to the fixed point equilibrium at $Z = 0$ at a rate scaled by the magnitude of $a$ (i.e., the more negative the $a$, the stronger the damping).

A single unit pulse (i.e., a Dirac delta function) is applied at $t = 0$ to illustrate the intrinsic response of the system in Supplementary Fig. S16. Further, to illustrate the response to continuous perturbation with a stochastic input, we add complex Gaussian white noise as:

$$\frac{dZ}{dt} = Z\left(i\omega - |Z^2| + a\right) + \beta\eta_1 + i\beta\eta_2,$$

where $\eta_1$ and $\eta_2$ are independently drawn from a Gaussian distribution with standard deviation $\beta = 1$ (integrated as $\beta\sqrt{dt}$). Simulations were obtained using the Euler method for numerical integration with a time step $dt = 10^{-3}$ s.

## Reporting summary

Further information on research design is available in the Nature Portfolio Reporting Summary linked to this article.

## Data availability

The raw structural and dynamic MR images used in this study are available for download as Maltab files (.mat) without restrictions from: https://drive.google.com/drive/u/5/folders/1JQ_1AmP4v-HEB_ll5ZwHEaBORtIoA17R. An example dataset with a reduced subset of fMRI scans sufficient to replicate the analysis is also included. Source Data for Fig. 2 and Supplementary Figs. 3 and 4 are provided as Excel files (.xlsx) with this paper. Source data are provided with this paper.

## Code availability

Source Code for the replication of the analysis and figure generation are made available as Matlab scripts with this work.

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

## Acknowledgements

Santa Casa da Misericórdia. Grant name and ID: Mantero Belard, MB-56-2020 (N.S.). La Caixa Foundation (Spain), grant LCF/BQ/PR22/11920014

(J.C.). European Research Council (ERC) grant 679058 (N.S., F.F.F.). Lisboa Regional Operational Program (Lisboa 2020), under the PORTU-GAL 2020 Partnership Agreement through the European Regional Development Fund (ERDF).Portuguese Foundation for Science and Technology (FCT) grant LISBOA-01-0145-FEDER-022170. Portuguese Foundation for Science and Technology (FCT) grant 275-FCT-PTDC/BBB/IMG/5132/2014 (N.S., F.F.F.). Portuguese Foundation for Science and Technology (FCT) grant UIDB/50026/2020 (J.C.). Portuguese Foundation for Science and Technology (FCT) grant UIDP/50026/2020 (J.C.). Portuguese Foundation for Science and Technology (FCT) grant CEE-CIND/03325/2017 (J.C.).

## Author contributions

J.C. and N.S. conceptualized the work. N.S. designed the MRI acquisition sequences and supervised the work. F.F.F. performed animal manipulation and data acquisition. J.C. performed the analysis and visualization and is responsible for the source code. J.C. and N.S. wrote the original draft of the manuscript. All authors contributed to reviewing and editing of the manuscript.

## Competing interests

The authors declare no competing interests.
