## [Peer Review File · Nature Communications]

Intrinsic macroscale oscillatory modes driving long range functional connectivity in rat brains detected by ultrafast fMRIReviewer #1 (Remarks to the Author):

In this manuscript, the authors developed an ultra-fast fMRI paradigm to track high-frequency brain functional oscillations, attempting to probe the fundamental mechanism underlying brain functional connectivity. They showed that obtained fMRI time course could be well fitted by a wave model; and hence concluded that observed spontaneous functional activity was driven by these resonant modes. While the resonant model is novel and connects with existing computational theories of brain activity, I have several questions/comments that hope the authors could address/clarify before publication.

1 One major technical novelty of this manuscript is the employment of ultra-fast temporal sampling (38 ms) through reduced slice coverage, however, it is unclear why such a high sampling rate is necessary given that the highest frequency of interest explored here is only 0.4 Hz. According to supplementary Fig. S13, a TR faster than 2 s already suffices to resolve the 0.2/0.25 Hz resonant peak. While faster sampling may improve the detection of frequency-specific differences between conditions (Fig. S12), would removal of aliased high-frequency physiological sources in the low-TR condition, e.g., through model-based approaches, achieve comparable efficacy? I'm raising this concern because faster sampling will incur a penalty in the spatial resolution (or spatial coverage here) and signal-to-noise ratios (due to reduced T1 recovery time, i.e., marked increase of white noise).

2 Prior to this work, several human/animal studies have already evaluated the temporal characteristics of high-frequency spontaneous functional connectivity (even > 0.5 Hz); this line of literature is missing in the current manuscript. To list a few:

[1] Lee, H.L., Zahneisen, B., Hugger, T., LeVan, P. and Hennig, J., 2013. Tracking dynamic resting-state networks at higher frequencies using MR-encephalography. *Neuroimage*, 65, pp.216-222.

[2] Chen, J.E. and Glover, G.H., 2015. BOLD fractional contribution to resting-state functional connectivity above 0.1 Hz. *Neuroimage*, 107, pp.207-218.

[3] Lin, F.H., Chu, Y.H., Hsu, Y.C., Lin, J.F.L., Tsai, K.W.K., Tsai, S.Y. and Kuo, W.J., 2015. Significant feed-forward connectivity revealed by high frequency components of BOLD fMRI signals. *Neuroimage*, 121, pp.69-77.

[4] Trapp, C., Vakamudi, K. and Posse, S., 2018. On the detection of high frequency correlations in resting state fMRI. *NeuroImage*, 164, pp.202-213.

3 It is unclear how the signal models evaluated for the current dataset (sedated rat) can extend to awake human conditions, due to several reasons. First, extensive human fMRI studies have failed to see the resonant peaks at 0.2/0.25 Hz as reported here, even with sampling rates faster than the Nyquist limits. Second, there has been emerging evidence based on rodent models supporting that brain connectivity may arise from oscillatory vasomotor activity, which however occurs at 0.1 Hz, instead of 0.2 Hz, during awake conditions (Mateo, C., Knutsen, P.M., Tsai, P.S., Shih, A.Y. and Kleinfeld, D., 2017. Entrainment of arteriole vasomotor fluctuations by neural activity is a basis of blood-oxygenation-level-dependent "resting-state" connectivity. *Neuron*, 96(4), pp.936-948.). Third, many studies have already shown that anesthesia could have a strong impact on the measured BOLD signal properties and associated brain connectivity (Paasonen, J., Stenroos, P., Salo, R.A., Kiviniemi, V. and Gröhn, O., 2018. Functional connectivity under six anesthesia protocols and the awake condition in rat brain. *Neuroimage*, 172, pp.9-20.).

4 Finally, the major conclusion from this work—functional network patterns emerge from a repertoire of standing waves resonating at frequencies up to 0.25 Hz—is not well justified. While the resonant models have already been adopted to depict neuronal oscillations, they may not extend to fMRI signals, because the spectral property of BOLD signals is blurred by region-specific hemodynamic changes. Apart from the distinct signal properties of "wake" vs "sedation" state as discussed above, even if the resonant model can well fit the fMRI data, we cannot conclude that the resonance phenomenon is the fundamental mechanism underlying spontaneous functional behavior. The authors should, at least, either demonstrate that alternative models (e.g., scale-free models) cannot fit the data, or perform simulations to validate whether the characterized covariance modes are reliable. For instance, the author could generate a batch of sham datasets (with coherent, scale-free fluctuations for different brain networks), and go through the identical

modeling to test whether the analytical outputs of the sham data have distinct spatial signatures from the modes reported here, e.g., not with "anti-nodes standing-wave" features.

Reviewer #2 (Remarks to the Author):

SUMMARY: Cabral et al. present an analysis of the low-dimensional components of rat fMRI, at 'ultrafast' temporal resolution in a slice and at standard temporal resolution in the whole brain. Applying a linear dimensionality reduction method based on covariance modes (resembling principal components analysis), they find spatial modes that resemble standing-wave patterns with time courses that have oscillatory components, consistent with prior work applying dimensionality reduction to fMRI time series in rodent and human. Unfortunately, relevant context is missing, the claims are over-stated given the evidence, the phenomenological model is not quantitatively connected to the experimental data and does not provide insights, and the methods are often over-interpreted.

MAJOR: Novelty, and appropriateness of methods for extracting 'resonant modes':

Summary of literature on modes and waves is incomplete in introduction. There is a lot of work that has been done on this that is not sufficiently explained/cited. It's therefore unclear what the scope of this study is given the amount of prior work.

E.g., lots of work has done similarly as here (something similar to principal components analysis of fMRI data) in many species and resolutions, and found similar results. Others have found low-dimensional gradients in rodent (mouse) fMRI (cf. Huntenburg, also in human). These highly explanatory spatial maps (in human, mouse, ...) always look like standing-wave patterns, as reported here. Of the many papers reporting low-dimensional modes, I can't think of any that didn't exhibit "clear standing wave properties" with the characteristics listed in this paper (i.e., having nodes and antinodes).

Others have done detailed and specific explorations of wave propagation (e.g., in mouse, cf. Liang), or used techniques explicitly designed to extract standing-wave patterns with oscillatory time courses from data, like dynamic mode decomposition (DMD, cf. Casorso and others), or derived spatial patterns consistent with eigenmodes of brain geometry (cf. Robinson).

Here, the authors should more completely explain the relevance of this prior work and the open questions addressed—many results are described as surprising or 'striking' that are consistent with similar prior studies.

While the authors use a linear dimensionality reduction method (PCA-like: an eigenvalue decomposition of the covariance matrix), it is interpreted as being "based on wave physics" (when it is not) and therefore explicitly extracts standing-wave patterns (whereas the temporal profiles are unrestricted with covariance modes). Applying a band-pass filter to a very right range of frequencies around the peak of the power spectrum accentuates the oscillatory appearance of these resulting time courses.

More could be done to set up what exactly is the expectation from prior literature in order to make the claim of what here is surprising/interesting and therefore why it might have an impact on the field.

MINOR:

- Opening results are unclear as to why the analyses are done (motivation is missing). Is the aim to understand whether different spatiotemporal patterns can be extracted at this high sampling rate relative to a standard sampling rate?
- "isoflurane nonlinearly modulates the power spectrum" -> 'non-uniformly'?
- It's not clear why it's "remarkable" that covariance modes are found. This is consistent with prior work. It would be remarkable if they were not found.
- It's not clear why it's "even more remarkable" that different frequency bands are associated with different covariance modes. It would be remarkable if the opposite was true: i.e., that modes were identical across all frequency ranges.
- "detached mode" undefined.

- Main text is too brief—the reader has to do a lot of work to understand why analyses are being done, and also to work out what was done—motivation is often missing and so are the methodological details required to understand what is being presented. E.g., the number of modes detected the reader needs to try to find what this means (either finding clues in annotations on the figure or reading supplement).
- P7: the "analytical framework based on wave physics" doesn't appear to have any grounding in wave physics, but closely resembles a principal components analysis decomposition.
- Unclear why data are bandpass filtered up to 0.4 Hz when the main novelty of this paper is billed as the dynamics on 'ultrafast' fMRI timescales. The higher frequency bands also don't seem relevant in Fig. 1. There is a disconnect between the narrative about the novelty of ultrafast fMRI versus the actual results presented, showing interesting patterns at slower frequencies (accessible to conventional fMRI).
- Eq. (1) is not a "mechanistic framework of wave superposition", it corresponds to keeping the first 10 PCs. A mechanistic framework is approximately there with DMD, and more closely there with geometric eigenmodes, but applying PCA to a data matrix doesn't have any clear grounding in physical/wave mechanisms. Also overstated: "this approach provides insight into the mechanisms driving spontaneous brain network dynamics." This claim is made but no examples of mechanistic insight are provided, and it's not clear how they could be derived.
- Figures are hard to read—lots of information not easily presented and often text is too small to be legible.
- Without proper demonstration of standing waves, or resonance, or other physical claims, it's hard to justify subsequent claims, e.g., that functional network activation is associated with the "transient resonance of a standing wave..."
- The modeling does not provide insight. The model is phenomenological and its dynamics are presented with no fitting or quantitative comparison to data. No comparisons to alternatives are given, no predictions are made, and no clear claims are made of the model providing any insights/understanding. It is well known that under-damped oscillators driven by noise produce noisy, intermittent oscillatory dynamics.
- P16: "Conflicting hypotheses have been..." this sentence should be expanded in the introduction to set a clear scene for what open questions this work addresses.
- Avoid the physical claim of "resonant modes" when 'resonance' has not been demonstrated.

REFS:

- Robinson, P. A. Physical brain connectomics. *Physical Review E* 99, 012421 (2019).
- Liang, Y. et al. Cortex-wide dynamics of intrinsic electrical activities: propagating waves and their interactions. *J. Neurosci.* (2021).
- Huntenburg, J. M., Yeow, L. Y., Mandino, F. & Grandjean, J. Gradients of functional connectivity in the mouse cortex reflect neocortical evolution. *NeuroImage* 225, 117528 (2021).
- Casorso, J. et al. Dynamic mode decomposition of resting-state and task fMRI. *NeuroImage* 194, 42–54 (2019).
- Shine, J. M. et al. Human cognition involves the dynamic integration of neural activity and neuromodulatory systems. *Nature Neuroscience* 20, 1–296 (2019).

Reviewer #3 (Remarks to the Author):

Cabral et al have analyzed ultra-fast single slice fMRI data from rats in three different states from sedation to full anesthesia. Along this gradient they find a reduction in the number of "modes" defined as the eigenvectors of the covariance matrix. They also projected the empirical data back onto a lower dimensional space defined by a small number of modes and demonstrate that these models are a good approximation of the data. Finally they conclude that these modes are standing waves that correspond to resonance phenomena in the brain which can be described by a subcritical Hopf oscillator with stochastic perturbation and they argue that it is these resonances that drive the formation of functional networks.

I read this manuscript with great interest and I want to congratulate the authors on their thorough work. It is definitely to their credit that the analysis was done completely transparently and that all scripts and data have been made available.

That said, I am still not entirely convinced that the authors have provided sufficient evidence for their main conclusion, namely that the brain operates on resonance phenomena which manifest as functional networks. The movies are certainly intriguing to watch, but on second thought they are also somewhat suspicious. I am no expert on rat anatomy but if I would see these large-scale wave patterns moving around in a human brain, my first thought would be "artifact". If the authors can convince me that what I am looking at is not just narrowly band-passed signals whose spatial PCA components, when projected back onto the data, show not entirely surprising narrow-band oscillations, this would greatly help my doubts.

Here follows a list of more detailed comments, page by page:

p. 3: the authors argue that the ultra-fast resolution of 38ms was required to see the phenomena described here, yet they seem to be restricted to frequencies below 0.25Hz, ie a sampling rate of 0.5Hz (ie a TR of 2s). That's almost 2 orders of magnitude of oversampling. I can see that these concerns have already been partly addressed in SI Fig S12, but I didn't find any explanation why this happens.

p. 6: about movie S1 the authors write "sustained oscillations emerge consistently" but given that the data have been bandpassed severely to 0.2-0.25Hz this isn't surprising, it's expected. It's still interesting to see the spatial pattern of course.

p. 6, bottom: I think it's crucial to point out here, that all "modes" very defined on a very narrowly band-passed signal, because this might get lost a bit further down when modes are just labeled through ψ_1 , ψ_2 etc (in descending order of eigenvalue magnitude I believe, and irrespective of condition?).

p. 7, bottom: in the equation defining tau it would be nice if the notation could indicate that this is a matrix multiplication that contracts the "n" dimension

p. 8, Fig 2C-D: I prefer power-spectra to be semi-log (or log-log) but more importantly I don't completely understand panel C-D. Surely mode 2 shown here was defined in the sedated condition and in frequency band 0.2 which is why it shows such a narrow spectrum by construction? How about showing the mode that was defined in the same frequency band but in the respective condition for all three?

p. 8, Fig 2F-G: the spectrum in panel F and the Q values in panel G don't seem to always correspond, ie for mode ψ_1 the Q value seems much higher for light anesthesia than for sedation, yet the bar plot shows them to be equal

p. 10: the narrow-band activity in this movie is very intriguing and clearly the low-dimensional model with separated variables "n" and "t" captures it well. But what are we to make of these large waves traveling up the whole length of the lateral rat cortex?

p. 11, Fig 3: overlaying the squared amplitude here doesn't add much info, it's more distracting than helpful, especially since it even intersects with the timeseries. Either drop altogether or offset from timeseries to make it clearer.

p. 13, Hopf equation: should be $|Z|^2$ rather than $|Z^2|$. Since in the sentence right after the equation "external perturbations" are referred to, perhaps the perturbation term should be included here right away (as indeed it is in the SI)

p. 18, first paragraph: I think this discussion fails to mention that the timescale of the rat HRF is substantially lower than in humans (Lambers 2019, doi:10.1016/j.neuroimage.2019.116446) and is in fact right on top of the 0.25Hz frequency band where the strongest effects are found. So a devil's advocate could easily claim that what is resonating here is the brain vasculature when perturbed by true neuronal activity ...

Minor comments:

SI page 4, bottom: again $|Z|^2$. Also the sentence following the equation is rather unclear (and missing a bracket).

SI page 6: "Error! Reference source not found."

SI Fig 12: I don't understand why this is happening since the cross-over at around $TR=0.2s$ (Nyquist $5/2$ Hz) is still far away from the 0.2Hz frequency band (Nyquist $5/2$ s)

Point-by-point Rebuttal

We again thank the reviewers for their insightful comments and support of our work. Below, we reproduce the reviewers' comments, renumbering them for clarity where relevant and providing our responses and changes to the Manuscript in **blue font**.

Reviewer #1 (Remarks to the Author):

R1: In this manuscript, the authors developed an ultra-fast fMRI paradigm to track high-frequency brain functional oscillations, attempting to probe the fundamental mechanism underlying brain functional connectivity. They showed that obtained fMRI time course could be well fitted by a wave model; and hence concluded that observed spontaneous functional activity was driven by these resonant modes. While the resonant model is novel and connects with existing computational theories of brain activity, I have several questions/comments that hope the authors could address/clarify before publication.

A1: Thank you very much for the positive assessment of our work and for pointing out the remaining unclear points, which we address below:

R1.1: One major technical novelty of this manuscript is the employment of ultra-fast temporal sampling (38 ms) through reduced slice coverage, however, it is unclear why such a high sampling rate is necessary given that the highest frequency of interest explored here is only 0.4 Hz. According to supplementary Fig. S13, a TR faster than 2 s already suffices to resolve the 0.2/0.25 Hz resonant peak. While faster sampling may improve the detection of frequency-specific differences between conditions (Fig. S12), would removal of aliased high-frequency physiological sources in the low-TR condition, e.g., through model-based approaches, achieve comparable efficacy? I'm raising this concern because faster sampling will incur a penalty in the spatial resolution (or spatial coverage here) and signal-to-noise ratios (due to reduced T1 recovery time, i.e., marked increase of white noise).

A1.1: Thank you for raising this important point on the benefits of a sampling rate much faster than the Nyquist frequency (i.e., 26Hz vs 0.4Hz) for detecting oscillatory modes and in particular, for adequately characterizing their phase relationships and temporal signatures. Following the reviewer's concern, we have now run new analyses, varying the sampling rate by downsampling the recorded data, to demonstrate that our results strongly benefit from ultrafast acquisition sequences. This is now clearly illustrated in the new Figure 4 in the Manuscript and in the Supplementary Figures S10 to S14 (copied below).

By downsampling the data we found that the repertoire of modes detected, as well as the associated spectral power and frequency specificity, are strongly affected by the sampling

rate. Indeed, already at 380 milliseconds, only 4 principal components (instead of 10) are detected with eigenvalue above the postmortem baseline in medetomidine sedated rats (Figure S10).

Fig. S10 - Effect of the sampling rate on the number and spatial signature of the principal components detected. Before the analysis, the fMRI signals recorded with a TR of 38 milliseconds (ms) were downsampled by considering only one in every 5, 10, 15, 20 and 25 frames (corresponding to intervals of 190, 380, 570, 760 and 950 milliseconds between frames). The number of principal components detected corresponds to the number of eigenvalues larger than the largest eigenvalue from postmortem scans (signals also downsampled). As can be seen, although some of the components can still be detected for TRs up to 950ms, the ultrafast sampling rate used in this work reveals a larger number of components, clearly defined within cortical and subcortical structures.

We further investigated why sampling nearly 2 orders of magnitude faster than the Nyquist frequency is relevant in this context. Although the Nyquist theorem states that sampling at 2 points per oscillation is sufficient to detect an oscillation, if - as our data indicates - multiple oscillatory components are superposed and appear only transiently in time, operating near

the Nyquist frequency fails to capture the full spectrum of oscillatory components of a complex system.

Indeed, our data reveals that some of the modes detected at ultrafast sampling oscillate only for a few sustained cycles. This short-lived periodicity - indicative of oscillatory damping - is particularly visible in the temporal signatures at ultrafast sampling rates (see the new Figure 4 below), but gradually becomes indistinguishable from aperiodic fluctuations as the sampling rate approaches the Nyquist frequency. This sustained periodicity is also now more visible in the new Figures 3 to 5 and in the autocorrelation functions in the new Figure 6.

Figure 4 – Effect of the sampling rate in the detection of short-lived oscillations. The temporal signal $\tau_7^S(t)$ associated to the 7th principal component detected in ultrafast fMRI signals from medetomidine-sedated rats (Time of Repetition, TR= 38 milliseconds (ms)) is downsampled by considering only one in every 5, 10, 15, 20, 25, 30 and 35 frames (corresponding to intervals of 190, 380, 570, 760, 950, 1140 and 1330 ms between frames). Plots shown for 150 seconds from a representative

scan **S** (same from **Figure 3**). The power spectra of the temporal signals computed for the entire 10 minutes recorded for scan **S** (blue) and for a scan performed postmortem (red).

Below we copy the relevant text insertions in the manuscript in the Introduction and Results:

New text in the Introduction:

'The precise characterization of oscillations across space and over time is complex and benefits from an adequate spatiotemporal resolution and high signal-to-noise (SNR) ratio to adequately capture transient phase-relationships between voxels. Ultra-high field fMRI studies in rats achieve increased SNR by attenuating thermal noise using cryogenic coils^{1,2}, allowing for increased precision in the characterization of oscillatory signals.'

New text in the Results section:

'Importantly, the detection of transiently sustained periodicity benefits from sampling rates significantly faster than the Nyquist frequency, to ensure that sufficient points are captured at the oscillation frequency and reveal a distinctive peak in the power spectrum. As shown in Figure 4, a transient oscillation detected at 0.22Hz in the temporal signature of a given principal component (here $\psi_{\alpha=7}$) becomes indistinguishable from aperiodic fluctuations as the sampling approximates the Nyquist frequency (corresponding to 2.27 seconds for 0.22 Hz oscillations). In Supplementary figures S10-S14 we further demonstrate how the detection of oscillatory modes benefits from ultrafast sampling rates, with component $\psi_{\alpha=7}$ failing to be detected with a sampling as fast as 190ms.'

Furthermore, we note that aliasing of physiological noise into the conventionally sampled data is certainly one of the reasons why it is more difficult to observe the 0.1-0.4 Hz peaks in the TR=2s downsampling. With this high sampling rate we were able to resolve frequency components up to 13 Hz, which prevents major physiological noises from aliasing with the signal of interest. Having said that, we found that a TR of around 150-200ms is already sufficient for observing most of the oscillatory modes in the brain, which is promising for future applications of the technique in e.g. human samples.

R1.2: Prior to this work, several human/animal studies have already evaluated the temporal characteristics of high-frequency spontaneous functional connectivity (even > 0.5 Hz); this line of literature is missing in the current manuscript. To list a few:

[1] Lee, H.L., Zahneisen, B., Hugger, T., LeVan, P. and Hennig, J., 2013. Tracking dynamic resting-state networks at higher frequencies using MR-encephalography. *Neuroimage*, 65, pp.216-222.

[2] Chen, J.E. and Glover, G.H., 2015. BOLD fractional contribution to resting-state functional connectivity above 0.1 Hz. *Neuroimage*, 107, pp.207-218.

[3] Lin, F.H., Chu, Y.H., Hsu, Y.C., Lin, J.F.L., Tsai, K.W.K., Tsai, S.Y. and Kuo, W.J., 2015. Significant feed-forward connectivity revealed by high frequency components of BOLD fMRI signals. *Neuroimage*, 121, pp.69-77.

[4] Trapp, C., Vakamudi, K. and Posse, S., 2018. On the detection of high frequency correlations in resting state fMRI. *NeuroImage*, 164, pp.202-213.

A1.2: Thank you very much for pointing out these important paper that we inadvertently overlooked. These papers are now fruitfully cited in the Introduction section (except the paper from Lin et al., which is specific to a visual task, while we focus here on intrinsic activity). It is worth mentioning that while these papers demonstrate the spatial detection of high-frequency correlations (reinforcing the validity of our results), they did not explore the temporal dynamics - and in particular the oscillatory nature - of the signals driving the correlation patterns, and we know reinforce this aspect to justify the focus of our work.

New text in the Introduction:

'Dynamically, correlated fluctuations in fMRI signals exhibit power at ultra-slow frequencies, peaking typically below 0.1 Hz in human brains at rest, although intrinsic functional networks have been detected at frequencies extending even beyond 0.5 Hz³⁻⁶. Crucially, it remains unclear whether the spectral power at low frequencies is associated with aperiodic activations of the characteristically slow hemodynamic response function or instead reflects the existence of oscillatory phenomena with sustained periodicity over consecutive cycles^{7,8}. Given recent insights demonstrating that the fMRI signals underpinning intrinsic networks cannot be exclusively associated to the blood oxygenation level dependent (BOLD) signal^{9,10} and relate to macroscopic waves of activity¹¹⁻¹³, it is crucial to obtain a detailed characterization of the modes' temporal signatures to investigate their generative mechanisms.'

R1.3: It is unclear how the signal models evaluated for the current dataset (sedated rat) can extend to awake human conditions, due to several reasons. First, extensive human fMRI studies have failed to see the resonant peaks at 0.2/0.25 Hz as reported here, even with sampling rates faster than the Nyquist limits. Second, there has been emerging evidence based on rodent models supporting that brain connectivity may arise from oscillatory vasomotor activity, which however occurs at 0.1 Hz, instead of 0.2 Hz, during awake conditions (Mateo, C., Knutsen, P.M., Tsai, P.S., Shih, A.Y. and Kleinfeld, D., 2017. Entrainment of arteriole vasomotor fluctuations by neural activity is a basis of blood-oxygenation-level-dependent "resting-state" connectivity. *Neuron*, 96(4), pp.936-948.). Third, many studies have already shown that anesthesia could have a strong impact on the measured BOLD signal properties and associated brain connectivity (Paasonen, J., Stenroos, P., Salo, R.A., Kiviniemi, V. and Gröhn, O., 2018. Functional connectivity under six anesthesia protocols and the awake condition in rat brain. *Neuroimage*, 172, pp.9-20.).

A1.3: The reviewer raises an important point on the generalizability of the current findings across species. We now more clearly explain that the resonant peaks at 0.2/0.25 are known to be induced by medetomidine, and we also now adequately refer to the resonance induced by arteriole vibrations at 0.1 Hz. We also more clearly describe previous findings across

sedation/anaesthesia protocols, which serve as the starting point for this study. We further expand on the issues raised by the Reviewer both in the Introduction and in the Discussion (see below).

New text in Introduction:

'Studies in rodents and humans have shown that ultra-slow (<0.5Hz) frequency components in fMRI signals have a periodic nature and are coupled with electrophysiological and electroencephalographic (EEG) signals¹⁴⁻¹⁸. These periodic fluctuations have been proposed to be linked to arteriole vibrations entrained by ultra-slow oscillations in local field potentials, pointing to a more direct relationship with the underlying neural activity^{19,20}. Still, how these oscillations organize at the macroscopic level and their relationship to 'functional connectivity' between brain areas remains unclear.'

'Intrinsic networks analogous to the ones identified in humans have been identified in rats and to be modulated by the sedation/anesthesia state²¹⁻²⁵. In particular, sedation with low doses of medetomidine has been shown to reveal consistent patterns but also to drive abnormal high amplitude oscillations in fMRI signals at frequencies extending beyond 0.1 Hz^{15,22}. The addition of isoflurane at low concentrations suppresses these 'high frequency' oscillations while maintaining the typical human resting-state frequencies <0.1 Hz, such that the combination medetomidine/isoflurane (MED/ISO) is currently the state-of-the-art protocol to approximate 'resting-state' brain activity in rats^{26,27}.'

New text in the Discussion

'The generalization of these findings to other animal species including humans can only be discussed in the light of existing literature. On one side, the similarity of the principal components detected herein with intrinsic network patterns detected using different methodologies suggest these are expressions of the same emergent phenomena, typically referred to as 'resting-state networks' (RSNs) or 'intrinsic connectivity networks' (ICNs). Since both rats, mice, monkeys, and humans exhibit qualitatively similar RSNs/ICNs, it can be expected that they are expressions of the same phenomenon. Indeed, a wide range of studies have demonstrated that intrinsic modes (detected across modalities) exhibit spatial features of standing waves, so it can be expected that their temporal signature reflects a damped harmonic motion. And even if no clear periodicity is detected and the fluctuations closely approximate the canonical hemodynamic response function, one cannot exclude the possibility of the intrinsic modes operating in the overdamped regime in resting humans, providing a new generative model for the signals observed empirically.'

'The resonant quality of the intrinsic modes was found to be modulated by anesthetics. While under medetomidine alone, strong oscillations were detected particularly in the cortex up to 0.3Hz, in agreement with previous literature¹⁵, the addition of, isoflurane at 1% concentration was found to particularly suppress oscillations between 0.15 and 0.25 Hz, leaving the activity

in the typical range considered in resting-state studies (i.e. < 0.1 Hz) mostly unchanged, approximating empirical observations of human resting state activity. Further increasing the isoflurane concentration to 3%, most oscillatory power is lost and only very slow (<0.05Hz) globally fluctuations are detected. Still, it remains unclear whether these non-linear effects are related with the differential effects of medetomidine and isoflurane on blood vessels or can be explained by more direct changes in the resistivity of the medium through which the waves propagate.'

In addition, we can reveal, only for the purpose of this rebuttal letter, that we have recently recorded ultrafast fMRI (TR=68ms) from a resting human (one of the authors) and we did detect components >0.1 that reveal oscillatory modes organized across space. However, it is important to consider that the human breathing frequency overlaps with the frequency range observed herein, so these components are typically removed at the preprocessing level in human data. Although the dataset is still under analysis and the results are very preliminary, it can clearly be seen in the following confidential video (<https://shorturl.at/ioqBD>) that spatially-organized fluctuations qualitatively similar to the ones detected in the current work can be detected up to 0.6 Hz (videos were differently accelerated to facilitate visualization of different frequencies). Although in the current paper we leave the source of the signals an open question, we note that the human fMRI signals in cortex appear closely linked to the fMRI signal in the ventricles, suggesting a link with CSF dynamics, in line with recent literature^{9,18}.

Since the scope of this paper was not to investigate human rsfMRI but rather to point out the importance of structured oscillations, we do not include human data yet, and reserve it for future publications.

R1.4: Finally, the major conclusion from this work—functional network patterns emerge from a repertoire of standing waves resonating at frequencies up to 0.25 Hz—is not well justified. While the resonant models have already been adopted to depict neuronal oscillations, they may not extend to fMRI signals, because the spectral property of BOLD signals is blurred by region-specific hemodynamic changes. Apart from the distinct signal properties of “wake” vs “sedation” state as discussed above, even if the resonant model can well fit the fMRI data, we cannot conclude that the resonance phenomenon is the fundamental mechanism underlying spontaneous functional behavior.

A1.4: We agree that our main message was not sufficiently well explained and justified in the previous version of the manuscript. As the reviewer will find, in this new version of the manuscript we have significantly expanded the results with more rigorous analysis and tempered our claims in the discussion regarding the conclusions that can be drawn. We now reinforce that this work proposes a novel mechanistic hypothesis, which, as in general scientific knowledge, can never be fully confirmed but remains valid until refuted.

In more detail, we now analyzed the correlation-based functional connectivity (new Fig. 1) and subsequently show that the long range patterns emerge from the transient oscillation of stationary waves (new Figs. 3-5), fortifying the conclusion that “functional connectivity” (both dynamic and static) is a signature of resonating stationary waves.

Regarding the reviewer’s commentary: *‘because the spectral property of BOLD signals is blurred by region-specific hemodynamic changes’*; we agree that understanding how functional connectivity itself relates with neurophysiological signals is an important question for future research. Nevertheless, we have opted to leave the relation to the hemodynamic response and/or neurophysiology as an open question, particularly given the recent literature questioning the ‘BOLD’ nature of resting-state fMRI signals. Instead, we leave our results purely objective, given that the damped oscillation of stationary wave patterns is a universal principle that is not bound to any particular physiological process but instead can be expressed through many forms in the natural world.

New text in the Abstract: *‘Oscillatory modes are found to vary between conditions, resonating at faster frequencies under medetomidine sedation and reducing both in number, frequency, and duration with the addition of isoflurane. Peaking in power within clear anatomical boundaries, these oscillatory modes point to an emergent systemic property, questioning current assumptions regarding the local origin of oscillations detected in fMRI and providing novel insights into the self-organizing principles underpinning spontaneous long-range functional connectivity.’*

New text in the Discussion: *‘Using fMRI experiments with hitherto unprecedented spatiotemporal resolution we were able to provide new insights into this problem by demonstrating the existence of oscillatory modes in fMRI signals, organizing with fixed phase relationships across space and oscillating transiently over time, driving correlated activity between distant regions.’*

R1.5 The authors should, at least, either demonstrate that alternative models (e.g., scale-free models) cannot fit the data, or perform simulations to validate whether the characterized covariance modes are reliable. For instance, the author could generate a batch of sham datasets (with coherent, scale-free fluctuations for different brain networks), and go through the identical modelling to test whether the analytical outputs of the sham data have distinct spatial signatures from the modes reported here, e.g., not with “anti-nodes standing-wave” features.

A1.5: Thank you for pointing this out. We agree that the previous ‘model’ was not adequately validated, and we now perform simulations to support our hypothesis that the phenomenology of intrinsic connectivity networks detected in the resting state is the

signature of transiently resonating standing waves. We now have a dedicated subsection entitled ‘Stochastic Resonance of standing waves’ (pages 16-18).

In addition, our main message is now better explained and more explicit in the revised paper. On one side, we obtain empirical evidence that, under sedation with medetomidine, the spatial patterns transiently oscillate in time (and these oscillations are lost if the spatial patterns are randomized).

This empirical evidence is shown in the new Figure 6 (copied below), where in sedated and lightly anesthetized rats, the temporal signatures associated to the spatial patterns have a resonant frequency (a narrow peak > 0 Hz) and exhibit sustained periodicity for several consecutive cycles (as captured by the Resonance Q factor). These transient oscillations disappear consistently in deeply anesthetized animals, and only ultra-slow (< 0.05 Hz) and aperiodic (i.e., without sustained periodicity) global fluctuations are detected.

Figure 6 – Principal components oscillate at higher frequencies and with less damping under medetomidine. (a,b) The temporal signatures associated with the principal components detected in each condition are characterized in terms of peak frequency and Q-factor for each of the 12 scans in each condition (2 scans per rat per condition). Error bars represent the mean \pm standard error across scans. **(c)** To illustrate the stability of the oscillations, the autocorrelation functions of the temporal signatures τ_1^S associated to the first principal component in each condition are reported. Examples are shown for 3 scans from the same rat and from a postmortem scan. As can be seen, the autocorrelation function under medetomidine

exhibits 3 oscillations before the amplitude decays to $1/e$ (~37%), 2 cycles after adding isoflurane at 1% and no complete cycle under deep anesthesia, similar to what is observed in the postmortem scan.

Regarding the reviewer's suggestion to 'perform simulations to validate whether the characterized covariance modes are reliable' we believe that the clear overlap between the patterns detected herein and the ones previously described in the rodent literature (we copy some examples below) provides the best evidence that the modes detected are reliable across studies and directly related to what is typically referred to as 'resting-state networks' in rats.

Bajic, Dusica, et al. "Identifying rodent resting-state brain networks with independent component analysis." *Frontiers in Neuroscience* 11 (2017): 685.
<https://doi.org/10.3389/fnins.2017.00685>

Li, Q., Li, G., Wu, D., Lu, H., Hou, Z., Ross, C. A., ... & Duan, W. (2017). Resting-state functional MRI reveals altered brain connectivity and its correlation with motor dysfunction in a mouse model of Huntington's disease. *Scientific reports*, 7(1), 1-9.

Gutierrez-Barragan, Daniel, et al. "Infraslow state fluctuations govern spontaneous fMRI network dynamics." *Current Biology* 29.14 (2019): 2295-2306.

<https://www.sciencedirect.com/science/article/pii/S0960982219307109#ack0010>

To validate that the oscillations are directly associated with the spatial patterns detected, we randomized the spatial patterns and obtained the associated temporal signatures. As shown in the new Supplementary Figure S15 (copied below), once the spatial configuration of phase relationships is randomized, no oscillations are detected, and the power spectrum approximates the one from postmortem scans. This serves to demonstrate that it is the spatial wave patterns that resonate at a fine tuned frequency.

Fig. S15 - The oscillations are specific to the spatial patterns. To demonstrate that the oscillations are specific to the spatial wave patterns detected, we randomize the elements in 3 representative spatial patterns and demonstrate that, when projecting the randomized spatial patterns (with size $1 \times N$) to the same, unfiltered, fMRI signals (with size $N \times T$), no oscillations are detected in the associated temporal signature (with size $1 \times T$), with the power being close to what is detected from a postmortem scan (red lines). The power spectra are shown in both linear and logarithmic (log) scales.

In the new section of the results dedicated to the phenomenological model, we now further explain that this model does not exclude the scenario of scale-free fluctuations, but it just considers it a particular case where the oscillatory modes are overdamped:

New text in the Results section:

'As shown in Figure 7, the stochastic resonance of a repertoire of standing waves (here considering the repertoire detected empirically in sedated animals) results in a spatiotemporal pattern sharing features with what is detected from fMRI recordings. This model includes the possibility to tune the oscillators in the overdamped regime, in which case it can approximate the results obtained in deeply anesthetized animals, where no resonant oscillations are detected but only aperiodic fluctuations. In other words, the model of stochastic resonance does not exclude the hypothesis of scale-free fluctuations driving the fMRI signals, but it considers it to be a particular case where the oscillatory modes are overdamped.'

Reviewer #2 (Remarks to the Author):

R2: SUMMARY: Cabral et al. present an analysis of the low-dimensional components of rat fMRI, at 'ultrafast' temporal resolution in a slice and at standard temporal resolution in the whole brain. Applying a linear dimensionality reduction method based on covariance modes (resembling principal components analysis), they find spatial modes that resemble standing-wave patterns with time courses that have oscillatory components, consistent with prior work applying dimensionality reduction to fMRI time series in rodent and human. Unfortunately, relevant context is missing, the claims are over-stated given the evidence, the phenomenological model is not quantitatively connected to the experimental data and does not provide insights, and the methods are often over-interpreted.

A2: Thank you for the critiques - we completely agree that the context could have been improved and trust that we have now done a better job at highlighting the novelty and uniqueness of our results. As the reviewer will find, this included a complete re-analysis of the paper to focus on the fact that the modes exhibit oscillatory behavior (which, despite having been mentioned in a few previous studies, has not been adequately demonstrated and is certainly not solidly established in the field). Furthermore, our phenomenological model now serves to demonstrate how the stochastic resonance of modes with damped oscillatory response drives correlation patterns (i.e., 'functional connectivity') similar to the ones detected empirically, validating it as a possible mechanistic scenario to explain the transient correlations detected in brain activity during rest. We expect this mechanistic hypothesis will provide fertile grounds for the formulation of falsifiable predictions to be further tested.

MAJOR: Novelty, and appropriateness of methods for extracting 'resonant modes':

R2.1: Summary of literature on modes and waves is incomplete in introduction. There is a lot of work that has been done on this that is not sufficiently explained/cited. It's therefore unclear what the scope of this study is given the amount of prior work. E.g., lots of work has done similarly as here (something similar to principal components analysis of fMRI data) in many species and resolutions, and found similar results. Others have found low-dimensional gradients in rodent (mouse) fMRI (cf. Huntenburg, also in human). These highly explanatory spatial maps (in human, mouse, ...) always look like standing-wave patterns, as reported here. Of the many papers reporting low-dimensional modes, I can't think of any that didn't exhibit "clear standing wave properties" with the characteristics listed in this paper (i.e., having nodes and antinodes).

A2.1: We agree with the reviewer that several studies have applied principal component analysis (or any type of component decomposition), and we now expanded the reference to previous literature in the Introduction. Indeed, the eigenmodes of covariance - or more colloquially the Principal Components - of brain signals have long been found to exhibit

standing wave features at the spatial level (ranging from EEG microstates to resting-state networks in fMRI). However, to our knowledge, it has not been ubiquitously demonstrated that the principal components of fMRI signals exhibit transient fine-tuned oscillations over time, and, although it is known that anesthetics affect the detection of resting-state networks, it has not been shown that it specifically affects the frequency and duration of these oscillations.

Following the reviewer's comment, in this new version of the manuscript we now make this more explicit, reinforcing that the true novelty of our work is not related to the spatial wave patterns of the components, but to the *oscillatory nature of their temporal signatures*, which is optimally captured at ultrafast sampling rates and is not induced by band pass filtering (as clearly demonstrated in the new Figure 4 copied below).

New text in Introduction:

'A wide range of low-rank decomposition techniques have been put forward to characterize the spatial organization of spontaneous fMRI signal fluctuations, including among others independent component analysis^{28,29}, co-activation patterns^{30,31}, low-dimensional gradients^{32,33}, leading eigenvector dynamics analysis^{34,35} or dynamic mode decomposition⁸. Despite the differences inherent to each technique, most methods converge in a discrete repertoire of intrinsic modes exhibiting features of stationary wave patterns, where correlated activity is detected among spatially distributed regions (or poles), with gradually varying phase relationships across space³⁶. These intrinsic modes have been shown to emerge transiently and recurrently during rest^{37,38}, to be selectively recruited during specific tasks²⁹ and to replicate across mammals^{24,39-42}.'

New Figure (next page)

Figure 4 – Effect of the sampling rate in the detection of short-lived oscillations. The temporal signal $\tau_7^S(t)$ associated to the 7th principal component detected in ultrafast fMRI signals from medetomidine-sedated rats (Time of Repetition, $TR=38$ milliseconds (ms)) is downsampled by considering only one in every 5, 10, 15, 20, 25, 30 and 35 frames (corresponding to intervals of 190, 380, 570, 760, 950, 1140 and 1330 ms between frames). Plots shown for 150 seconds from a representative scan S (same from Figure 3). The power spectra of the temporal signals computed for the entire 10 minutes recorded for scan S (blue) and for a scan performed postmortem (red). No band-pass filtering is applied to the time series.

R.2.2: Others have done detailed and specific explorations of wave propagation (e.g., in mouse, cf. Liang), or used techniques explicitly designed to extract standing-wave patterns with oscillatory time courses from data, like dynamic mode decomposition (DMD, cf. Casorso and others), or derived spatial patterns consistent with eigenmodes of brain geometry (cf. Robinson).

Here, the authors should more completely explain the relevance of this prior work and the open questions addressed—many results are described as surprising or 'striking' that are consistent with similar prior studies.

A2.2: We completely agree with the reviewer that a number of studies have previously addressed the oscillatory nature of intrinsic modes in fMRI signals, and we now adequately cite the prior work (cf. our previous comment). Importantly, as the reviewer may verify, the papers suggested above mainly look at the *spatial* variation of different modes (which indeed resemble standing waves) and do not report the associated temporal signatures nor the associated power spectra.

As the reviewer will agree, it is definitely not widely accepted in the field that long range functional networks emerge from the transient oscillation of the eigenmodes, and we now better reinforce the open questions in the Introduction.

New text in the Introduction:

“Dynamically, correlated fluctuations in fMRI signals exhibit power at ultra-slow frequencies, peaking typically below 0.1 Hz in human brains at rest, although intrinsic functional networks have been detected at frequencies extending even beyond 0.5 Hz³⁻⁶. Crucially, it remains unclear whether the spectral power at low frequencies is associated with aperiodic activations of the characteristically slow hemodynamic response function or instead reflects the existence of oscillatory phenomena with sustained periodicity over consecutive cycles^{7,8}. Given recent insights demonstrating that the fMRI signals underpinning intrinsic networks cannot be exclusively associated to the blood oxygenation level dependent (BOLD) signal^{9,10} and relate to macroscopic waves of activity¹¹⁻¹³, it is crucial to obtain a detailed characterization of the modes’ temporal signatures to investigate their generative mechanisms.”

R2.3. While the authors use a linear dimensionality reduction method (PCA-like: an eigenvalue decomposition of the covariance matrix), it is interpreted as being "based on wave physics" (when it is not) and therefore explicitly extracts standing-wave patterns (whereas the temporal profiles are unrestricted with covariance modes). Applying a band-pass filter to a very tight range of frequencies around the peak of the power spectrum accentuates the oscillatory appearance of these resulting time courses.

More could be done to set up what exactly is the expectation from prior literature in order to make the claim of what here is surprising/interesting and therefore why it might have an impact on the field.

A2.3: We agree that there was an ambiguity in the previous manuscript in what refers to PCA or wave physics, and how this relates to prior work. In this new version of the manuscript we now start by explaining that previous works have detected ‘standing-wave like’ patterns, and expose the missing characterization of their temporal signatures.

In the Results section, we now separate the analysis of principal components in a subsection entitled: ‘*Spatial, temporal and spectral properties of principal components*’. The following subsection entitled ‘*The oscillatory nature of principal components*’ shows that the principal components oscillate consistently across rats and exhibit consistent changes in spectral power with the addition of isoflurane.

Only after verifying that the principal components oscillate transiently in time, do we now mention the hypothesis that the oscillatory modes represent the transient resonance of stationary waves, which, given general principles of wave superposition (from physics), are captured using PCA since their temporal signatures are orthogonal.

Furthermore, regarding the passband argument, we can ensure that the oscillations are NOT accentuated due to band pass filtering. In this new version of the manuscript we now demonstrate that the oscillations are tuned at specific frequencies, being clearly visible in broadband signals (as shown in Figure 4 copied in the previous comment). Additionally, to demonstrate that it is the spatial patterns that are associated to the oscillations, we randomized the spatial patterns and obtained the associated temporal signatures. As observed in the new Supplementary Figure S15 (copied below), no resonant frequency peak is detected when the spatial patterns are randomized, and the power spectrum of the associated temporal signatures approximates the postmortem baseline.

Fig. S15 - The oscillations are specific to the spatial patterns. To demonstrate that the oscillations are specific to the spatial wave patterns detected, we randomize the elements in 3 representative spatial patterns and demonstrate that, when projecting the randomized spatial patterns (with size $1 \times N$) to the same, unfiltered, fMRI signals (with size $N \times T$), no oscillations are detected in the associated temporal signature (with size $1 \times T$), with the power being close to what is detected from a postmortem scan (red lines). The power spectra are shown in both linear and logarithmic (log) scales.

We are confident that these new analyses reinforce the validity of our results.

MINOR:

R2.4:- Opening results are unclear as to why the analyses are done (motivation is missing). Is the aim to understand whether different spatiotemporal patterns can be extracted at this high sampling rate relative to a standard sampling rate?

A2.4: Thank you for raising this concern. Indeed, the benefits of a high sampling rate were not adequately demonstrated. In this revised version of the manuscript we now further explore the impact of the sampling rate and we expand on this point in our response to your comment R2.11.

R2.5:- "isoflurane nonlinearly modulates the power spectrum" -> 'non-uniformly'?

A2.5: Revised as suggested, thank you.

R2.6:- It's not clear why it's "remarkable" that covariance modes are found. This is consistent with prior work. It would be remarkable if they were not found.

A2.6: We agree and have rephrased. What is remarkable is that they oscillate, which is not necessarily a property of principal components given that signals can co-vary aperiodically, without necessarily oscillating. This has been corrected.

R2.7 It's not clear why it's "even more remarkable" that different frequency bands are associated with different covariance modes. It would be remarkable if the opposite was true: i.e., that modes were identical across all frequency ranges.

A2.7: We have now removed this part of the analysis, since we now perform the component analysis in a broad frequency range and only subsequently analyze the temporal signatures associated to each mode in each scan. This new analysis revealed that although the spatial modes show strong spatial similarities across rats, the frequency at which the modes resonate varies strongly across rats and decreases with anesthesia (as shown in new Figure 6).

Figure 6 – Principal components oscillate at higher frequencies and with less damping under medetomidine. (a,b) The temporal signatures associated with the principal components detected in each condition are characterized in terms of peak frequency and Q-factor for each of the 12 scans in each condition (2 scans per rat per condition). Error bars represent the mean \pm standard error across scans. **(c)** To illustrate the stability of the oscillations, the autocorrelation functions of the temporal signatures T_1^S associated to the first principal component in each condition are reported. Examples are shown for 3 scans from the same rat and from a postmortem scan. As can be seen, the autocorrelation function under medetomidine exhibits 3 oscillations before the amplitude decays to $1/e$ ($\sim 37\%$), 2 cycles after adding isoflurane at 1% and no complete cycle under deep anesthesia, similar to what is observed in the postmortem scan.

R2.8 "detached mode" undefined.

A2.8: Thank you for pointing that out. This has now been removed.

R2.9 Main text is too brief—the reader has to do a lot of work to understand why analyses are being done, and also to work out what was done—motivation is often missing and so are the methodological details required to understand what is being presented. E.g., the number of modes detected the reader needs to try to find what this means (either finding clues in annotations on the figure or reading supplement).

A2.9: We agree with the reviewer and have now extended the Introduction, Results and Methods sections. Given the comments from all 3 referees, the entire text was significantly revised to better accommodate the logic of our workflow.

With respect to the number of modes specifically, we have now modified the approach to ensure we report the modes detected in each different condition. This is now explicitly explained in the Methods and more clearly illustrated in Figures 3 and 5.

R2.10 P7: the "analytical framework based on wave physics" doesn't appear to have any grounding in wave physics, but closely resembles a principal components analysis decomposition.

A2.10: We thank the reviewer for raising the importance of better explaining our rationale, and of what exactly is based on wave physics, which is certainly not the PCA.

In line with our reply to your comment R2.3, in this revised version, these two effects (PCA and oscillating of PC's) are separately defined in two distinct Results subsections: '*Spatial, temporal and spectral properties of principal components*' and '*The oscillatory nature of principal components*'.

Only after verifying that the principal components oscillate transiently in time, do we now mention the hypothesis that the oscillatory modes represent the transient resonance of stationary waves, which, given general principles of wave superposition (from physics), can be captured using PCA since their temporal signatures are orthogonal.

R2.11 Unclear why data are bandpass filtered up to 0.4 Hz when the main novelty of this paper is billed as the dynamics on 'ultrafast' fMRI timescales. The higher frequency bands also don't seem relevant in Fig. 1. There is a disconnect between the narrative about the novelty of ultrafast fMRI versus the actual results presented, showing interesting patterns at slower frequencies (accessible to conventional fMRI).

A2.11: Thank you for raising this important point on the benefits of a sampling rate much faster than the Nyquist frequency (i.e., 26Hz vs 0.4Hz) for detecting ultra-slow oscillatory modes and in particular, for adequately characterizing their phase relationships and temporal signatures, while avoiding aliasing from higher frequency components.

Following the reviewer's concern, we have run new analysis, varying the sampling rate by downsampling the recorded data, to demonstrate that our results, despite being focused in the ultra-slow range <0.4Hz, strongly benefit from ultrafast acquisition sequences. This is now clearly illustrated in the new Figure 4 in the Manuscript and in the Supplementary Figures S10 to S14 (copied below).

By downsampling the data we found that the repertoire of modes detected, as well as the associated spectral power and frequency specificity, are strongly affected by the sampling rate. Indeed, already at 380 milliseconds, only 4 principal components (instead of 10) are detected with eigenvalue above the postmortem baseline in medetomidine sedated rats (Figure S10).

Fig. S10 - Effect of the sampling rate on the number and spatial signature of the principal components detected. Before the analysis, the fMRI signals recorded with a TR of 38 milliseconds (ms) were downsampled by considering only one in every 5, 10, 15, 20 and 25 frames (corresponding to intervals of 190, 380, 570, 760 and 950 milliseconds between frames). The number of principal components detected corresponds to the number of eigenvalues larger than the largest eigenvalue from postmortem scans (also downsampled). As can be seen, although some of the components can still be detected for TRs up to 950ms, the ultrafast sampling rate used in this work reveals a larger number of components, clearly defined within cortical and subcortical structures.

We further investigated why sampling nearly 2 orders of magnitude faster than the Nyquist frequency is relevant in this context. Although the Nyquist theorem states that sampling at 2 points per oscillation is sufficient to detect an oscillation, if - as our data indicates - multiple oscillatory components are superposed and appear only transiently in time, operating near the Nyquist frequency fails to capture the full spectrum of oscillatory components of a complex system.

Indeed, our data reveals that some of the modes detected at ultrafast sampling oscillate only for a few sustained cycles. This short-lived periodicity is particularly visible in the temporal signatures of a selected mode at ultrafast sampling rates (see the new Figure 4 below), but gradually becomes indistinguishable from aperiodic fluctuations as the sampling rate approaches the Nyquist frequency. This transiently sustained periodicity is also now more visible in the new Figures 3 to 5 and in the autocorrelation functions in the new Figure 6.

Figure 4 – Effect of the sampling rate in the detection of short-lived oscillations. The temporal signal $\tau_7^S(t)$ associated to the 7th principal component detected in ultrafast fMRI signals from medetomidine-sedated rats (Time of Repetition, $TR= 38$ milliseconds (ms)) is downsampled by considering only one in every 5, 10, 15, 20, 25, 30 and 35 frames (corresponding to intervals of 190, 380, 570, 760, 950, 1140 and 1330 ms between frames). Plots shown for 150 seconds from a representative scan S (same from **Figure 3**). The power spectra of the temporal signals computed for the entire 10 minutes recorded for scan S (blue) and for a scan performed postmortem (red).

Below we copy the relevant text insertions in the manuscript in the Introduction and Results:

New text in the Introduction:

‘The precise characterization of oscillations across space and over time is complex and benefits from an adequate spatiotemporal resolution and high signal-to-noise (SNR) ratio to adequately capture transient phase-relationships between voxels. Ultra-high field fMRI

studies in rats achieve increased SNR by attenuating thermal noise using cryogenic coils^{1,2}, allowing for increased precision in the characterization of oscillatory signals.'

New text in the Results section:

'Importantly, the detection of transiently sustained periodicity benefits from sampling rates significantly faster than the Nyquist frequency, to ensure that sufficient points are captured at the oscillation frequency and reveal a distinctive peak in the power spectrum. As shown in Figure 4, a transient oscillation detected at 0.22Hz in the temporal signature of a given principal component, $\tau_7^S(t)$, becomes indistinguishable from aperiodic fluctuations as the sampling approximates the Nyquist frequency (corresponding to 2.27 seconds for 0.22 Hz oscillations). In Supplementary figures S10-S14 we further demonstrate how the detection of oscillatory modes benefits from ultrafast sampling rates.'

Furthermore, we note that aliasing of physiological noise into the conventionally sampled data is certainly one of the reasons why it is more difficult to observe the 0.1-0.4 Hz peaks in the TR=2s downsampling. With this high sampling rate we were able to resolve frequency components up to 13 Hz, which prevents major physiological noises from aliasing with the signal of interest. Having said that, we found that a TR of around 150-200ms is already sufficient for observing most of the oscillatory modes in the brain, which is promising for future applications of the technique in e.g. human samples.

R2.12 Eq. (1) is not a "mechanistic framework of wave superposition", it corresponds to keeping the first 10 PCs. A mechanistic framework is approximately there with DMD, and more closely there with geometric eigenmodes, but applying PCA to a data matrix doesn't have any clear grounding in physical/wave mechanisms. Also overstated: "this approach provides insight into the mechanisms driving spontaneous brain network dynamics." This claim is made but no examples of mechanistic insight are provided, and it's not clear how they could be derived.

A2.12: We again apologize for the lack of clarity in the previous version of the manuscript. We now separate the different parts of the results, to reinforce that PCA is only the first step of the analysis to detect the modes. We subsequently analyze the temporal signatures, first for a representative rat (Figures 3 and 5) and subsequently across rats (Figure 6), to show that the temporal signals of the principal components do not simply co-vary aperiodically, but oscillate with transiently sustained periodicity.

We fully agree with the mechanistic hypothesis that the spatial configuration of the phase relationships is associated with geometric eigenmodes defined from the anatomical structure of the brain, and here we provide complementary evidence for this hypothesis, given that we demonstrate that the principal components (likely shaped by brain geometry) oscillate. We find it relevant to reinforce that the detection of oscillatory standing waves from experimental

signals (i.e., in acoustics and fluid dynamics), is a signature of resonance phenomena, which can be subsequently linked to geometric eigenmodes in order to find its generative mechanisms.

We also corrected from 'giving insight' to suggesting. The idea is to engage the reader for the next section.

R2.13 Figures are hard to read—lots of information not easily presented and often text is too small to be legible.

A2.13: All figures were modified and improved for clarity. We hope the reviewer will find them better but are also open to suggestions for improvement.

R2.14 Without proper demonstration of standing waves, or resonance, or other physical claims, it's hard to justify subsequent claims, e.g., that functional network activation is associated with the "transient resonance of a standing wave..."

A2.14: Thank you for raising this critical point. We now clarify the fact that here we do not aim to demonstrate the link of the spatial patterns to brain geometry, but instead we demonstrate that the spatial patterns detected as principal components from brain signals oscillate in time, and change with anesthesia.

Among the possible scenarios to generate oscillations in spatial patterns, we consider resonance to be a valid hypothesis, given that the phenomenology of standing waves is inherently associated with resonance phenomena, and the principal components exhibit properties of standing waves, i.e, organizing with gradual phase gradients across well-defined brain structures (i.e., the cortex and striatum) into nodes and anti-nodes, and oscillating in time.

We now adequately temper our claims throughout the manuscript to reinforce that the scenario of stochastic resonance is just an hypothetical scenario for spontaneous brain activity, to be further tested and demonstrated while considering brain geometry. This phenomenological model is now described in the new Results subsection 'Stochastic resonance of standing waves' (P16-18).

R2.15 The modeling does not provide insight. The model is phenomenological and its dynamics are presented with no fitting or quantitative comparison to data. No comparisons to alternatives are given, no predictions are made, and no clear claims are made of the model providing any insights/understanding. It is well known that under-damped oscillators driven by noise produce noisy, intermittent oscillatory dynamics.

A2.15: In this new version of the manuscript, we include a dedicated section termed 'Stochastic resonance of standing waves' which tests the hypothetical link between the well-

known phenomenology of *'under-damped oscillators driven by noise producing noisy, intermittent oscillatory dynamics'*, to the dynamics (i.e., temporal signature) of the principal components. To do so, we generate time series for each mode by numerically integrating the Stuart-Landau equation with a natural frequency fitted to the peak frequency of each mode and add gaussian white noise. Subsequently we multiply each (Tx1) simulated timeseries to the corresponding spatial mode (1xN), resulting in a NxT spatiotemporal pattern for each mode. We subsequently add the 10 spatiotemporal patterns together and report instantaneous snapshots of the 1xN patterns of activity. This serves to demonstrate that the proposed scenario is a valid explanation for the complex activity patterns observed in brain activity over time. However, this scenario does not exclude other hypothetical scenarios, and further experiments are needed to disambiguate between existing hypothetical mechanisms.

R2.16 P16: "Conflicting hypotheses have been..." this sentence should be expanded in the introduction to set a clear scene for what open questions this work addresses.

A2.16: Thank you, following the reviewer's comments we have now re-written the introduction to set a clear scene for the open questions addressed (in **bold** below).

New text in the Introduction:

*'Dynamically, correlated fluctuations in fMRI signals exhibit power at ultra-slow frequencies, peaking typically below 0.1 Hz in human brains at rest, although intrinsic functional networks have been detected at frequencies extending even beyond 0.5 Hz³⁻⁶. **Crucially, it remains unclear whether the spectral power at low frequencies is associated with aperiodic activations of the characteristically slow hemodynamic response function or instead reflects the existence of oscillatory phenomena with sustained periodicity over consecutive cycles^{7,8}. Given recent insights demonstrating that the fMRI signals underpinning intrinsic networks cannot be exclusively associated to the blood oxygenation level dependent (BOLD) signal^{9,10} and relate to macroscopic waves of activity¹¹⁻¹³, it is crucial to obtain a detailed characterization of the modes' temporal signatures to investigate their generative mechanisms.***

*Studies in rodents and humans have shown that ultra-slow (<0.5Hz) frequency components in fMRI signals have a periodic nature and are coupled with electrophysiological and electroencephalographic (EEG) signals¹⁴⁻¹⁸. These periodic fluctuations have been proposed to be linked to arteriole vibrations entrained by ultra-slow oscillations in local field potentials, pointing to a more direct relationship with the underlying neural activity^{19,20}. **Still, how these oscillations organize at the macroscopic level and their relationship to 'functional connectivity' between brain areas remains unclear.***

R2.17 Avoid the physical claim of "resonant modes" when 'resonance' has not been demonstrated.

A2.17: Following the reviewer's comment, we have removed the term 'resonance' from the title and replaced it with 'oscillatory modes' which is more objective. In addition, we now reinforce that 'resonance' is just a likely hypothetical mechanistic scenario to explain the origin of the macroscopic standing wave patterns oscillating transiently over time. We reinforce that computing the geometric eigenmodes of the brain to demonstrate that the patterns detected empirically match the ones predicted analytically is beyond the scope of this work and certainly deserves further investigation. Here our aim was to verify if the standing wave patterns detected (here and in previous works with different modalities) exhibit oscillatory signatures, which, to our knowledge, provides additional experimental evidence to support a mechanistic link with resonance phenomena.

While there is a long history of research linking the oscillations detected with electro- and magnetoencephalographic signals to a field theory of brain function (demonstrating that the oscillatory modes can be approximated by geometric eigenmodes such as spherical harmonics) these works refer exclusively to M/EEG signals, and the field theory has not been extended to the signals captured with fMRI. (i.e., works by Robinson and colleagues, Jirsa, Haken and colleagues, Nunez and colleagues, just to name a few groups).

We have revised the manuscript to reinforce that stochastic resonance is only a 'proposed' mechanism to explain the spontaneous emergence of such patterns (which we test using a phenomenological model).

Computing the geometric eigenmodes is certainly the missing evidence to demonstrate the link with resonance phenomena. We find it is important to note that the principal components are found to change shape across conditions (while the geometry of the brain remains unchanged). These results indicate that there are additional factors to take into consideration when computing the resonant modes of the brain, which may be linked to the boundary conditions and the resistivity of the medium through which the waves propagate. Considering these additional parameters may serve to obtain better analytic predictions of the modes detected empirically and explain why the patterns change across conditions.

REFS:

- Robinson, P. A. Physical brain connectomics. *Physical Review E* 99, 012421 (2019).
- Liang, Y. et al. Cortex-wide dynamics of intrinsic electrical activities: propagating waves and their interactions. *J. Neurosci.* (2021).
- Huntenburg, J. M., Yeow, L. Y., Mandino, F. & Grandjean, J. Gradients of functional connectivity in the mouse cortex reflect neocortical evolution. *NeuroImage* 225, 117528 (2021).

- Casorso, J. et al. Dynamic mode decomposition of resting-state and task fMRI. *NeuroImage* 194, 42–54 (2019).
- Shine, J. M. et al. Human cognition involves the dynamic integration of neural activity and neuromodulatory systems. *Nature Neuroscience* 20, 1–296 (2019).

Reviewer #3 (Remarks to the Author):

Cabral et al have analyzed ultra-fast single slice fMRI data from rats in three different states from sedation to full anesthesia. Along this gradient they find a reduction in the number of "modes" defined as the eigenvectors of the covariance matrix. They also projected the empirical data back onto a lower dimensional space defined by a small number of modes and demonstrate that these models are a good approximation of the data. Finally they conclude that these modes are standing waves that correspond to resonance phenomena in the brain which can be described by a subcritical Hopf oscillator with stochastic perturbation and they argue that it is these resonances that drive the formation of functional networks.

I read this manuscript with great interest and I want to congratulate the authors on their thorough work. It is definitely to their credit that the analysis was done completely transparently and that all scripts and data have been made available.

A: Thank you for the positive assessment of our manuscript!

R3.1. That said, I am still not entirely convinced that the authors have provided sufficient evidence for their main conclusion, namely that the brain operates on resonance phenomena which manifest as functional networks. The movies are certainly intriguing to watch, but on second thought they are also somewhat suspicious. I am no expert on rat anatomy but if I would see these large-scale wave patterns moving around in a human brain, my first thought would be "artifact". If the authors can convince me that what I am looking at is not just narrowly band-passed signals whose spatial PCA components, when projected back onto the data, show not entirely surprising narrow-band oscillations, this would greatly help my doubts.

A3.1: We agree with the reviewer that the previous version of the manuscript did not adequately demonstrate the evidence to support our claims, which led to some confusion in the assessment of the manuscript. In this new version of the manuscript we have modified the workflow and performed additional analysis to demonstrate that the results reported are solid and not due to artifacts nor to bandpass filtering or other preprocessing step. In addition, we now reinforce that the proposed mechanistic scenario for the origin of the oscillatory modes is hypothetical. The phenomenological model only tests the validity of our hypothesis, which remains valid until refuted by further tests.

To exclude the possibility of artifacts related to band-pass filtering, we now perform the PCA not within narrow frequency bands but across the broad frequency range where power in brain voxels was detected significantly above postmortem baseline, i.e., between 0.01 and 0.3 Hz. In all 3 conditions, we now apply exactly the same analysis pipeline to detect the

principal components with eigenvalue above postmortem baseline and evaluate the spectral content of their temporal signatures.

We now reinforce that our main finding is not the detection of spatial patterns exhibiting standing wave features, nor the detection of oscillations at the voxel level (both of which have been previously reported). Instead, our main findings are that 1) the principal components oscillate transiently in time; 2) the frequency and stability of these oscillations is modulated by anesthesia.

We have performed additional analysis to reassure the reader that the oscillations are actually present in the fMRI signals and are not induced by narrow band pass filtering. For instance, in the new Supplementary Figure S15 shown below, we took 3 representative spatial patterns (1xN) and multiply by the (NxT) broadband fMRI recordings to obtain the temporal signature by collapsing the N dimension. As can be seen below, the spatial patterns act like a ‘filter’ or ‘mask’ that scales the weights in each voxel such that the resulting time series reveals fine tuned oscillations that are not detected if the spatial patterns are randomized.

Fig. S15 - The oscillations are specific to the spatial patterns. To demonstrate that the oscillations are specific to the spatial wave patterns detected, we randomize the elements in 3 representative spatial patterns and demonstrate that, when projecting the randomized spatial patterns (with size 1xN) to the same, unfiltered, fMRI signals (with size NxT), no oscillations are detected in the associated temporal signature (with size 1xT), with the power being close to what is detected from a postmortem scan (red lines). The power spectra are shown in both linear and logarithmic (log) scales.

Regarding the reviewer's comment

'I am no expert on rat anatomy but if I would see these large-scale wave patterns moving around in a human brain, my first thought would be "artifact',

we wish to add that the spatial patterns detected herein overlap closely with the ones previously described in the rodent literature (we copy some examples below), making the link to what is typically referred to as 'resting-state networks' in rats. In additions, our controls showed no such artifact so there is very little reason to believe that these are artifactual waves.

Bajic, Dusica, et al. "Identifying rodent resting-state brain networks with independent component analysis." *Frontiers in Neuroscience* 11 (2017): 685.

<https://doi.org/10.3389/fnins.2017.00685>

Li, Q., Li, G., Wu, D., Lu, H., Hou, Z., Ross, C. A., ... & Duan, W. (2017). Resting-state functional MRI reveals altered brain connectivity and its correlation with motor dysfunction in a mouse model of Huntington's disease. *Scientific reports*, 7(1), 1-9.

Gutierrez-Barragan, Daniel, et al. "Infraslow state fluctuations govern spontaneous fMRI network dynamics." *Current Biology* 29.14 (2019): 2295-2306.
<https://www.sciencedirect.com/science/article/pii/S0960982219307109#ack0010>

Here follows a list of more detailed comments, page by page:

R3.2: p. 3: the authors argue that the ultra-fast resolution of 38ms was required to see the phenomena described here, yet they seem to be restricted to frequencies below 0.25Hz, ie a sampling rate of 0.5Hz (ie a TR of 2s). That's almost 2 orders of magnitude of oversampling. I can see that these concerns have already been partly addressed in SI Fig S12, but I didn't find any explanation why this happens.

A3.2: Thank you for raising this important point on the benefits of a sampling rate much faster than the Nyquist frequency (i.e., 26Hz vs 0.4Hz) for detecting ultra-slow oscillatory modes and in particular, for adequately characterizing their phase relationships and temporal signatures, while avoiding aliasing from higher frequency components.

Following the reviewer's concern, we have performed a new analysis, varying the sampling rate by downsampling the recorded data, to demonstrate that our results, despite being focused in the ultra-slow range <0.4Hz, strongly benefit from ultrafast acquisition sequences. This is now clearly illustrated in the new Figure 4 in the Manuscript and in the Supplementary Figures S10 to S14 (copied below).

By downsampling the data we found that the repertoire of modes detected, as well as the associated spectral power and frequency specificity, are strongly affected by the sampling rate. Indeed, already at 380 milliseconds, only 4 principal components (instead of 10) are

detected with eigenvalue above the postmortem baseline in medetomidine sedated rats (Figure S10).

Fig. S10 - Effect of the sampling rate on the number and spatial signature of the principal components detected. Before the analysis, the fMRI signals recorded with a TR of 38 milliseconds (ms) were downsampled by considering only one in every 5, 10, 15, 20 and 25 frames (corresponding to intervals of 190, 380, 570, 760 and 950 milliseconds between frames). The number of principal components detected corresponds to the number of eigenvalues larger than the largest eigenvalue from postmortem scans (also downsampled). As can be seen, although some of the components can still be detected for TRs up to 950ms, the ultrafast sampling rate used in this work reveals a larger number of components, clearly defined within cortical and subcortical structures.

We further investigated why sampling nearly 2 orders of magnitude faster than the Nyquist frequency is relevant in this context. Although the Nyquist theorem states that sampling at 2 points per oscillation is sufficient to detect an oscillation, if - as our data indicates - multiple oscillatory components are superposed and appear only transiently in time, operating near

the Nyquist frequency fails to capture the full spectrum of oscillatory components of a complex system.

Indeed, our data reveals that some of the modes detected at ultrafast sampling oscillate only for a few sustained cycles. This short-lived periodicity is particularly visible in the temporal signatures of a selected mode at ultrafast sampling rates (see the new Figure 4 below), but gradually becomes indistinguishable from aperiodic fluctuations as the sampling rate approaches the Nyquist frequency. This transiently sustained periodicity is also now more visible in the new Figures 3 to 5 and in the autocorrelation functions in the new Figure 6.

Figure 4 – Effect of the sampling rate in the detection of short-lived oscillations. The temporal signal $\tau_7^S(t)$ associated to the 7th principal component detected in ultrafast fMRI signals from medetomidine-sedated rats (Time of Repetition, $TR= 38$ milliseconds (ms)) is downsampled by considering only one in every 5, 10, 15, 20, 25, 30 and 35 frames (corresponding to intervals of 190, 380, 570, 760, 950, 1140 and 1330 ms between frames). Plots shown for 150 seconds from a representative

scan S (same from **Figure 3**). The power spectra of the temporal signals computed for the entire 10 minutes recorded for scan S (blue) and for a scan performed postmortem (red).

Below we copy the relevant text insertions in the manuscript in the Introduction and Results:

New text in the Introduction:

'The precise characterization of oscillations across space and over time is complex and benefits from an adequate spatiotemporal resolution and high signal-to-noise (SNR) ratio to adequately capture transient phase-relationships between voxels. Ultra-high field fMRI studies in rats achieve increased SNR by attenuating thermal noise using cryogenic coils^{1,2}, allowing for increased precision in the characterization of oscillatory signals.'

New text in the Results section:

'Importantly, the detection of transiently sustained periodicity benefits from sampling rates significantly faster than the Nyquist frequency, to ensure that sufficient points are captured at the oscillation frequency and reveal a distinctive peak in the power spectrum. As shown in Figure 4, a transient oscillation detected at 0.22Hz in the temporal signature of a given principal component, $\tau_j^S(t)$, becomes indistinguishable from aperiodic fluctuations as the sampling approximates the Nyquist frequency (corresponding to 2.27 seconds for 0.22 Hz oscillations). In Supplementary figures S10-S14 we further demonstrate how the detection of oscillatory modes benefits from ultrafast sampling rates.'

Furthermore, we note that aliasing of physiological noise into the conventionally sampled data is certainly one of the reasons why it is more difficult to observe the 0.1-0.4 Hz peaks in the TR=2s downsampling. With this high sampling rate we were able to resolve frequency components up to 13 Hz, which prevents major physiological noises from aliasing with the signal of interest. Having said that, we found that a TR of around 150-200ms is already sufficient for observing most of the oscillatory modes in the brain, which is promising for future applications of the technique in e.g. human samples.

R3.3: p. 6: about movie S1 the authors write "sustained oscillations emerge consistently" but given that the data have been bandpassed severely to 0.2-0.25Hz this isn't surprising, it's expected. It's still interesting to see the spatial pattern of course.

A3.3: Thank you for the comment, we have corrected the text. We now more clearly state that the surprise when watching the video is the organization in space, rather than the actual frequency content itself. Furthermore, given that we now extend the frequency range of the video from 0.01 to 0.3 Hz this effect is perhaps a bit more "surprising".

New text in the Results section:

'While the space-frequency analysis provides information about which voxels have more power in each frequency band, it does not reveal how the signals evolve in time or

organize in space. As can be seen in **Video 1**, fluctuations are not globally correlated, but instead exhibit complex phase relationships across space that appear recurrent over time and consistent across different rats in the same condition. Signals co-varying in phase across distant voxels symmetrically aligned with respect to the vertical midline point to a link with long-range functional connectivity. In deep anesthesia, despite applying exactly the same filtering, no particular spatial organization or fine structure is detected except for ultra-slow globally correlated fluctuations.'

Video 1 (still image)– fMRI signals band-pass filtered between 0.01 and 0.3 Hz in 3 different rats and in 3 different conditions (Sedation: Medetomidine only; Light Anesthesia: Medetomidine + 1% isoflurane; Deep anesthesia: Medetomidine + 3% isoflurane). To account for expected differences in power across conditions, colorbar limits are set to ± 4 standard deviations of the band-pass filtered signals in each scan.

R3.4: p. 6, bottom: I think it's crucial to point out here, that all "modes" very defined on a very narrowly band-passed signal, because this might get lost a bit further down when modes are just labeled through ψ_1 , ψ_2 etc (in descending order of eigenvalue magnitude I believe, and irrespective of condition?).

A3.4: We agree with the reviewer that the approach to define the modes was not adequate. We now define a separate set of modes in each of the 3 conditions detected across the whole range of frequencies where significant power was detected, i.e. between 0.01 and 0.3 Hz (shown in the new Figures 3 and 5). Across conditions, only the modes with eigenvalue above postmortem baseline are considered, resulting in 10 modes under medetomidine, 6 modes when adding isoflurane at 1% and only 1 mode with isoflurane at 3%.

R3.5: p. 7, bottom: in the equation defining tau it would be nice if the notation could indicate that this is a matrix multiplication that contracts the "n" dimension

A3.5: Thank you, we agree and have revised accordingly.

Modified text in the Results section:

'The temporal signature $\tau_{\alpha}^S(t)$ of each principal component α for each scan S is obtained by performing a matrix multiplication that contracts the 'n' dimension as: $\tau_{\alpha}^S(t) = \psi_{\alpha}(n) \Psi^S(n, t)$, where $\psi_{\alpha}(n)$ represents the spatial pattern of each principal component α and $\Psi^S(n, t)$ represents the activity recorded with fMRI across all voxels n and timepoints t for scan S (Figure 3).'

R3.6: p. 8, Fig 2C-D: I prefer power-spectra to be semi-log (or log-log) but more importantly I don't completely understand panel C-D. Surely mode 2 shown here was defined in the sedated condition and in frequency band 0.2 which is why it shows such a narrow spectrum by construction? How about showing the mode that was defined in the same frequency band but in the respective condition for all three?

A3.6: Thank you, we agree that the previous analysis was not adequate and we now obtain the principal components in the broad frequency range of interest (0.01-0.3Hz) across scans in each condition. We completely changed Fig 2, to the new Fig 3 and Fig 5.

Regarding the suggestion to report the power spectra in log scales, we note that in this case the resonant peaks were so prominent that they were visible even in linear scale. We now report in the new Figure S15 the power spectra in both linear and semi-log scales, to illustrate our argument (see below).

Fig. S15 - The oscillations are specific to the spatial patterns. To demonstrate that the oscillations are specific to the spatial wave patterns detected, we randomize the elements in 3 representative spatial patterns and demonstrate that, when projecting the randomized spatial patterns (with size $1 \times N$) to the same, unfiltered, fMRI signals (with size $N \times T$), no oscillations are detected in the associated temporal signature (with size $1 \times T$), with the power being close to what is detected from a postmortem scan (red lines). The power spectra are shown in both linear and logarithmic (log) scales.

R3.7: p. 8, Fig 2F-G: the spectrum in panel F and the Q values in panel G don't seem to always correspond, ie for mode ψ_1 the Q value seems much higher for light anesthesia than for sedation, yet the bar plot shows them to be equal

A3.7: As the reviewer will find, we completely re-analyzed the data and generated more clear figures. The Q-factors in the new Figure 6b (copied below) are now reported for each single scan and refer to the temporal signature associated to the condition-specific spatial patterns detected across scans, as illustrated by the patterns in the X ticks.

Figure 6 – Principal components oscillate at higher frequencies and with less damping under medetomidine. (a,b) The temporal signatures associated to the principal components detected in each condition are characterized in terms of peak frequency and Q-factor for each of the 12 scans in each condition (2 scans per rat per condition). Error bars represent the mean \pm standard error across scans. **(c)** To illustrate the stability of the oscillations, the autocorrelation functions of the temporal signatures τ_1^S associated to the first principal component in each condition are reported. Examples are shown for 3 scans from the same rat and from a postmortem scan. As can be seen, the autocorrelation function under medetomidine exhibits 3 oscillations before the amplitude decays to $1/e$ (~37%), 2 cycles after adding isoflurane at 1% and no complete cycle under deep anesthesia, similar to what is observed in the postmortem scan.

R3.8: p. 10: the narrow-band activity in this movie is very intriguing and clearly the low-dimensional model with separated variables "n" and "t" captures it well. But what are we to make of these large waves traveling up the whole length of the lateral rat cortex?

A3.8: Thank you for this comment. We agree that these traveling waves are indeed very intriguing, but fully aligned with recent reports of slowly propagating traveling waves detected in resting-state fMRI studies both in rodents and humans. In fact, the same patterns of traveling waves have been previously described through the framework of Quasi-Periodic Patterns (QPPs) (Thompson et al., 2014, we invite the reviewer to see a 40 second video extract of a presentation given by the senior author Shella Keilholz at the Wellcome Center for Integrative neuroimaging on the 13/01/2021 where we clearly see the same 'large waves traveling up the whole length of the lateral rat cortex' (please click <https://shorturl.at/jkpFX>).

Despite operating at lower spatiotemporal resolution, this work shows that these waves are found to correlate with electrical activity.

More recently, Raut and colleagues (2021) described *'using fMRI in humans, we show that ongoing arousal fluctuations are associated with global waves of activity that slowly propagate in parallel throughout the neocortex, thalamus, striatum, and cerebellum. We show that these waves can parsimoniously account for many features of spontaneous fMRI signal fluctuations, including topographically organized functional connectivity.'*

Our work extends on these previous findings by demonstrating that these traveling waves can result from the superposition of a reduced basis of standing waves oscillating transiently over time at fine-tuned frequencies. We now point to these previous works more clearly and reinforce the need to further investigate these wave patterns at high temporal resolution in future research.

Raut, R. V., Snyder, A. Z., Mitra, A., Yellin, D., Fujii, N., Malach, R., & Raichle, M. E. (2021). Global waves synchronize the brain's functional systems with fluctuating arousal. *Science advances*, 7(30), eabf2709.

Thompson, G. J., Pan, W. J., Magnuson, M. E., Jaeger, D., & Keilholz, S. D. (2014). Quasi-periodic patterns (QPP): large-scale dynamics in resting state fMRI that correlate with local infraslow electrical activity. *Neuroimage*, 84, 1018-1031.

R3.9: p. 11, Fig 3: overlaying the squared amplitude here doesn't add much info, it's more distracting than helpful, especially since it even intersects with the timeseries. Either drop altogether or offset from timeseries to make it clearer.

A3.9: Thanks for the comment. As suggested, we dropped the amplitude altogether in new figures.

R3.10: p. 13, Hopf equation: should be $|Z|^2$ rather than $|Z^2|$. Since in the sentence right after the equation "external perturbations" are referred to, perhaps the perturbation term should be included here right away (as indeed it is in the SI)

A3.10: Thank you for catching this. We now reformulated the description of the phenomenological model and now include the noise perturbation directly in the equation. We also generated a new figure 7 (copied below) to illustrate the model.

New text in the Results section:

To demonstrate that the stochastic resonance of stationary wave patterns can generate the patterns of intrinsic functional connectivity observed experimentally, we model the signals in the brain slice as the superposition (i.e., linear sum) of modes whose spatial configuration $\psi_\alpha(n)$ is fixed and given by the principal components detected empirically, and the temporal signature $Z_\alpha(t)$ is obtained using the Stuart-Landau equation to simulate the behavior of an oscillator in the underdamped regime in the presence of background noise as:

$$\Psi^{Model}(n, t) = \sum_{\alpha} \psi_{\alpha}(n) Z_{\alpha}(t),$$

with

$$dZ_{\alpha}/dt = Z_{\alpha}(i\omega_{\alpha} - |Z_{\alpha}|^2 + a) + \beta\eta,$$

where ω_{α} is the resonant frequency of each mode, a (negative) scales the decay rate and η is the added gaussian white noise η with standard deviation β .

Figure 7 – Stochastic resonance of standing waves drives transient long range correlations in simulated signals. The spatial configurations and temporal signatures of the principal components align with the hypothesis that they represent standing waves, whose phenomenology is inherently associated with resonance phenomena. To model the dynamics emerging from the transient resonance of standing waves in the presence of background noise, we simulate a temporal signature for each of the spatial patterns detected in medetomidine sedated rats (a) as the behavior of an underdamped oscillator perturbed with gaussian white noise, with natural frequency fitted to the peak frequency obtained from one representative scan, and fitting the standard deviation to the temporal signatures of the same scan (b). Multiplying the $T \times 1$ temporal signatures by the corresponding $1 \times N$ spatial patterns and summing across modes results in a $N \times T$ spatiotemporal pattern representing the result from the stochastic resonance of a repertoire of standing waves.

R3.11: p. 18, first paragraph: I think this discussion fails to mention that the timescale of the rat HRF is substantially lower than in humans (Lambers 2019, doi:10.1016/j.neuroimage.2019.116446) and is in fact right on top of the 0.25Hz frequency band where the strongest effects are found. So a devil's advocate could easily claim that what is resonating here is the brain vasculature when perturbed by true neuronal activity ...

A3.11: We thank the reviewer for pointing out this relevant work, which deserved our dedicated attention. Indeed, the authors report a faster HRF in rats but do not mention the detection of sustained periodicity (oscillations) so it is not straightforward to relate this to our results. Importantly, the authors note in the conclusion:

'Strictly, our cortical rat HRF is valid only for the range of experimental conditions tested, which were male or female Fischer or SD rats, mechanical or electrical paw stimulation (both innocuous and noxious) or optogenetic stimulation, using frequencies from 5 Hz up to 12 Hz and durations of 5 s or 10 s, under medetomidine anesthesia or medetomidine supplemented with 0.7 % isoflurane, using ventilation or spontaneous breathing animals.' (Lambers et al., 2019)

Although we did not use any stimulation and focused solely on the resting state, it has been proposed that the fluctuations during rest are associated with a stochastic activation of HRF responses (in which case they would be aperiodic).

However, our new analysis reveals that the temporal signatures exhibit sustained periodicity over consecutive cycles. This is made particularly clear in the autocorrelation functions shown in the new Figure 6c (copied below).

Recently, Drew and colleagues have addressed the origin of 'locally detected' periodicity in fMRI signals and proposed these were linked to arteriole vibrations entrained by ultra-slow oscillations in local field potentials, pointing to a more direct relationship with the underlying neural activity. However, this work does not explain how the oscillations organize at the macroscopic scale.

P. J. Drew, C. Mateo, K. L. Turner, X. Yu, D. Kleinfeld, Ultra-slow oscillations in fMRI and Resting-State connectivity: neuronal and vascular contributions and technical confounds. *Neuron*, (2020).

New text in the Introduction

'Dynamically, correlated fluctuations in fMRI signals exhibit power at ultra-slow frequencies, peaking typically below 0.1 Hz in human brains at rest, although intrinsic functional networks have been detected at frequencies extending even beyond 0.5 Hz (Lee et al., 2013; Chen and Glover, 2015; Trapp et al., 2018; Vohryzek et al., 2020). Crucially, it remains unclear whether the power at low frequencies is associated with aperiodic activations of the characteristically slow hemodynamic response function or instead reflects the existence of oscillatory phenomena with sustained periodicity over consecutive cycles (Casorso et al., 2019). Given recent insights demonstrating that the fMRI signals underpinning intrinsic networks cannot be exclusively associated to the blood oxygenation level dependent (BOLD) signal (Liu, 2016;; Chen et al., 2020) and exhibit macroscopic wave-like phenomena (Raut et al., 2021), it is crucial to obtain a detailed characterization of the modes' temporal signatures to investigate their generative mechanisms.

Studies in rodents have shown that ultra-slow (<0.5Hz) frequency components in fMRI signals have a periodic nature and are coupled with electrophysiological and electroencephalographic (EEG) signals (19-23). These periodic fluctuations have been proposed to be linked to arteriole vibrations entrained by ultra-slow oscillations in local field potentials, pointing to a more direct relationship with the underlying neural activity (18). Still,

how these oscillations organize at the macroscopic level and their relationship to ‘functional connectivity’ between brain areas remains unclear.

Figure 6 – Principal components oscillate at higher frequencies and with less damping under medetomidine. (a,b) The temporal signatures associated to the principal components detected in each condition are characterized in terms of peak frequency and Q-factor for each of the 12 scans in each condition (2 scans per rat per condition). Error bars represent the mean \pm standard error across scans. **(c)** To illustrate the stability of the oscillations, the autocorrelation functions of the temporal signatures τ_1^S associated to the first principal component in each condition are reported. Examples are shown for 3 scans from the same rat and from a postmortem scan. As can be seen, the autocorrelation function under medetomidine exhibits 3 oscillations before the amplitude decays to $1/e$ ($\sim 37\%$), 2 cycles after adding isoflurane at 1% and no complete cycle under deep anesthesia, similar to what is observed in the postmortem scan.

Minor comments:

R3.12: SI page 4, bottom: again $|Z|^2$. Also the sentence following the equation is rather unclear (and missing a bracket).

A3.12: Thanks for noticing, corrected.

R3.13: SI page 6: "Error! Reference source not found."

A3.13: Thanks for noticing, corrected.

R3.14: SI Fig 12: I don't understand why this is happening since the cross-over at around $TR=0.2s$ (Nyquist 5/2 Hz) is still far away from the 0.2Hz frequency band (Nyquist 5/2 s)

A3.14: We agree with the reviewer that this was intriguing and questions the validity of the Nyquist theorem. Our new analysis revealed that the oscillations were less clearly detected at sampling rates far higher than the Nyquist frequency essentially because the oscillations are transient and short-lived, so the analysis benefits from high sampling rates in order to detect more points per oscillation.

Given the importance of these findings for the fMRI community in general, we have included a new Figure 4 in the manuscript to illustrate why sampling rates as fast as 380 ms are insufficient to detect oscillatory modes oscillating at 0.22 Hz (with a Nyquist frequency close to 2 seconds).

Figure 4 – Effect of the sampling rate in the detection of short-lived oscillations. The temporal signal $\tau_7^S(t)$ associated to the 7th principal component detected in ultrafast fMRI signals from medetomidine-sedated rats (Time of Repetition, $TR= 38$ milliseconds (ms)) is downsampled by considering only one in every 5, 10, 15, 20, 25, 30 and 35 frames (corresponding to intervals of 190, 380, 570, 760, 950, 1140 and 1330 ms between frames). Plots shown for 150 seconds from a representative scan S (same from Figure 3). The power spectra of the temporal signals computed for the entire 10 minutes recorded for scan S (blue) and for a scan performed postmortem (red).

References

- 1 Ratering, D., Baltes, C., Nordmeyer-Massner, J., Marek, D. & Rudin, M. Performance of a 200-MHz cryogenic RF probe designed for MRI and MRS of the murine brain. *Magnetic Resonance in Medicine: An Official Journal of the International Society for Magnetic Resonance in Medicine* **59**, 1440-1447 (2008).
- 2 Arbabi, A. *et al.* Multiple-mouse magnetic resonance imaging with cryogenic radiofrequency probes for evaluation of brain development. *NeuroImage* **252**, 119008 (2022).
- 3 Lee, H.-L., Zahneisen, B., Hugger, T., LeVan, P. & Hennig, J. Tracking dynamic resting-state networks at higher frequencies using MR-encephalography. *NeuroImage* **65**, 216-222 (2013).
- 4 Chen, J. E. & Glover, G. H. BOLD fractional contribution to resting-state functional connectivity above 0.1 Hz. *NeuroImage* **107**, 207-218 (2015).
- 5 Trapp, C., Vakamudi, K. & Posse, S. On the detection of high frequency correlations in resting state fMRI. *NeuroImage* **164**, 202-213 (2018).
- 6 Vohryzek, J., Deco, G., Cessac, B., Kringelbach, M. L. & Cabral, J. Ghost attractors in spontaneous brain activity: Recurrent excursions into functionally-relevant BOLD phase-locking states. *Frontiers in systems neuroscience* **14**, 20 (2020).
- 7 Cabral, J., Kringelbach, M. L. & Deco, G. Functional connectivity dynamically evolves on multiple time-scales over a static structural connectome: Models and mechanisms. *NeuroImage*, doi:10.1016/j.neuroimage.2017.03.045 (2017).
- 8 Casorso, J. *et al.* Dynamic mode decomposition of resting-state and task fMRI. *NeuroImage* **194**, 42-54 (2019).
- 9 Chen, J. E. *et al.* Resting-state “physiological networks”. *NeuroImage* **213**, 116707 (2020).
- 10 Liu, T. T. Noise contributions to the fMRI signal: An overview. *NeuroImage* **143**, 141-151 (2016).
- 11 Gu, Y. *et al.* Brain activity fluctuations propagate as waves traversing the cortical hierarchy. *Cerebral cortex* **31**, 3986-4005 (2021).
- 12 Raut, R. V. *et al.* Global waves synchronize the brain’s functional systems with fluctuating arousal. *Science advances* **7**, eabf2709 (2021).
- 13 Schwalm, M. *et al.* Cortex-wide BOLD fMRI activity reflects locally-recorded slow oscillation-associated calcium waves. *eLife* **6**, e27602 (2017).
- 14 Pan, W.-J., Thompson, G. J., Magnuson, M. E., Jaeger, D. & Keilholz, S. Infraslow LFP correlates to resting-state fMRI BOLD signals. *NeuroImage* **74**, 288-297 (2013).
- 15 Thompson, G. J., Pan, W.-J., Magnuson, M. E., Jaeger, D. & Keilholz, S. D. Quasi-periodic patterns (QPP): large-scale dynamics in resting state fMRI that correlate with local infraslow electrical activity. *NeuroImage* **84**, 1018-1031 (2014).
- 16 He, B. J., Snyder, A. Z., Zempel, J. M., Smyth, M. D. & Raichle, M. E. Electrophysiological correlates of the brain's intrinsic large-scale functional architecture. *Proceedings of the National Academy of Sciences of the United States of America* **105**, 16039-16044, doi:10.1073/pnas.0807010105 (2008).
- 17 Lewis, L. D., Setsompop, K., Rosen, B. R. & Polimeni, J. R. Fast fMRI can detect oscillatory neural activity in humans. *Proceedings of the national academy of sciences* **113**, E6679-E6685 (2016).

- 18 Fultz, N. E. *et al.* Coupled electrophysiological, hemodynamic, and cerebrospinal fluid oscillations in human sleep. *Science* **366**, 628-631 (2019).
- 19 Drew, P. J., Mateo, C., Turner, K. L., Yu, X. & Kleinfeld, D. Ultra-slow oscillations in fMRI and resting-state connectivity: neuronal and vascular contributions and technical confounds. *Neuron* **107**, 782-804 (2020).
- 20 Mateo, C., Knutsen, P. M., Tsai, P. S., Shih, A. Y. & Kleinfeld, D. Entrainment of arteriole vasomotor fluctuations by neural activity is a basis of blood-oxygenation-level-dependent “resting-state” connectivity. *Neuron* **96**, 936-948. e933 (2017).
- 21 van Alst, T. M. *et al.* Anesthesia differentially modulates neuronal and vascular contributions to the BOLD signal. *NeuroImage* **195**, 89-103 (2019).
- 22 Paasonen, J., Stenroos, P., Salo, R. A., Kiviniemi, V. & Gröhn, O. Functional connectivity under six anesthesia protocols and the awake condition in rat brain. *NeuroImage* **172**, 9-20 (2018).
- 23 Weber, R., Ramos-Cabrera, P., Wiedermann, D., Van Camp, N. & Hoehn, M. A fully noninvasive and robust experimental protocol for longitudinal fMRI studies in the rat. *NeuroImage* **29**, 1303-1310 (2006).
- 24 Gutierrez-Barragan, D., Basson, M. A., Panzeri, S. & Gozzi, A. Infralow state fluctuations govern spontaneous fMRI network dynamics. *Current Biology* **29**, 2295-2306. e2295 (2019).
- 25 Nasrallah, F. A., Tay, H.-C. & Chuang, K.-H. Detection of functional connectivity in the resting mouse brain. *NeuroImage* **86**, 417-424 (2014).
- 26 Pradier, B. *et al.* Combined resting state-fMRI and calcium recordings show stable brain states for task-induced fMRI in mice under combined ISO/MED anesthesia. *NeuroImage* **245**, 118626 (2021).
- 27 Grandjean, J., Schroeter, A., Batata, I. & Rudin, M. Optimization of anesthesia protocol for resting-state fMRI in mice based on differential effects of anesthetics on functional connectivity patterns. *NeuroImage* **102**, 838-847 (2014).
- 28 Damoiseaux, J. S. *et al.* Consistent resting-state networks across healthy subjects. *Proceedings of the National Academy of Sciences of the United States of America* **103**, 13848-13853, doi:10.1073/pnas.0601417103 (2006).
- 29 Smith, S. M. *et al.* Correspondence of the brain's functional architecture during activation and rest. *Proceedings of the national academy of sciences* **106**, 13040-13045 (2009).
- 30 Liu, X. & Duyn, J. H. Time-varying functional network information extracted from brief instances of spontaneous brain activity. *Proceedings of the National Academy of Sciences of the United States of America* **110**, 4392-4397, doi:10.1073/pnas.1216856110 (2013).
- 31 Eickhoff, S. B. *et al.* Co-activation patterns distinguish cortical modules, their connectivity and functional differentiation. *NeuroImage* **57**, 938-949 (2011).
- 32 Margulies, D. S. *et al.* Situating the default-mode network along a principal gradient of macroscale cortical organization. *Proceedings of the National Academy of Sciences* **113**, 12574-12579 (2016).
- 33 Huntenburg, J. M., Bazin, P.-L. & Margulies, D. S. Large-scale gradients in human cortical organization. *Trends in cognitive sciences* **22**, 21-31 (2018).
- 34 Cabral, J. *et al.* Cognitive performance in healthy older adults relates to spontaneous switching between states of functional connectivity during rest. *Scientific reports* **7**, 5135, doi:10.1038/s41598-017-05425-7 (2017).

- 35 Lord, L.-D. *et al.* Dynamical exploration of the repertoire of brain networks at rest is modulated by psilocybin. *NeuroImage* **199**, 127-142 (2019).
- 36 Uddin, L. Q., Yeo, B. & Spreng, R. N. Towards a universal taxonomy of macro-scale functional human brain networks. *Brain topography* **32**, 926-942 (2019).
- 37 Calhoun, V. D., Miller, R., Pearlson, G. & Adali, T. The chronnectome: time-varying connectivity networks as the next frontier in fMRI data discovery. *Neuron* **84**, 262-274, doi:10.1016/j.neuron.2014.10.015 (2014).
- 38 Preti, M. G., Bolton, T. A. & Van De Ville, D. The dynamic functional connectome: State-of-the-art and perspectives. *NeuroImage* **160**, 41-54 (2017).
- 39 Coletta, L. *et al.* Network structure of the mouse brain connectome with voxel resolution. *Science Advances* **6**, eabb7187 (2020).
- 40 Fulcher, B. D., Murray, J. D., Zerbi, V. & Wang, X.-J. Multimodal gradients across mouse cortex. *Proceedings of the National Academy of Sciences* **116**, 4689-4695 (2019).
- 41 Lu, H. *et al.* Rat brains also have a default mode network. *Proceedings of the National Academy of Sciences* **109**, 3979-3984 (2012).
- 42 Hutchison, R. M. *et al.* Resting-state networks in the macaque at 7 T. *NeuroImage* **56**, 1546-1555 (2011).

Reviewer #1 (Remarks to the Author):

In this submitted revision and rebuttal letter, the authors have put substantial efforts into addressing my previous comments. However, some of the concerns have not been convincingly addressed, as detailed below:

Previous comment R1.1 – the necessity of employing ultra-short TR: In the revision, the authors down-sampled the fMRI data at different levels to simulate the influence of TR on the specificity of identified oscillatory modes. It is worth noting that downsampling with a fixed scan duration will lead to reduced data points, which may also contribute to reduced frequency specificity and the # of identified eigenmodes, particularly given that the total duration of scan time was only 150 s (Fig. 4). It would be helpful if the authors could perform the comparison across sampling TRs with comparable # of time points for each condition. Additionally, as I stated in the previous comment, fast sampling at a TR of 38 ms can incur a substantial penalty on the fMRI spatial resolution, which may also affect the specificity of derived oscillatory modes. The authors should be cautious about this recommendation.

Previous comment R1.3 – generalizability of the study findings to awake humans: As acknowledged by the authors, the major resonant peaks at 0.2/0.25 Hz in the rodent data were induced by medetomidine, further rendering it unconvincing that the identified oscillatory modes can be generalized to awake human data without the anesthetic drugs. Also, I may not agree with the authors that spontaneous activity in awake humans is periodic, based on reports of extensive literatures including those cited by the authors. Therefore, the first paragraph of the new text in Introduction should be revised.

Previous comment R1.4 – influence of hemodynamic blurring: While in the revision, the authors acknowledged the influence of hemodynamic blurring on the oscillatory modes, this concern has not been addressed. Given that the hemodynamic response functions are highly variable across the cortex (e.g., Gonzalez-Castillo, J., Saad, Z.S., Handwerker, D.A., Inati, S.J., Brenowitz, N. and Bandettini, P.A., 2012. Whole-brain, time-locked activation with simple tasks revealed using massive averaging and model-free analysis. Proceedings of the National Academy of Sciences, 109(14), pp.5487-5492.) and obscure local spectral properties, it poses a concern regarding to which extent the identified oscillatory modes arise from neuronal origins.

Reviewer #2 (Remarks to the Author):

The authors have responded clearly, thoughtfully, and substantially to my comments. The manuscript has fixed many inaccuracies in conceptual understanding (around the extracted components being 'based on wave physics') and terminology (now referring to the covariance modes as principal components). The claiming of novelty around parts of the study that are actually novel, and setting up of the unique problems that their analysis sheds light on (e.g., in Abstract and through the Introduction) is also much improved (although I have a few further queries on this below). It is clear from the updated results (Fig. 4) that my suggestion (mirroring other reviewers) that oscillations were accentuated by the application of narrow band-pass filtering is not correct—they are indeed very clear in the broadband signals (which are now plotted). Overall it is a major improvement: the structure is logical, clear, and correct, and the results are interesting.

I have a smaller number of remaining responses. First to the reviewer responses, and second from a reading of the updated manuscript:

---REVIEWER RESPONSE---

A2.2: While I agree the spatial maps have received more attention than temporal properties, there is still substantial work involving modal decompositions that have characterized their oscillatory properties, including interpreting the properties of their power spectra (from neural field theory treatments, and a comparison to PCA), and damping times and oscillatory periods from dynamic mode decomposition (including a few examples from neural field theory, listed below). The text could be rephrased to more fairly cover prior work that has performed detailed characterizations of the oscillatory temporal properties of spatial modes (and clarifying the unique parts of this work e.g., phase relationships). The novelty of this study could then be refined given that NFT modes are associated with a power spectrum and DMD modes are constructed (as \sim rotations of PCA modes) to have coherent standing-wave oscillations.

A major comment comes from this—this study looks at the temporal properties of PCA modes directly (which do not necessarily have to exhibit oscillations), and shows they indeed have interesting oscillatory properties... But DMD already solves this problem of 'rotating' the PCA modes to yield standing-wave modes, so if they are of interest, why not study the DMD modes directly? This concern recurs in the authors response in A2.14—unclear why the authors perform PCA (which doesn't extract resonance/standing-wave modes) instead of DMD (which does). A clear explanation/justification for why PCA is chosen instead of DMD is required.

- Gabay NC, Babaie-Janvier T, Robinson PA. Dynamics of cortical activity eigenmodes including standing, traveling, and rotating waves. Physical Review E. 2018 Oct;98(4):042413.

- Mukta KN, MacLaurin JN, Robinson PA. Theory of corticothalamic brain activity in a spherical geometry: Spectra, coherence, and correlation. Physical Review E. 2017 Nov;96(5):052410.

A2.3: The claim that the authors can "expose the missing characterization of their temporal signatures", seems to overlook prior work on this (above) and should be revised.

A major comment: A complete spatial randomization (Fig. S15) seems inappropriate: this is too extreme a null model. The data are so heavily spatially embedded (and thus strongly spatially autocorrelated) that, after shuffling them in a completely unconstrained way, it's unsurprising that it also destroys the temporal structure. Would spatially constrained nulls be more appropriate—holding generic spatial autocorrelation fixed, are these particular spatial patterns surprising?

cf. <https://www.biorxiv.org/content/10.1101/2020.08.13.249797v1>

A2.17: Authors claim modal decompositions have not been applied to fMRI. This has indeed been done. E.g., should incorporate Henderson et al. (2022) as an example of an application of spatiotemporal modal decomposition to fMRI, and how the predictions of neural field theory relate to PCA modes. Given the conceptual similarities to the current paper, perhaps greater discussion of it (and/or similar work) is warranted [optional].

- Henderson JA, Aquino KM, Robinson PA. Empirical estimation of the eigenmodes of macroscale cortical dynamics: Reconciling neural field eigenmodes and resting-state networks. Neuroimage: Reports. 2022 Sep 1;2(3):100103.

---TEXT---

- "the fundamental self-organizing principles" Consistent with the previous manuscript (in using words/phrases to hype/obscure a clear description of what is done) this phrase recurs through the manuscript. The use of the word 'self-organizing' should be made clearer. I would favor 'organizing' over 'self-organizing' if the authors do not

provide a model of self-organization, or if the connections to the field of self-organization cannot be made concrete (e.g., self-organization of larger structures from individual molecules in chemistry, or the concept of self-organized criticality in complexity theory, etc.).

- Typo in abstract: "was found to modulate by anesthesia".
- Rephrase: "Dynamically, correlated" (P3) ["dynamically fluctuations" doesn't make sense].
- Check all axes in plots—many are not labeled (e.g., vertical axes of 3e,f; 4); units should also be provided if possible.
- [optional] Carpet plots seem to contain interesting structure that is hard to visualize. Could consider (i) using a clearer color map to distinguish positive (red) from negative (blue) deflections; and/or (ii) reordering rows to reveal structure, as per Aquino et al. *NeuroImage*. 2020; 212:116614.
- "1/e" is clearer than " e^{-1} " when using a superscript reference format.
- Should clearly define the autocorrelation function—most will not be used to a definition of autocorrelation that involves both real and imaginary components. In many cases in text and in caption, the typical usage is used (and shown in upper of Fig. 6c), but the plots in lower panels of 6c should be better explained/interpreted/labeled (interpreting sinks from Hilbert transform—it is also unclear what this representation provides in addition to the conventional autocorrelation function).
- Some visualization error appears to have occurred in the rendering of Figure 8b—parentheses are all empty.
- Fig. 4: Should use mathematical notation for multiplication (x) rather than computer-code notation (*).

Reviewer #3 (Remarks to the Author):

I believe the manuscript has much benefited by addressing the concerns of the reviewers. While I am still not 100% convinced by the claims made by the authors, considering the massive amount of work that has clearly gone into this analysis, I think it is fair to present it to the readers at this point and let them judge for themselves.

Response to Reviewers

We'd like to again thank the Reviewers, most sincerely, for their time and effort and their valuable comments, which have again improved and strengthened our work. We appreciate it and reply below to every comment raised in **blue font**. Please note that in some cases we have broken down long comments to ensure that we replied to every aspect of the comment in full.

Reviewer #1

In this submitted revision and rebuttal letter, the authors have put substantial efforts into addressing my previous comments.

Author Reply: Thank you, we really appreciate the recognition of our efforts to address the reviewers' comments. Please see below our comments to the remaining issues:

However, some of the concerns have not been convincingly addressed, as detailed below:

Previous comment R1.1 – the necessity of employing ultra-short TR: In the revision, the authors down-sampled the fMRI data at different levels to simulate the influence of TR on the specificity of identified oscillatory modes. It is worth noting that downsampling with a fixed scan duration will lead to reduced data points, which may also contribute to reduced frequency specificity and the # of identified eigenmodes, particularly given that the total duration of scan time was only 150 s (Fig. 4). It would be helpful if the authors could perform the comparison across sampling TRs with comparable # of time points for each condition.

Author Reply: Thank you for raising this critique and we apologize that we have not been sufficiently clear in our previous response.

First - we completely agree with the reviewer that the recommendation of employing an ultra-short TR needs to be well justified and contextualized. Therefore, following the reviewer's comment, we have performed an even more detailed analysis and now better explain the main advantages of having a short TR, which, beyond increasing the frequency specificity given a fixed scan duration, improves the precision of the analysis by preventing frequency aliasing from undersampled oscillatory components in the signals.

We also wish to clarify that, although the signals were *shown* only for 150 seconds in Figure 4 for illustration, the power spectra were analyzed over the entire duration of the scans (i.e., 10 minutes or 600 seconds) – and therefore we did not suffer from a limited time window (we apologize again that this was not made sufficiently clear in the previous version). We now

explain this clearly, and increase the interval shown in Figure 4 to 250 seconds so that this is clearer.

Furthermore, following the Reviewer's suggestion, we now perform a more complete comparison of power spectra arising from the original and a signal downsampled at $10 \times TR$ (380ms), but now considering the same number of time points, i.e. 1551 time points (please see the new supplementary Figure copied below). Importantly, when considering the same number of time points, the total duration of the signals will differ by the same factor of 10, corresponding to 59 seconds for a $TR=38ms$ and 590 seconds for a TR of 380 ms, which of course affects the power spectrum by reducing the frequency resolution - but does not add artifact peaks due to frequency aliasing- as we illustrate below.

New Supplementary Figure S10 - Effect of sampling rate and scan duration on the power spectrum of the temporal signature associated with a principal component. (Top) The raw TxN fMRI signals with $N=1463$ voxels and $T=15501$ time points are multiplied by the $N \times 1$ principal component, resulting in a $1 \times T$ time series. The associated power spectrum is shown on the right (blue), with frequency resolution of 0.0064Hz and a Nyquist frequency of 13.1579 Hz (vertical dashed line) using the default parameters of Welch's power spectral density estimate in Matlab. The frequency axis is in log scale to highlight both low and high frequency components. On the far right panel, the power spectrum is reported for a time series obtained by multiplying the same principal component to the raw signals of a postmortem scan (red). (Middle) The same analysis is performed by considering the same sampling rate but only $1/10$ th of the total scan duration, resulting in 1551 time points. With reduced scan duration, the Nyquist frequency is still 13.1579 Hz but the frequency resolution is reduced to 0.0514 Hz . (Bottom) The same analysis is performed by reducing the sampling rate by $1/10$ th while keeping the original scan duration, resulting also in 1551 time points. With reduced sampling rate, the Nyquist frequency is 1.3158 Hz and the frequency resolution is 0.0051 Hz . This analysis shows that while long scans with low sampling rate may have the same (or even higher) frequency resolution as the original fast sampled signal, the undersampled signals can exhibit artifacts resulting from aliasing of high frequency components in the signal both from physiological sources (breathing and heartbeat) or from scanner artifacts (present also in postmortem scans).

Please note that in the downsampled signals, new peaks appear in the power spectra, particularly when the Nyquist frequency is below the breathing frequency (as shown in the new supplementary Figure copied below). While the frequency aliasing associated with scanner noise (here detected at 7.6 Hz in all scans, including postmortem) can be easily detected and removed, the components associated with physiological rhythms are more difficult to remove when undersampled, because they have high variability between animals

and even within sessions and are not purely sinusoidal, so can result in complex aliasing patterns.

New Supplementary Figure S11 - Effect of the sampling rate in frequency aliasing. The power spectra are calculated for the same signal (fixed scan duration of 10 minutes) with different downsampling factors $S=1,3,5,10,20,30,40$ ranging from the original Repetition Time (TR) of 38 ms (darker blue) to 1.52 seconds (by considering only one in every $S=40$ images) (cyan). As the downsampling S is increased, the Nyquist frequency is reduced as $f_{Nyq}=1/(2*S*TR)$. When the Nyquist frequency is below the breathing frequency (red line), the breathing frequency cannot be adequately resolved, and new frequency peaks appear in the power spectrum (see the black arrows for TR=380 ms).

Overall, our analysis shows that both scan duration and sampling frequency will affect the precision of the frequency analysis (as could be expected). As such, for an adequate analysis of oscillatory components, it might be preferable to both i) acquire long scan durations of several minutes to maximize the frequency resolution, but also ii) use as high sampling rate as possible to ensure a high Nyquist frequency and avoid aliasing of physiological rhythms such as the breathing. We now state this clearly in the Introduction and results sections.

This new analysis corroborates our previous analysis in Supplementary Figure 10 (below) where we found that for a TR of 380ms, the number of principal components detected (with eigenvalue above postmortem baseline) was strongly reduced from 10 to 4. This coincides with the Nyquist frequency (1.3 Hz) being below the breathing frequency (~2Hz), and we now better explain this.

Medetomidine

Medetomidine + isoflurane 1%

Fig. S10 - Effect of the sampling rate on the number and spatial signature of the principal components detected. Before the analysis, the fMRI signals recorded with a TR of 38 milliseconds (ms) were downsampled by considering only one in every 5, 10, 15, 20 and 25 frames (corresponding to intervals of 190, 380, 570, 760 and 950 milliseconds between frames). The number of principal components detected corresponds to the number of eigenvalues larger than the largest eigenvalue in postmortem scans (signals also downsampled). As can be seen, although some of the components can still be detected for TRs up to 950ms, the ultrafast sampling rate used in this work reveals a larger number of components, clearly defined within cortical and subcortical structures.

To summarize: given a fixed scan duration (which is typically defined by the experiment), a sampling rate that adequately resolves the breathing frequencies is recommended, even if the signal of interest is below the Nyquist frequency, because it prevents frequency aliasing from undersampled oscillatory components in the fMRI signals. While in rats the breathing frequency is around 2 Hz (for which a TR of 250ms at most is needed), in humans the breathing frequency is generally below 0.5 Hz, so sampling below 1 second should be sufficient to resolve it adequately. Moreover, although in this example the heartbeat frequency was not clearly detected, we note that our ultrafast sampling rate also adequately resolves the rats' heartbeat frequency (around 4~8 Hz) which may contribute to the added precision of our results. In humans, the heartbeat rate ranges between 0.75 Hz and 2.5 Hz, which would correspond to an optimal TR of 200 ms to resolve heartbeat components. If these components are adequately sampled, they can be removed more precisely, ensuring that the frequency range of interest in fMRI studies is not contaminated by the higher frequency components.

The new Figure 4 is shown below, using the unfiltered fMRI signals (in the previous version they were filtered below 0.5Hz) over 250 seconds (instead of 150s) and showing the whole frequency spectrum up to the original Nyquist frequency. We now state that this analysis assumes a fixed scan duration of 10 minutes.

Figure 4 – Effect of the sampling rate given a fixed scan duration. The temporal signal associated to the 7th principal component detected in ultrafast fMRI signals from medetomidine-sedated rats (Time of Repetition, TR= 38 milliseconds (ms)) is downsampled by considering only one in every 3, 5, 10, 20, 30, 40 and 50 frames (corresponding to intervals of 114, 190, 380, 760, 1140, 1520 and 1900 ms between frames). Plots shown for 250 seconds from a representative scan *S* (same from Figure 3). **Right.** The power spectral density (PSD) of the sampled signals for scan *S* (blue) and for a scan performed postmortem (red). PSD is normalized by the total power in the postmortem scan with the same sampling factor. PSD are computed over the entire scan duration of 590 seconds.

Modified text in the Introduction section:

‘Moreover, for increased precision in the characterization of oscillatory signals, long scanning times are needed to ensure high frequency specificity at low frequencies, and fast sampling helps preventing frequency aliasing from undersampled periodic components of physiological and/or scanner artifacts. At the spatial level, a large field of view is necessary to capture macroscale organization, while ensuring a sufficient spatial resolution to resolve distinct brain regions.

Therefore, we harness an ultrafast ultrahigh field fMRI approach, with long scan durations of 10 minutes sampled at 38 milliseconds (16000 frames per scan) to characterize the spatial

organization of oscillations detected in fMRI signals in a single slice of the rat brain, achieving high SNR ratio via a 9.4 Tesla magnetic field and a cryogenic coil.'

Modified text in the Results section:

'The detection of oscillations associated with the spatial patterns benefited from the fast sampling combined with long scan durations (totaling 16000 images per 10 minute scan), by preventing frequency aliasing from physiological rhythms (i.e., with Nyquist frequency above breathing and cardiac frequencies) and by ensuring sufficient resolution in the power spectrum at low frequencies, i.e., with precision below 0.01 Hz (see Supplementary Figures S10-11). As shown in Figure 4 (top row), when projecting the $1 \times N$ spatial component (here $\psi_{(\alpha=7)}$) on the $N \times T$ ultrafast unfiltered fMRI signals, the temporal signature $\tau_{(\alpha=7)}^S$ exhibits clearly visible oscillations between positive and negative representations of the spatial pattern. As the sampling factor is increased, even if a resonant peak frequency can still be detected, the signal to noise ratio is decreased (comparing with postmortem). Indeed, we find that the principal component $\psi_{(\alpha=7)}$ fails to be detected with a sampling as fast as 380 ms, which coincides with the sampling rate at which the breathing frequency (~ 2 Hz) cannot be adequately resolved given the Nyquist theorem (analysis shown in Supplementary figures S120-S164).'

Additionally, as I stated in the previous comment, fast sampling at a TR of 38 ms can incur a substantial penalty on the fMRI spatial resolution, which may also affect the specificity of derived oscillatory modes. The authors should be cautious about this recommendation.

Author Reply: We agree with the reviewer that the fast sampling can incur a penalty on the spatial resolution and we now better explain that here we are interested in the macroscopic organization of phase relationships occurring over extended brain areas and therefore we favor fast sampling *at the expense* of spatial resolution. Indeed, those are often traded off. Unlike works focusing on the precise localization of a signal response, here the micrometric precision is less relevant, because we are interested in the most macroscopic patterns of phase relationships extending over several millimeters (up to 1.2 cm between opposite sides of the brain slice) so our voxel resolution of 250 square micrometers is more than sufficient to resolve spatial patterns extending over several millimeters.

New text added in the Introduction:

'At the spatial level, a large field of view is necessary to capture macroscale organization, while ensuring a sufficient spatial resolution to resolve distinct brain regions.'

Previous comment R1.3 – generalizability of the study findings to awake humans: As acknowledged by the authors, the major resonant peaks at 0.2/0.25 Hz in the rodent data were induced by medetomidine, further rendering it unconvincing that the identified oscillatory modes can be generalized to awake human data without the anesthetic drugs.

Also, I may not agree with the authors that spontaneous activity in awake humans is periodic, based on reports of extensive literature including those cited by the authors. Therefore, the first paragraph of the new text in Introduction should be revised.

Author Reply: We agree with the reviewer that the strong resonance detected in sedated rats could be facilitated by medetomidine, which can induce strong LFP oscillations (as for example shown in Pradier et al, NeuroImage 2021). Importantly, we did not claim *'that the identified oscillatory modes can be generalized to awake human data without the anesthetic drugs'*, and we now try to make sure that this is not implied. Thus, we now revised the text to better explain our rationale:

1. The fact that the modes oscillate over several sustained cycles under medetomidine is indicative of an oscillatory process associated with those modes.
2. The modes detected in other conditions still exhibit qualitatively similar macroscopic patterns of phase relationships (and/or functional connectivity), but their temporal response **is significantly more damped**.
3. So we propose that medetomidine decreases the damping, revealing the oscillatory nature of the modes, which are otherwise overdamped under more conventional conditions and so just exhibit aperiodic activations.

We now clearly state that any extrapolation of these results to other conditions or species is purely hypothetical considering the spatial similarity of the modes detected across species, and that further experiments are needed to verify our hypothesis.

Modified text in the Discussion:

'The generalization of these findings to other animal species including humans can only be discussed in the light of existing literature and needs further experimental validation.'

[...]

'Even if no clear periodicity is detected in resting state fMRI in humans and the fluctuations closely approximate the canonical hemodynamic response function, one cannot exclude the possibility that the fluctuations reflect an overdamped oscillatory response associated with the transient and short-lived resonance of a stationary wave, providing a new generative hypothesis for the dynamic patterns observed empirically.'

As an aside, we'd like to point the reviewer back to our previous response where we confidentially included preliminary data from awake humans that in fact does show at least some of these oscillatory modes quite clearly; we hope that this will serve as a nice starting point for future studies!

Previous comment R1.4 – influence of hemodynamic blurring: While in the revision, the authors acknowledged the influence of hemodynamic blurring on the oscillatory modes, this concern has not been addressed. Given that the hemodynamic response functions are highly

variable across the cortex (e.g., Gonzalez-Castillo, J., Saad, Z.S., Handwerker, D.A., Inati, S.J., Brenowitz, N. and Bandettini, P.A., 2012. Whole-brain, time-locked activation with simple tasks revealed using massive averaging and model-free analysis. *Proceedings of the National Academy of Sciences*, 109(14), pp.5487-5492.) and obscure local spectral properties, it poses a concern regarding to which extent the identified oscillatory modes arise from neuronal origins.

Author Reply: Thank you for this important comment. Indeed, the extent to which the identified oscillatory modes arise from neuronal origins is unclear and therefore we have refrained throughout the text to claim a specific origin (neuronal, vascular, or other) for the oscillatory modes we have detected. Rather, we would like to argue that the oscillatory modes we detect in the signals are in a sense “driving”, or “underpinning” the correlations observed in resting-state fMRI - whatever their origin might be. In that sense the importance of this study is that it provides evidence for the existence of macroscopically-organized oscillatory modes that can be measured very reliably in rodents sedated with medetomidine. When adding a low concentration of isoflurane (as typically done in rat fMRI experiments to approximate more conventional resting-state conditions in humans and restoration of the neurovascular coupling to a state that resembles the coupling in awake systems), we find the oscillatory modes are more damped and oscillate at slower frequencies (~0.1Hz). This reinforces the fact that the anesthetics not only affect the neurovascular coupling, but also differentially modulate the frequency of oscillations.

We do surmise that the oscillations we detect are related to Kleinfeld’s ~0.1 Hz arteriole vibration frequency (Drew et al., 2020, Ultra-slow Oscillations in fMRI and Resting-State Connectivity: Neuronal and Vascular Contributions and Technical Confounds, *Neuron* 2020; Mateo et al, Entrainment of arteriole vasomotor fluctuations by neural activity is a basis of blood-oxygenation-level-dependent “resting-state” connectivity, *Neuron* 2017). While showing entrainment of vasomotor fluctuations due to LFPs locally, these experimental studies have not been able to investigate how the oscillations organize at the macroscale or whether the arteriole vibration frequency is increased under medetomidine.

In our study, this mechanism is only given as a hypothesis for the origin of the signals that needs to be tested in the future. In that sense, hemodynamic blurring may be slightly less relevant as the signal sources are already low-passed. However, we now further reinforce that the physiological origin of the signals is beyond the scope of the current work and leave this in the Discussion for future validation with experiments coupling in-vivo Ephys or calcium recordings and fMRI (our experiments in this vein are already ongoing! We hope to report the results in due course).

We do agree that whatever the origin is, hemodynamic blurring can be a confound. Thus, we have adapted the text to mention that the HRF is region-specific in the Introduction and fruitfully cited the papers suggested:

‘Crucially, it remains unclear whether the spectral power at low frequencies is associated solely with aperiodic activations of the characteristically slow and region-specific hemodynamic response function or additionally reflects the existence of damped oscillatory components (Gonzalez-Castillo, Saad et al. 2012, Lewis, Setsompop et al. 2016, Cabral, Kringelbach et al. 2017, Casorso, Kong et al. 2019).

Modified text in the Discussion: “...given that hemodynamic blurring is expected (Gonzalez-Castillo, J., Saad, Z.S., et al., 2012.) further local spectral properties may have been obscured by this blurring.”

Reviewer #2

The authors have responded clearly, thoughtfully, and substantially to my comments. The manuscript has fixed many inaccuracies in conceptual understanding (around the extracted components being 'based on wave physics') and terminology (now referring to the covariance modes as principal components). The claiming of novelty around parts of the study that are actually novel, and setting up of the unique problems that their analysis sheds light on (e.g., in Abstract and through the Introduction) is also much improved (although I have a few further queries on this below). It is clear from the updated results (Fig. 4) that my suggestion (mirroring other reviewers) that oscillations were accentuated by the application of narrow band-pass filtering is not correct—they are indeed very clear in the broadband signals (which are now plotted). Overall it is a major improvement: the structure is logical, clear, and correct, and the results are interesting.

Author Reply: We sincerely thank the reviewer for her/his positive comments, and appreciate very much their good words. We share the opinion that the manuscript has substantially benefited from the Reviewers' detailed suggestions. In this new version, we made our best efforts to address the remaining queries, please see below:

I have a smaller number of remaining responses. First to the reviewer responses, and second from a reading of the updated manuscript:

---REVIEWER RESPONSE---

A2.2: While I agree the spatial maps have received more attention than temporal properties, there is still substantial work involving modal decompositions that have characterized their oscillatory properties, including interpreting the properties of their power spectra (from neural field theory treatments, and a comparison to PCA), and damping times and oscillatory periods from dynamic mode decomposition (including a few examples from neural field theory, listed below). The text could be rephrased to more fairly cover prior work that has performed detailed characterizations of the oscillatory temporal properties of spatial modes (and clarifying the unique parts of this work e.g., phase relationships). The novelty of this study could then be refined given that NFT modes are associated with a power spectrum and DMD modes are constructed (as \sim rotations of PCA modes) to have coherent standing-wave oscillations.

- Gabay NC, Babaie-Janvier T, Robinson PA. Dynamics of cortical activity eigenmodes including standing, traveling, and rotating waves. *Physical Review E*. 2018 Oct;98(4):042413.
- Mukta KN, MaLaurin JN, Robinson PA. Theory of corticothalamic brain activity in a spherical geometry: Spectra, coherence, and correlation. *Physical Review E*. 2017 Nov;96(5):052410.

Author Reply: We thank the reviewer for guiding us towards important previous literature addressing oscillatory properties of spatial components, which we agree should be cited. We further note that the references mainly provide theoretical predictions of EEG power spectra (i.e., the alpha and delta peak), and while important and relevant, do not predict or show the frequency of macroscale ultra-slow oscillations in fMRI signals. Still the relationship is clear, and so we have added the following text in the Introduction (and further comment on this in the discussion when addressing the following replies):

'Mainly detected with electro- and magnetoencephalography (EEG/MEG), macroscale oscillatory components in brain activity have been targeted by neural field theories, demonstrating how the frequency spectrum and correlation structure can be predicted from brain geometry (Mukta, MacLaurin et al. 2017, Gabay, Babaie-Janvier et al. 2018, Tewarie, Abeysuriya et al. 2018). Although the intrinsic modes detected with fMRI have been shown to spatially align with eigenmodes of brain structure (either from surface geometry of diffusion networks), theoretical predictions of mode-specific temporal responses remain to be adequately addressed in fMRI (Friston, Kahan et al. 2014, Atasoy, Donnelly et al. 2016, Robinson, Zhao et al. 2016, Xie, Cai et al. 2021).'

A major comment comes from this—this study looks at the temporal properties of PCA modes directly (which do not necessarily have to exhibit oscillations), and shows they indeed have interesting oscillatory properties... But DMD already solves this problem of 'rotating' the PCA modes to yield standing-wave modes, so if they are of interest, why not study the DMD modes directly? This concern recurs in the authors response in A2.14—unclear why the authors perform PCA (which doesn't extract resonance/standing-wave modes) instead of DMD (which does). A clear explanation/justification for why PCA is chosen instead of DMD is required.

Author Reply: Thank you for pointing this out. We now clarify that it is precisely the fact that the temporal signatures *'do not necessarily have to exhibit oscillations'* that makes PCA adequate for our study. This approach allowed to first determine the PCs representative of each condition, and only subsequently to analyze their temporal signatures in each scan in terms of power spectrum, peak frequency and resonant quality. Furthermore, unlike DMD, PCA allows for the same component to oscillate at different frequencies over time (which we did observe under MED/ISO1%) and to detect components without sinusoidal response (as observed with MED/ISO3%).

We also wish to add that the intrinsic modes of covariance (or more colloquially PCs) are straightforward to obtain from simple linear algebra, i.e., $\text{eig}(\text{cov}(X))$, which we see as an advantage for future replicability. Further, assumptions of orthogonality intrinsic to PCA (unlike DMD), aligns with our hypothesis - supported by NFT - that they represent stationary waves, which should obey to the general principle of wave superposition.

As such, our argument is that although PCA is not specifically-designed to extract resonance/standing-wave modes from empirical data, it serves at an explorative level to detect orthogonal modes of covariance whose the oscillatory nature is unknown.

We now explain this in the results section:

'Principal component analysis has the advantage of returning orthogonal modes of covariance without making any assumption regarding the oscillatory properties of the components, unlike other decomposition techniques that a priori assume an oscillatory nature of the components, such as dynamic decomposition analysis (Schmid 2010, Casorso, Kong et al. 2019).'

A2.3: The claim that the authors can "expose the missing characterization of their temporal signatures", seems to overlook prior work on this (above) and should be revised.

Author Reply: Agreed, we should be more specific. Thus, we have now contextualized the novel empirical features that our analysis does expose with previous work, and reinforce that our empirical results provide strong evidence corroborating previous predictions pointing to macroscale oscillatory processes underlying resting state activity.

A major comment: A complete spatial randomization (Fig. S15) seems inappropriate: this is too extreme a null model. The data are so heavily spatially embedded (and thus strongly spatially autocorrelated) that, after shuffling them in a completely unconstrained way, it's unsurprising that it also destroys the temporal structure. Would spatially constrained nulls be more appropriate—holding generic spatial autocorrelation fixed, are these particular spatial patterns surprising?

cf. <https://www.biorxiv.org/content/10.1101/2020.08.13.249797v1>

Author Reply: Thanks for pointing this out. Following the Reviewer's comment, we have now also reordered the signals in the components from left to right, thus explicitly defining a spatial gradient. We note however, that this reordering using a spatial gradient may approximate a different component, and therefore the temporal signature may still exhibit oscillations, but uncorrelated with the original signal and with lower amplitude. Given that the first principal components represent the modes of covariance with larger associated energy (i.e. with larger eigenvalue), it is expected that any shuffling of the spatial patterns will incur in a loss of energy/power/amplitude in the associated signals.

Fig. S15 - The oscillations are specific to the spatial patterns. To demonstrate that the oscillations are specific to the spatial wave patterns detected, we reorder the elements in 3 representative spatial patterns by randomizing the voxels or sorting them according to the phase relationships, ensuring a spatial gradient. This analysis demonstrates that, when projecting the randomized spatial patterns (with size $1 \times N$) to the same, unfiltered, fMRI signals (with size $N \times T$), no oscillations are detected in the associated temporal signature (with size $1 \times T$), with the power being close to what is detected from a postmortem scan (red lines). The power spectra are shown in both linear and logarithmic (log) scales. When the elements are sorted to keep a spatial gradient (increasing from left to right), we still find oscillations, but with lower amplitude and uncorrelated with the original temporal signature. This occurs because the sorted patterns approximate a different intrinsic spatial pattern.

A2.17: Authors claim modal decompositions have not been applied to fMRI. This has indeed been done. E.g., should incorporate Henderson et al. (2022) as an example of an application of spatiotemporal modal decomposition to fMRI, and how the predictions of neural field theory relate to PCA modes. Given the conceptual similarities to the current paper, perhaps greater discussion of it (and/or similar work) is warranted [optional].

- Henderson JA, Aquino KM, Robinson PA. Empirical estimation of the eigenmodes of macroscale cortical dynamics: Reconciling neural field eigenmodes and resting-state networks. *Neuroimage: Reports*. 2022 Sep 1;2(3):100103.

Author Reply: Thank you for pointing to this paper, which was not published by the time of the previous submission. Indeed, the work is very relevant since it shows high similarity between the modes detected empirically from data (as covariance eigenvectors) and the Helmholtz modes predicted from surface geometry. This new work reinforces our hypothesis that the modes are shaped by the anatomical structure and we are now citing it in the discussion (please see below).

We also add that the Helmholtz eigenmodes are computed from the brain surface mesh alone, and therefore do not explain why the modes would differ across conditions (given that the brain geometry is constant across conditions). So we comment that future theoretical models should incorporate additional features such as the properties of the medium (i.e. resistivity), which are known to also contribute to the frequency and spatial shape of the modes.

We have added the following paragraphs in the Discussion:

‘The oscillatory modes detected were found to be consistent across rats within the same anesthetic condition, but to vary terms of spatial configuration, peak frequency and damping coefficient across conditions. Despite these differences, the modes detected across conditions are qualitatively similar in terms of organization through phase gradients within anatomically defined cortical and subcortical boundaries, indicating they likely share a common generative principle. These results align with neural field theories for macroscale brain dynamics describing large-scale wave propagation of neuronal activity including a spatial Laplacian to incorporate the brain geometry of the brain (Jirsa and Haken 1996, Robinson, Rennie et al. 1997, Deco, Jirsa et al. 2008, Gabay and Robinson 2017). While these neural field theories have historically been used to describe macroscale brain activity detected with EEG, recent studies suggest that the structural eigenmodes (defined either from brain surface geometry or from diffusion networks) may also be at the origin of macroscopic activity patterns detected with fMRI, namely the so-called ‘resting-state networks’ or ‘intrinsic connectivity networks’ (Friston, Kahan et al. 2014, Atasoy, Donnelly et al. 2016, Robinson, Zhao et al. 2016, Tewarie, Abey Suriya et al. 2018, Xie, Cai et al. 2021). This has been recently reinforced by a study demonstrating high spatial similarity between the covariance eigenvectors of fMRI signals and the theoretical prediction of Helmholtz eigenmodes of the Laplace-Beltrami operator starting from a brain surface mesh (Henderson, Aquino et al. 2022). Overall, these studies support our interpretation that the principal

components detected empirically from the fMRI signals are eigenmodes intrinsic to the brain structure, including not only the cortex but also subcortical structures, such as the striatum.'

and:

'The differences detected across anesthetic conditions question the theoretical predictions of modes depending on the brain geometric structure alone, because it is implicit that the anatomy of the brain is invariant across conditions. Indeed, it is generally known that the resonant modes of a system depend not only on the structural geometry of the system, but also on the resistivity of the propagating medium, which directly affects not only the spatial patterns, but also the resonant frequency and the stability of the oscillations. Given that anesthetics directly affect diverse properties of the brain tissue and vasculature, our results raise the importance to consider not only the brain geometry but also the resistivity of the medium through which the waves propagate to possibly explain the differences in resonant quality observed across anesthetic conditions..'

---TEXT---

- "the fundamental self-organizing principles" Consistent with the previous manuscript (in using words/phrases to hype/obscure a clear description of what is done) this phrase recurs through the manuscript. The use of the word 'self-organizing' should be made clearer. I would favor 'organizing' over 'self-organizing' if the authors do not provide a model of self-organization, or if the connections to the field of self-organization cannot be made concrete (e.g., self-organization of larger structures from individual molecules in chemistry, or the concept of self-organized criticality in complexity theory, etc.).

Author Reply: We acknowledge that the 'self-organization was not demonstrated, and therefore replaced by *'organization'* throughout the manuscript.

- Typo in abstract: "was found to modulate by anesthesia".

Author Reply: Corrected to *'and is modulated by anesthesia level'*.

- Rephrase: "Dynamically, correlated" (P3) ["dynamically fluctuations" doesn't make sense].

Author Reply: Thanks for noticing, now corrected to *'In the frequency domain,'*

- Check all axes in plots—many are not labeled (e.g., vertical axes of 3e,f; 4); units should also be provided if possible.

Author Reply: We apologize for this oversight. We have now added labels to the vertical axes of the Power spectra in Figures 3,e f and 4.

- [optional] Carpet plots seem to contain interesting structure that is hard to visualize. Could consider (i) using a clearer color map to distinguish positive (red) from negative (blue) deflections; and/or (ii) reordering rows to reveal structure, as per Aquino et al. NeuroImage. 2020; 212:116614.

Author Reply: We agree with the Reviewer that interesting wave patterns can be seen when zooming into the carpet plots, which are not clear at the scale plotted. It is true that the visualization of the waves associated to each mode depend on the way the voxels are sorted, and we were just using the original order of the brain mask (i.e., sorting voxels from top to bottom, left to right).

Following the reviewers suggestion, we now reorder the voxels in panels a and b of Figures 3 and 5 according to the elements in the first Principal Component in each condition. We also add a zoom of the 60 voxels with larger values in PC1, over 70 seconds to illustrate the wavelet patterns associated with this mode:

We note that the visualization of wave patterns associated with the other modes would require reordering according to the other eigenvectors.

In addition, we now provide the .fig of all images, such that the readers can zoom to explore the wave patterns further.

- "1/e" is clearer than "e^{-1}" when using a superscript reference format.

Author Reply: corrected in the text.

- Should clearly define the autocorrelation function—most will not be used to a definition of autocorrelation that involves both real and imaginary components. In many cases in text and in caption, the typical usage is used (and shown in upper of Fig. 6c), but the plots in lower panels of 6c should be better explained/interpreted/labeled (interpreting sinks from Hilbert

transform—it is also unclear what this representation provides in addition to the conventional autocorrelation function).

Author Reply: We now better explain how we estimated the phase portrait of the autocorrelation function, and why this plot serves to link with the theory of dynamical systems and classify the responses according to the Poincaré diagram (see below). This also serves to introduce the concepts developed in the following section, where we purpose to model the temporal response by an oscillator in the subcritical part of a Hopf bifurcation (given that Hopf bifurcations are also typically represented in the phase plane).

Poincaré Diagram: Classification of Phase Portraits in the $(\det A, \text{Tr } A)$ -plane

New text in the Results section:

‘We use the Hilbert transform to obtain a representation of the autocorrelation functions in complex domain (with real and imaginary components) and plot the corresponding phase portraits (Figure 6c bottom). The representation of the phase portraits serves to classify the temporal signatures of the components within the framework of stability theory of dynamical systems, demonstrating that the components have a ‘spiral sink’ trajectory to equilibrium, according to the classification of phase portraits in the Poincaré diagram (Teschl 2012).’

- Some visualization error appears to have occurred in the rendering of Figure 8b—parentheses are all empty.

Author Reply: In our version of the manuscript this appears to be correct. Before publication we will of course double- and triple-check that all figures are properly rendered in the proofs, using higher resolution images where necessary. Thanks For noticing!

- Fig. 4: Should use mathematical notation for multiplication (x) rather than computer-code notation (*).

Author Reply: Thank you for noting this. Corrected.

Reviewer #3 (Remarks to the Author):

I believe the manuscript has much benefited by addressing the concerns of the reviewers. While I am still not 100% convinced by the claims made by the authors, considering the massive amount of work that has clearly gone into this analysis, I think it is fair to present it to the readers at this point and let them judge for themselves.

Author Reply: We appreciate the recognition of our efforts in improving the quality of the manuscript and validity of our findings. We hope these new corrections will make the paper even more convincing.

In line with the reviewer's thought to let the audience 'judge for themselves', we wish to share that we have put a preliminary version of our article online in BiorXiv and we have already been contacted by a group that replicated our results. I have asked for consent to share their email and figure:

On 17/10/2022, 17:13, "Yi Chen" <yi.chen@tuebingen.mpg.de> wrote:

Dear Dr. Joana Cabral,

wish this email finds you well!

My name is Yi Chen, currently a Postdoc at Max Planck Institute for Biological Cybernetics in Tuebingen. I am writing to consult if you have an interest to collaborate on a proposed project focusing on a multi-center whole-brain rs-fMRI at high-frequency bands in rodents.

It was great that you reported brain activity > 0.1 Hz fluctuation in your paper titled "Resonant waves drive long-range correlations in fMRI signals". We analyzed the whole brain rs-fMRI and did ALFF analysis on the data from ETH and Michigan State University. We observed brain dynamics > 0.1 Hz, and up to 0.3 Hz, both in rats and mice with different anesthesia from MSU and ETH, which are highly consistent with your results. Please check the attached image at your convenience.

We are excited about these results and plan to approach other labs for more rodent data and propose a manuscript to report the high frequency at the whole brain level across multi-centers. Since you are the pioneer in this topic, we would like to ask for your endorsement and if you have an interest to collaborate.

If your feedback is positive, I would love to organize a Zoom meeting to discuss more in detail in early November with you and Dr. Norman Scheel, my colleague at MSU.

Looking forward to your reply and wish you a nice evening!

Best regards,
Yi

ETH data, consistent with "Light anesthesia"

MSU data

Reviewer #1 (Remarks to the Author):

In this revision, the authors have addressed all my previous comments. I don't have any further questions.

Reviewer #2 (Remarks to the Author):

The authors have shown a great degree of stamina(!) and dedication to understanding, depth, and clarity in the two rounds of responses, which is highly commendable. The only remaining points I would request is:

- A clearer description of the rationale for the spatial randomization null (even if no more analysis on this point is done). limitations of the current method should be explained clearly, while citing relevant literature on options for spatiotemporal constrained randomization (e.g., cf.

<https://onlinelibrary.wiley.com/doi/epdf/10.1002/hbm.20045>,

<https://www.biorxiv.org/content/10.1101/2021.06.01.446561v1>, etc.)

- [OPTIONAL]: I marked as 'optional' a previous comment about a recently published paper, and I will do so again with this one (no action required, just in case it is of interest to the broader hypotheses from this work about modes as a signature of a potential mechanistic underpinning for brain dynamics): <https://www.biorxiv.org/content/10.1101/2022.10.04.510897v1>

Response to Reviewers for Nature Communications manuscript NCOMMS-21-27556B

We'd like to thank the Reviewers one final time for their useful comments along the review process. We are delighted that they found the paper ready for publication and have made the final small revisions that were asked, including those marked "optional".

REVIEWERS' COMMENTS

Reviewer #1 (Remarks to the Author):

In this revision, the authors have addressed all my previous comments. I don't have any further questions.

Author Reply: Thank you for all the insightful comments along the way!

Reviewer #2 (Remarks to the Author):

The authors have shown a great degree of stamina(!) and dedication to understanding, depth, and clarity in the two rounds of responses, which is highly commendable.

Author Reply: We appreciate that the reviewer recognized our efforts to improve the quality of our manuscript, and thank her/him for the kind words and insightful comments along the way – they really improved the work. Thank you!

The only remaining points I would request is:

- A clearer description of the rationale for the spatial randomization null (even if no more analysis on this point is done). limitations of the current method should be explained clearly, while citing relevant literature on options for spatiotemporal constrained randomization (e.g., cf. <https://onlinelibrary.wiley.com/doi/epdf/10.1002/hbm.20045>, <https://www.biorxiv.org/content/10.1101/2021.06.01.446561v1>, etc.)

Author reply: Thank you for this suggestion - we agree. We have revised the suggested literature and we agree that there may be additional ways to verify that the principal components resonate more – resulting in temporal signatures with stronger amplitudes - that any other pattern. Our example provides only 2 straightforward ways to illustrate this fact, but a full demonstration would require iteratively testing with a large number of possible patterns. Still, the fact that the repertoire of spatial patterns is obtained as the eigenmodes of covariance with largest eigenvalue (or power), indicates that none of the remaining eigenmodes will resonate with stronger amplitudes than the ones detected. We now explain this in new text inserted in the Discussion: 'This hypothesis could not, however, be fully validated in the current work, given that the eigenmodes of covariance were obtained from the fMRI signals alone and not compared with the ones predicted from the structure. Although these spatially-defined modes were found to be consistent across animals and to amplify the signal with respect to spatially-reordered vectors (i.e., either by randomization or by sorting the phases in a gradient from left to right), our validation tests using may not be sufficient to demonstrate that this is the adequate basis of spatial patterns to describe functional neuroimaging data and further validations are needed⁷⁷.'

We also added a note in the Discussion about the limitation that the relationship between the temporal signatures of the different modes was not addressed:

Text added in the Discussion: 'We note however, that the temporal coordination between the different modes was not addressed in the current work and deserves further investigation.'

- [OPTIONAL]: I marked as 'optional' a previous comment about a recently published paper, and I will do so again with this one (no action required, just in case it is of interest to the broader hypotheses from this work about modes as a signature of a potential mechanistic underpinning for brain dynamics): <https://www.biorxiv.org/content/10.1101/2022.10.04.510897v1>

Author reply: We thank the reviewer for pointing to this recent paper published on biorxiv, which is now cited and fully aligns with the interpretation made in the current work. In particular, it is interesting to note that different brain structures have their own harmonic modes. We are confident that these cumulative efforts to identify modes in brain dynamics and compare the patterns predicted analytically from the structure with the ones detected empirically from dynamic signals is a promising new avenue to obtain a mechanistic understanding of the rules governing brain dynamics.